# Tags for DAGs:
# Graph Refinement with Meta-Informed Relations

## Abstract

Causal discovery has shifted from data-centric methods to hybrid strategies that integrate semantic knowledge from experts or large language models (LLMs). Such external information is vital for identifying causal structures beyond the Markov Equivalence Class (MEC), which data alone cannot resolve. However, expert availability is often limited, and LLMs frequently misidentify causal directions in specialized domains. To overcome such shortcomings, we propose a tag-based approach that leverages semantically meaningful labels while deriving causal directionality directly from data. Using variable-level tag assignments from available sources (e.g., LLMs), our tags for DAGs method learns from identifiable data structures to extract higher-level causal relations. These are then used to orient undirected edges, enabling causal discovery to move beyond the MEC without reliance on fallible external knowledge.

## 1 Introduction

Understanding variables not just as sole columns of data, but rather as semantic entities that can share common properties with each other is something well understood by humans, who have been shown to often leverage meta-information about them when making predictions (Bonawitz et al., 2010; Meltzoff, 2007; Bailey et al., 2024). Knowledge about a particular semantic *type* of a variable can be highly indicative of certain causal interactions. For example, demographic factors usually cause consumer behavior, socioeconomic policies typically drive individual financial behaviors, and biological traits are a cause of symptoms. Even without domain knowledge, the relation between *Bandwidth* and *Latency* can be better explained by knowing meta-statistics of *Infrastructure* commonly causing *Performance Metric*.

The task of causal discovery (CD) aims to recover causal structures from data (Spirtes & Glymour, 1991; Hoyer et al., 2008; Mooij et al., 2011; Schölkopf et al., 2021) and while extensive consideration has been directed at recovering causal graphs from data alone, results derived from Pearl's do-calculus (Verma & Pearl, 1990; Frydenberg, 1990; Pearl, 2009) limit the identification of causal structures from purely observational data, such that the direction of some edges can remain unknown. Such edges could, for example, be directed with the aid of experts or through additional experimental data. However, expert time is scarce, and additional data sources can be costly to obtain. To still improve graph recovery, more accessible forms of information need to be considered to further utilize the already available data. Towards this, we propose a tagging-

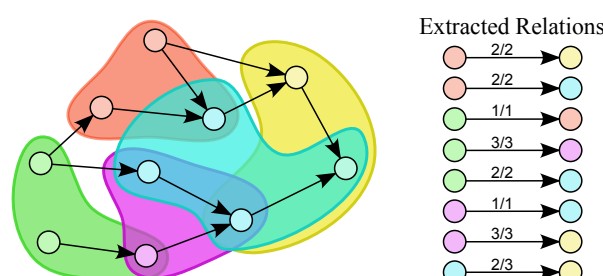

Figure 1: **Collecting Tag Information.** Our tagging approach considers the assignment of multiple tags per variable, (1) allowing for the flexible assignment of light-weight meta-information, and (2) repurposing already discovered directed edges for identifying higher-level causal relations.

based approach on top of standard causal discovery that further leverages meta-level information induced by relations formed between the semantic types of groups of variables (Fig. 1).

We follow the philosophy that it is more economical to leverage available data to its full extent by supplementing it with light-weight tagging information before requesting additional experimental data or repeatedly querying external experts for their support in providing additional information on individual causal relations. The idea of using the semantic knowledge of the variables for causal discovery is not new. Following the ongoing development of large language models (LLMs), causal discovery has been opened up to incorporate ubiquitous external knowledge that was previously hard to provide in practice. With arguments for and against these models yielding notions of 'true' causal understanding (Zhiheng et al., 2022; Zečević et al., 2023; Lampinen et al., 2024; Joshi et al., 2024; Chi et al., 2024), LLMs have been utilized to predict the direction of edges in causal discovery. While such information can be helpful, approaches based purely on LLMs have been found to struggle in practice when tasked with performing causal inference or discovery (Kıcıman et al., 2023; Jin et al., 2023; Zečević et al., 2023). More recently, hybrid approaches that jointly leverage existing data and the knowledge learned by LLMs have been shown to be more reliable (Brouillard et al., 2022; Wang et al., 2024; Clivio et al., 2025). Our tagging approach falls into this latter category. By making use of classical CD to recover the initial causal graph and only then to determine the edge direction based on tagging-induced statistics, our approach is 'data first', circumventing possible deficits in LLMs' learned causal domain knowledge.

**Contributions.**

1. We introduce a theory for extracting high-level causal relations from existing causal graphs by leveraging meta information in the form of tags and using this to narrow down the MEC.

2. We propose an algorithm that implements this idea and includes several strategies for increasing robustness to graph faults and noisy tagging information.

3. We demonstrate the conservative robustness of our tagging approach through ablations, inducing severe failures in the underlying recovered causal graph structure and the tagging assignment itself.

4. We show how the discovered tag relations extract meaningful causal relations on abstract concepts.

Our code is made available at `https://anonymous.4open.science/r/tagging_causality-EC75`.

## 2 Background and Related Work

**Notation.** Throughout the paper, we denote univariate variables with uppercase letters $(X, Y)$ and their realizations as lowercase letters $(x, y)$. (Indexed) sets are indicated in boldface, $(\mathbf{X}; \boldsymbol{\mathcal{X}})$, and their realizations as lowercase boldface letters $(\mathbf{x})$. Individual elements are denoted with subscript, $X_i \in \mathbf{X}$; $X_i \in \boldsymbol{\mathcal{X}}$.

**Causal Discovery.** A causal graph $\mathcal{G}$ is an acyclic directed graph (DAG) that represents the structure of cause-effect relations over an indexed set of variables $\boldsymbol{\mathcal{X}}$ (Spirtes et al., 2000; Pearl, 2009). For every cause-effect relation between variables $X_i, X_j \in \boldsymbol{\mathcal{X}}$, individual edges $(X_i, X_j) \in \boldsymbol{\mathcal{X}} \times \boldsymbol{\mathcal{X}}$ in the graph point from the cause $X_i$ to the effect $X_j$. Causal discovery is concerned with identifying the underlying causal graph from data (Peters et al., 2017). Even when disregarding the presence of hidden confounders, causal models can often only be discovered up to Markov equivalence (Verma & Pearl, 2022; 1992) from observational data without further assumptions. Completed partially directed acyclic graphs (CPDAGs) comprise all Markov equivalent DAGs that are admissible under the given data but leave some edges undirected (Andersson et al., 1997). The approach in this paper aims to refine CPDAGs in an attempt to further direct remaining undirected edges. A primary class of causal discovery methods is constraint-based algorithms, which utilize independence tests to recover the graph skeleton. We will rely on the well-known PC algorithm (Spirtes & Glymour, 1991), which performs skeleton and immorality discovery followed by Meek rules (Meek, 1995). Another primary class is score-based algorithms that find the best graph by maximizing a particular scoring function. A common representative of this class of methods is greedy equivalence search (GES), which greedily adds and removes edges, starting from an empty graph (Chickering, 2002).

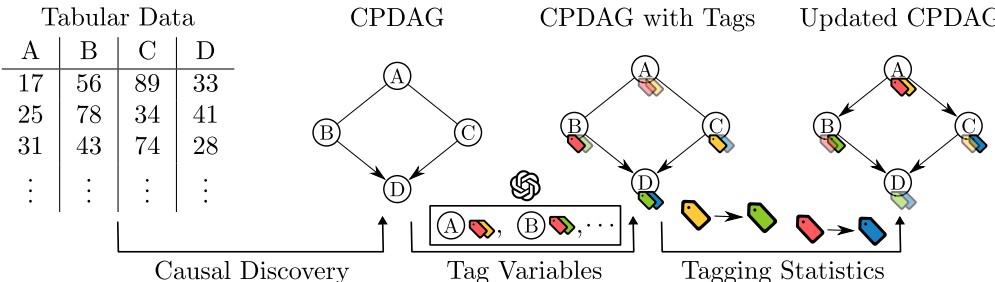

Figure 2: **Tag-Supported Causal Discovery.** The figure illustrates the tagging-based causal discovery described in this paper: First, a causal discovery algorithm such as PC or GES is used to obtain a CPDAG. Variables are tagged with semantically meaningful tags, for example, by using an LLM. Lastly, tag relations on the CPDAG are used to predict the direction of undirected edges.

**Breaking Markov Equivalence.** Additional assumptions are needed to break Markov equivalence when trying to recover causal graphs from observational data and moving beyond CPDAGs by leveraging background knowledge (Perković et al., 2017). Here, the provision of interventional data (Hauser & Bühlmann, 2012; Brouillard et al., 2020) or additional knowledge about the types of causal relations and noise can be used (Shimizu et al., 2006; Hoyer et al., 2008). Furthermore, meta-considerations about the structure of graphs (Mansinghka et al., 2012) or edge directionality based on CPDAG conforming model counting (Long et al., 2023) have been proposed. Recently, the use of meta-information based on variable names has been suggested. Here, Brouillard et al. (2022) put forward the idea of assigning particular *types* to variables and using these types to enforce a *type consistency* for edges such that one type can not be a cause and an effect of a second type at the same time. While our work is close to Brouillard et al. (2022) in that we equip variables with meta-information, our tag-based strategy has several advantages over the purely type-based approach. For one, we can assign as many tags as we can derive from meta knowledge, without being constrained to a single type per variable. Additionally, Brouillard et al. (2022)'s *type consistency* assumption, which enforces a unique causal edge direction between types, is often violated in practice and results in incorrect graphs. Our tagging approach employs a probabilistic approach for dealing with tagging inconsistencies, calibrating the impact of individual tagging pairs on the recovered graph. Since 'faulty' tag relations can be overruled by others, our algorithm becomes robust to faults in graph recovery and tag assignment.

**Relation to Causal Abstractions and Tiered Background Knowledge.** Section 4.3 is concerned with the extraction of abstract causal relations from known, tag annotated, graphs. Intuitively, one could think of our tagging approach as a way to extract higher-level causal models (Rubenstein et al., 2017; Beckers & Halpern, 2019) with the help of individual tags as the new nodes. As such, our work would be related to works on cluster DAGs (Anand et al., 2023; Wahl et al., 2023; Massidda et al., 2024; de Vargas et al., 2025) that are primarily concerned with the identifiability of causal effects, or view tags as a form of self-inducing tiered constraints on the causal graph (Andrews et al., 2020; Mooij et al., 2020; Bang & Didelez, 2023). In contrast to these prior works, our approach does not require 'hard' knowledge about any higher-level clusters or tiered background knowledge constraints. The 'higher-level' tag statistics arise solely from variable annotation and causal graph discovery on the lower level. Furthermore, our approach does nor require, neither enforce assumptions on the acyclicity of the higher-level model. Most importantly, most prior works use the strong notion of 'constructive' causal abstractions (Beckers & Halpern, 2019) that assume compact, non-overlapping partitions of low-level variables. In these cases, typing of Brouillard et al. (2022) would suffice, as every variable would be uniquely assigned to a partition. With tagging, however, variables not only share tags but are usually assigned multiple tags at the same time, such that individual nodes do not belong to a unique cluster or tier, breaking the partition requirement of constructive abstractions.

**Causal Discovery and LLM.** Multiple works have investigated the abilities of LLM to reason about causal concepts (Kıcıman et al., 2023; Jin et al., 2023; Zečević et al., 2023; Gendron et al., 2024) including the direct questioning of LLMs for causal discovery (Long et al.; Kıcıman et al., 2023; Zečević et al., 2023; Jiralerspong et al., 2024) concluding mixed performance and motivating the need for approaches that jointly leverage statistical methods and LLMs (Choi et al., 2022; Long et al., 2023; Mathur et al., 2024; Clivio et al., 2025).

## 3 Tagging Informed Edge Direction

The underlying assumption of our approach is that observing causal directions between higher-level concepts can inform decisions about the direction of causality of their lower-level realizations. In particular, the causal directions discovered by an initial CD algorithm can be used to compute preferences between higher-level tag relations, which can then be leveraged for predicting edge directions. Fig. 2 shows the basic approach.

One such approach by Brouillard et al. (2022) assigns a single *type* to each variable to gather the necessary statistics. Under the assumption of *type consistency*, all edges between variables of type $A - B$ are assumed to be directed in the same way. This rather strong assumption has several drawbacks. First, it can never provide information about the relations between variables of the same type. Second, it cannot consider particular aspects shared between variables of different types since only one type can be assigned to any variable. Therefore, in this work, we generalize the notion of types by associating multiple *tags* $\mathbf{t}$ to every variable $X_i$. Tags are assigned by some external mechanism $\mathbf{t} := \text{tag}(X_i)$, such that individual tags $\text{t}^a \in \mathbf{t}$ are general concepts such as "Symptom" or "Demographic Information", expressed in natural language. Tagging allows for drawing more evidence from a causal graph by considering multiple, possibly noisy tag relations between variables rather than a single type relation.

The starting point of the approach is an existing CPDAG. An edge $E_{ij}$ between the variables $X_i, X_j$ either indicates $X_i$ causing $X_j$ ($X_i \rightarrow X_j$), $X_j$ causing $X_i$ ($X_j \rightarrow X_i$) or the edge being undecided ($X_i - X_j$). Edge direction is generally denoted as $d_{ij} \in \{-1, 0, 1\}$, where a value of 1 indicates $X_i \rightarrow X_j$, $-1$ indicates $X_j \rightarrow X_i$ and 0 indicates $X_i - X_j$. The following symmetry holds: $d_{ji} = -d_{ij}$.

**Notation:** The following section describes how tagging statistics are extracted from different subsets of variables. To differentiate various tags and tagging statistics we use the following notation: (1) We write superscripts $(t^a, t^b, \dots)$ to indicate unbound arbitrary tags. (2) We use subscripts to connect tags and variables: $\mathbf{t}_i$ is the set of tags assigned to variable $X_i$. (3) We use sub- and superscript in conjunction to reference specific tags of a variable's tag set: $t_i^a \in \mathbf{t}_i$.

**Structure.** This first part of the section is concerned with the theoretical modeling and the required underlying assumptions of the approach. The resulting practically applied formulas that no longer track noise statistics explicitly are presented Sec. 3.1.

**Tag Informative Value.** In order to transfer information about the observation of one edge's direction onto another edge, the level of similarity between both needs to be assessed. Here, the overlap in the tagging sets of both variables serves as the common denominator of edge similarity. Consequently, the edge direction is assumed to be the result of some *edge direction function* $f$ with $d_{ij} := f(\mathbf{t}_i, \mathbf{t}_j, N_{ij})$, taking the tag sets of both variables and an edge-specific noise term $N_{ij}$. Consequently, our approach assumes that $d_{ij}$ can be predicted from knowledge of $\mathbf{t}_i, \mathbf{t}_j$ and that the influence of individual $N_{ij}$ is rather small.

> **Informal Idea:** The overall idea of the following *Tag Informative Value* (Def. 1) is to compute the fraction of observed directed edges pointing from $t^a$-tagged nodes to $t^b$-tagged nodes (instead of pointing from $t^b$ to $t^a$).

Beyond the informal description, the following notation is provided to formally capture the (possibly varying) tagging statistics within different subparts of the graph. These terms will be required for proving the correctness of the approach in the variable limit of $|\mathbf{X}| \rightarrow \infty$.

When observing a CPDAG, one is interested in quantifying how much the observation of a particular pair of tags attached to some edge informs about the directionality of other edges when observing that particular tag pair again. For this, a *tag informative value* is defined, which measures the probability of observing a particular pair of tags with respect to a particular edge direction, $(t_i^a, t_j^b) \sim d_{ij}$.

The following tag informative value is defined as a conditional probability with respect to (some subset of) the graph. As we are trying to compute statistics on edge directions, a sample space for a node set $\mathcal{Y} \subseteq \mathcal{X}$ is the set of all edges between variables in $\mathcal{Y}$ in that involve tags $\text{t}^a$ and $\text{t}^b$. We use superscript $\cdot^{\mathcal{Y}}$ to indicate expressions that are constraint to the (sub)set of edges between $\mathcal{Y}$. First, an auxiliary function is defined

to determine whether any two variables are tagged with the tags of interest: $\text{tagged}(X_i, X_j, \text{t}^a, \text{t}^b) := (\text{t}^a \in \text{tag}(X_i)) \wedge (\text{t}^b \in \text{tag}(X_j))$. Then, out of the set of edges between $\mathcal{Y}$, Eq. 1 filters the set of directed edges where either endpoint carries tag $t^a$ and the other carries $t^b$:

$$\mathbf{E}^{\mathcal{Y}, \text{t}^a, \text{t}^b} = \{(X_i, X_j) \in \mathbf{E} \mid X_i \in \mathcal{Y} \wedge X_j \in \mathcal{Y} \wedge (\text{tagged}(X_i, X_j, \text{t}^a, \text{t}^b) \vee \text{tagged}(X_j, X_i, \text{t}^a, \text{t}^b))\} \qquad (1)$$

and Eq. 2 filters the edges where exactly the starting point carries tag $t^a$ and the endpoint carries tag $t^b$:

$$\mathbf{E}^{\mathcal{Y}, \text{t}^a \to \text{t}^b} = \{(X_i, X_j) \in \mathbf{E} \mid X_i \in \mathcal{Y} \wedge X_j \in \mathcal{Y} \wedge \text{tagged}(X_i, X_j, \text{t}^a, \text{t}^b)\} \qquad (2)$$

In particular, $\mathbf{E}^{\mathcal{Y}, \text{t}^a \to \text{t}^b} \cap \mathbf{E}^{\mathcal{Y}, \text{t}^b \to \text{t}^a} = \emptyset$ and $\mathbf{E}^{\mathcal{Y}, \text{t}^a \to \text{t}^b} \cup \mathbf{E}^{\mathcal{Y}, \text{t}^b \to \text{t}^a} = \mathbf{E}^{\mathcal{Y}, \text{t}^a, \text{t}^b}$, such that $|\mathbf{E}^{\mathcal{Y}, \text{t}^a \to \text{t}^b}| + |\mathbf{E}^{\mathcal{Y}, \text{t}^b \to \text{t}^a}| = |\mathbf{E}^{\mathcal{Y}, \text{t}^a, \text{t}^b}|$ can be used to normalize probabilities on the following tag information value:

**Definition 1. Tag Informative Value.** *A pair of tags $(t^a, t^b)$ over a set of variables $\mathcal{Y} \subseteq \mathcal{X}$ has tag informative value*

$$I^{\mathcal{Y}}(t^a, t^b) := \frac{|\mathbf{E}^{\mathcal{Y}, \text{t}^a \to \text{t}^b}|}{|\mathbf{E}^{\mathcal{Y}, \text{t}^a, \text{t}^b}|}.$$

Put in simple words, $I^{\mathcal{Y}}(t^a, t^b)$ is the probability of any edge $X$—$Y$ in $\mathcal{Y}$ carrying tags $t^a, t^b$ to be directed as $X_i \to X_j$. In the following, we assume that the direction of an edge $d_{ij}$, annotated with tags $(\text{t}_i^a, \text{t}_j^b)$, is more likely to also be directed $X_i \to X_j$ ($d_{ij} = 1$) if a high percentage of similar edges in $\mathcal{Y}$, carrying tags $\text{t}_i^a, \text{t}_j^b$, have previously been observed to point in the same direction $\text{t}_i^a \to \text{t}_j^b$.

Some particular combinations of tags $(t^a, t^b)$ might never be observed ($\mathbf{E}^{\mathcal{Y}, \text{t}^a, \text{t}^b} = \emptyset$), making $I^{\mathcal{Y}}(t^a, t^b)$ undefined. We filter out those cases by an indicator function $o^{\mathcal{Y}}(t^a, t^b) := \mathbb{I}(\mathbf{E}^{\mathcal{Y}, \text{t}^a, \text{t}^b} \neq \emptyset)$, where $\mathbb{I}$ is 1 if $\mathbf{E}^{\mathcal{Y}, \text{t}^a, \text{t}^b} \neq \emptyset$ and 0 otherwise (with $I^{\mathcal{Y}}(t^a, t^b)$, in the absence of evidence, being defaulted to 0.5). When observing a novel, previously undirected edge with tag sets $\mathbf{t}_i, \mathbf{t}_j$, we can now use the tag information value to reason about the true edge direction.

> **Informal Idea:** For an undirected edge $X_i$—$X_j$, we infer the *preferred edge direction* (Eq. 3) as the average of tag information values between all tagging pairs $(\text{t}^a, \text{t}^b) \in \mathbf{t}_i \times \mathbf{t}_j$.

When assuming the underlying edge direction function $f$ to be realized as the average tag informative value of all pairs of tags in $\mathbf{t}_i, \mathbf{t}_j$ –filtering out unobserved tagging pairs via the indicator function $I^{\mathcal{Y}}(t^a, t^b)$– the *preferred edge direction* is computed as follows:

$$d_{ij}'^{\mathcal{Y}} = \frac{\sum_{(\text{t}^a, \text{t}^b) \in \mathbf{t}_i \times \mathbf{t}_j} I^{\mathcal{Y}}(\text{t}^a, \text{t}^b) o^{\mathcal{Y}}(\text{t}^a, \text{t}^b) + N_{ij}}{\sum_{(\text{t}^a, \text{t}^b) \in \mathbf{t}_i \times \mathbf{t}_j} o^{\mathcal{Y}}(\text{t}^a, \text{t}^b)} \qquad (3)$$

Since $I^{\mathcal{Y}}(\text{t}^a, \text{t}^b)$ is already normalized, it does not require another normalization term in the denominator. As $d_{ij}'^{\mathcal{Y}}$ is still a continuous quantity, we discretize it into one of the possible states captured by $d_{ij}^{\mathcal{Y}}$: $d_{ij}^{\mathcal{Y}}$ is set to 1 (indicating $X_i \to X_j$) for $d_{ij}'^{\mathcal{Y}} > 0.5 + \epsilon$, $d_{ij}^{\mathcal{Y}} := -1$ (indicating $X_j \to X_i$) for $d_{ij}'^{\mathcal{Y}} < 0.5 - \epsilon$ and no decision on the direction ($d_{ij}^{\mathcal{Y}} = 0$) is made whenever $0.5 - \epsilon \leq d_{ij}'^{\mathcal{Y}} \leq 0.5 + \epsilon$, for some small $\epsilon \in \mathbb{R}_{>0}$.

Different subsets of edges might yield different edge statistics. To reliably predict the direction of undirected edges from observed evidence, it must be assumed that edges in the directed and undirected cases both follow the same statistics:

**Assumption 1. Tag Distribution Consistency.** *For any two sufficiently large subsets $\mathcal{Y}', \mathcal{Y}'' \subseteq \mathcal{X}$ and some small $\varepsilon \in \mathbb{R}_{>0}$, the tag informative values of both sets converge towards each other: $\forall (\text{t}^a, \text{t}^b) \in \mathbf{t}_i \times \mathbf{t}_j$. $\lim_{|\mathcal{Y}'|, |\mathcal{Y}''| \to \infty} |I^{\mathcal{Y}'}(\text{t}^a, \text{t}^b) - I^{\mathcal{Y}''}(\text{t}^a, \text{t}^b)| < \varepsilon$.*

The tag distribution consistency assumption asserts that information about the edge direction of yet undirected edges is related to that of already directed edges. I.e., it rejects scenarios in which the probability of

directing edges correlates with their directionality. This assumption is usually not verifiable in practice, but we assume it holds under mild conditions, e.g., when applying causal discovery methods to domains with sufficiently diverse structures.

## 3.1 Tagging Approach

Our algorithm follows the definitions of the tag informative value (Def. 1) and preferred edge direction (Eq. 3) by first collecting statistics on the directional preferences between tagging pairs from the directed edges of a CPDAG and subsequently using this information to direct the remaining edges one by one. We assume the underlying CPDAG to be sound. Our algorithm therefore tackles the problem of refining the graph beyond the observational Markov equivalence with the use of additional tagging information. We do not intend to modify to the underlying skeleton by adding or removing any existing edges based on our tagging statistics. Due to the error-prone nature of many causal baseline algorithms, the observed tagging statistics might be inaccurate. Several strategies are employed during the construction of our algorithm in Sec. 3.2 to mitigate these issues. We provide the full pseudo-code of our algorithm in App. C.

**Collect Tagging Evidence.** Similar to the previous tag informative value (Def. 1) we first count the number of edges $(X_i, X_j) \in \mathbf{E}$ carrying tags $(t^a, t^b)$ for all tag pairs between all tags, $\mathbf{T} = \mathbf{t}_1 \cup \cdots \cup \mathbf{t}_N$, in the graph. This is stored in an $\mathbb{N}^{|\mathbf{T}| \times |\mathbf{T}|}$-matrix $C$ with entries:

$$C_{ab}^{\mathcal{Y}} := |\mathbf{E}^{\mathcal{Y}, t^a \to t^b}| \tag{4}$$

The following convergence limit of Eq. 7 suggests that all statistics should be computed over the largest possible set of evidence, that is, over all variables ($C_{ab}^{\mathcal{X}}$). If not stated otherwise, we therefore do not explicitly note the $\mathcal{X}$-superscript in the following, and simply write $C_{ab}$ instead. Different to the tag informative value, we do not directly normalize direction statistics, but retain the absolute edge counts in $C$. This helps us to later reject direction decision that would be based on little evidence, e.g., a single observed edge pair, making the algorithm more robust in practice. The observable finite sample correspondent to the tag informative value is then called $\hat{p}_{ab}$ and estimated as:

$$\hat{p}_{ab} = \frac{C_{ab}}{C_{ab} + C_{ba}} \tag{5}$$

Similar to $I^{\mathcal{Y}}(t^a, t^b)$, a $\hat{p}_{ab}$ of 0.5 indicates no preference for either direction, while $\hat{p}_{ab} = 0$ and $\hat{p}_{ab} = 1$ indicate a full support for directing edges as $t^a \to t^b$ or $t^b \to t^a$, respectively.

**Decision Rule.** Similar to before, and to eventually decide on the directionality of an edge $\mathrm{E}_{ij} \in \mathbf{E}$, we compute the *evidence edge preference*, $Q(\mathrm{E}_{ij})$, as the finite sample correspondent to the preferred edge direction $d'^{\mathcal{Y}}_{ij}$ of Eq. 3:

$$Q(\mathrm{E}_{ij}) = \frac{\sum_{(t^a, t^b) \in \mathbf{t}_i \times \mathbf{t}_j} \hat{p}_{ab} \cdot o(t^a, t^b)}{\sum_{(t^a, t^b) \in \mathbf{t}_i \times \mathbf{t}_j} o(t^a, t^b)} \tag{6}$$

All edges not introducing cycles are directed as $\hat{d}_{ij} := 1$ (the hat, again, indicates assignment based on the observed statistics, which is possible different to the ground truth value) if $Q(\mathrm{E}_{ij}) > 0.5 + \alpha$; $\hat{d}_{ij} := -1$ if $Q(\mathrm{E}_{ij}) < 0.5 - \alpha$ and leaving the edge undirected ($\hat{d}_{ij} := 0$) otherwise for some $\alpha \in \mathbb{R}_{>0}$ Due to acyclicity constraints and the possible application of Meek rules, the order of directing edges matters. To prioritize edges with higher confidence, the algorithm greedily directs edges with the highest $Q(\mathrm{E}_{ij})$ values first.

**Asymptotic Guarantees.** So far, we illustrated our tagging approach and how it aims to predict the correct edge directions. Under Assumption 1, this can be shown to lead to correct edge predictions in the asymptotic limit. Let the tag informative values $\mathbf{I}^{\mathcal{Y}} = \{I_1^{\mathcal{Y}}, I_2^{\mathcal{Y}}, \ldots, I_n^{\mathcal{Y}}\}$ be sampled from a Beta distribution Beta$(\alpha, \beta)$. From the Central Limit Theorem, it follows that the variance of our prediction on $n$ is given as:

$$\mathrm{Var}[d'^{\mathcal{Y}}_{ij}] = \mathrm{Var}\left[\frac{1}{n} \sum_{m=1}^{n} I_m^{\mathcal{Y}}\right] = \frac{\alpha\beta}{(\alpha + \beta)^2(\alpha + \beta + 1)n}, \quad n = |\mathbf{t}_i \times \mathbf{t}_j| \tag{7}$$

Full derivation in App. A. An increasing number of tag pairs reduces the variance, thus improving the accuracy of the prediction. In the limit, we have $\text{Var}[d'^{\mathcal{Y}}_{ij}]_{n \to \infty} = 0$, which even leads to perfect predictions $\mathbb{P}[d'^{\mathcal{Y}}_{ij} > 0.5] = 1$. However, in practice, neither an infinite number of useful tag pairs nor enough edges to correctly estimate the tag informative scores are usually present. Next, we will introduce parameters aimed at mitigating these issues in practical scenarios with limited data.

## 3.2 Robustness Strategies

While our convergence results are a strong theoretical basis for the tagging approach, data and tag limitations can result in incorrect edge directions. To compensate for this, we introduce several strategies for leveraging the information provided by tags, while not relying on the perfect correctness of the CPDAG, and to increase robustness in settings with sparse and noisy tags. In our experiments, we will show that our implementation is quite robust to such settings.

**Absence of V-Structures.** Taking further inspiration from the previous algorithm of Brouillard et al. (2022) which enforced type consistency within variable triplets[1], we propose a variant of our method that considers nodes within triplets that contain the same tags as a source of information. Following our tagging procedure, we assign tagging evidence to any tag pair supported by an identified v-structure $A \to C \leftarrow B$. If an unshielded triplet is not identified as a v-structure, we assign evidence to any tag pairs in the opposite direction (corresponding to the fork structure $A \leftarrow C \to B$) for only those tag pairs where one tag is shared across both $A$ and $B$. We collect evidence using this procedure and then start directing edges on the skeleton, where no edges are yet directed, not even v-structures. The underlying idea is that, as all evidence still originates from v-structures, this information can be leveraged and combined in such a manner that the occasional incorrectly identified v-structure can possibly be overruled by overwhelming tag evidence from other structures. We denote this variant as 'AntiV' in our experiments, but find that it generally performs worse than directing edges on the CPDAG.

**Calibrating Conservativeness of the Algorithm.** Our theoretical guarantees hold in the sample limit and under sufficiently large graphs. Practical applications, however, are commonly limited by the availability of data and the size of the graph. Under these circumstances, evidence statistics might become noisy due to either erroneous tagging assignments or errors of the initial discovery graph. The actual $\alpha$ value mentioned in conjunction with Eq. 6 should depend on the system's inherent noise level and controls the '*conservativeness*' of the prediction. A low $\alpha$ leads to more, but possibly noisy, decisions, while a high $\alpha$ can make the algorithm abstain too often, leaving most undirected edges unchanged.

The optimal $\alpha$-threshold for a newly discovered MEC is calibrated via leave-one-out cross-validation on the MEC itself. For every split with a held-out edge $\text{E}_{ij}$, edges of the calibration set are used to compute the tagging statistics and, in turn, to predict the hold-out edge. A score function $s(\text{E}^{GT}, \text{E}^{pred})$ is introduced that returns an error of 0 for correct and 1 for incorrect predictions. To prevent the algorithm from always abstaining from making a decision, undirected edges are scored with an error of 0.5. Possible values for $\alpha$ are all $Q(\text{E}_{ij})$ encountered over all splits, $\boldsymbol{\alpha} := \bigcup_{i,j \in N \times N, i \neq j} Q(\text{E}_{ij})$. For every $\alpha' \in \boldsymbol{\alpha}$, the cumulative score $S(\alpha') := \sum s(\text{E}^{GT}_{ij}, \hat{d}_{ij}(Q(\text{E}_{ij}))$ is computed and the optimal $\alpha^*$ is selected as $\alpha^* := \arg\min_{\alpha' \in \boldsymbol{\alpha}} S(\alpha')$.

**Minimal Evidence and Meek Rules.** In addition to the considered $\alpha$ conservativeness, we enforce the evidence for the final $Q(\text{E}_{ij})$ to stem from either at least one or two different evidence edges. Our experiments found that a configuration with a two-edge boundary performed slightly more robustly. Further, the algorithm allows for the optional application of Meek rules after each edge-direction step. The four best configurations, however, do not require their application.

**Redirecting Edges.** The recovered CPDAG serves as the baseline of our approach. However, it might contain errors, stemming from too little data or measurement noise. While we assume the overall graph is trustworthy, we might also use our tagging statistics to detect edges that are likely to be incorrectly inferred. Therefore, we optionally allow our algorithm to redirect such wrongly inferred edges before moving on to

---

[1]Following the type consistency assumption, any unshielded triplet $A - C - B$ where $A$ and $B$ share a type that is different from $C$'s ($\text{t}^A = \text{t}^B \neq \text{t}^C$) must either be directed as a v-structure $A \to C \leftarrow B$ or as a fork $A \leftarrow C \to B$. Therefore, edges can even be directed when such a triplet does not represent a v-structure.

the undirected edges. (Edges that would induce cycles are still excluded.) Upon doing so, we allow the redirection to only take place at edges with a decision boundary of $Q(\mathrm{E}_{ij}) \geq 0.6$. As one might not trust the evidence stemming from the edge under consideration, one might not include it in the statistics. Again, this is a hyperparameter that is up to the user to decide. Our experiments showed a small preference for a conservative redirection strategy that includes the edge in the evidence. Lastly, we provide two variants of a *redirection strategy*, with either updating or not updating the evidence after redirecting edges.

## 4 Experiments

We conduct an extensive empirical evaluation over eleven datasets. We consider all possible combinations of the described parameters, resulting in 560 runs per dataset and seed (– 61,600 evaluations in total). We employ the PC (Spirtes & Glymour, 1991) and GES (Chickering, 2002) algorithms for initial CPDAG discovery, as those are the most common representatives of their respective classes of constraint-based and score-based CD algorithms. Our tagging approach, on average, achieves the best results on top of GES and also outperforms variants of the *naive* and *majority* typing approaches by Brouillard et al. (2022). We answer the following research questions: **Q1.** Is tagging information a suitable indicator for determining edge directions with competitive performance? **Q2.** Are tagging approaches robust under faulty information? **Q3.** Does the homogeneity assumption hold, and can it be used to identify higher-level causal relations?

**Tagging Annotation.** The key driver of our algorithm's performance is meaningful variable annotations with sets of tags. These sets could, for example, be distilled by leveraging meta-information about the domain, such as variable names or short variable descriptions (Long et al., 2023). In our experiments, we use LLMs as stand-in experts to provide a fair comparison across datasets, specifically GPT-5.2 (OpenAI, 2025), Claude Opus 4.6 (Anthropic, 2026), Gemini 3 Pro (Google, 2025), Llama 3.3 70b Instruct (Meta, 2024), Qwen3.5 397b-a17b Instruct (Qwen, 2026), Minimax M2.5 (MiniMax, 2026) and GLM-5 (Zeng et al., 2026). We make use of the fact that most modern LLMs are typically familiar with the given domains and therefore provide a good baseline across all datasets. Results are queried with the respective standard parameters per model and a temperature of 1.0. The LLMs are presented with a list of all variables per dataset and prompted to assign each variable to (possibly) multiple tags or a single type. The exact prompts are given in App. D. Answers from all LLMs are included in the code repository.

**Datasets & Evaluation.** We evaluate algorithm performance over several datasets of the bnlearn repository (Scutari, 2010; Taskesen, 2020). Additionally, we also include a Lung Cancer dataset (LUCAS; Guyon et al. (2011)). As we rely on language models to produce tags, we filter datasets that lack meaningful variable names. Eventually, we evaluate on the following datasets: Asia, Cancer, Earthquake, Survey, Alarm, Child, Insurance, Hailfinder, HEPAR2, Win95pts, and LUCAS. All experiments are carried out over 10 seeds with 10,000 randomly sampled datapoints per dataset. (We use the 2,000 provided ground-truth samples for the LUCAS dataset.) We include additional comparisons in the appendix (App. E.3), where we also compare to a different class of causal discovery approaches: orienting edges by prompting LLMs for the correct orientation directly. Compared to our tagging approach, such approaches require specific knowledge of causal directions rather than annotation of variables independent of their causal relation. Consequently, these families of methods rely on distinctly different assumptions.

We measure the individual performance of edge prediction using Structural Hamming Distance (SHD; Tsamardinos et al. (2006)), Structural Intervention Distances (SID; Peters & Bühlmann (2015)), as well as Precision, Recall, and $F_1$-score of all directed edges. SHD measures the correct or incorrect prediction of edges, with classical SHD increasing its score for every incorrect prediction. We also report an alternative ($\mathrm{SHD}_{\mathrm{double}}$), which counts a doubled error when encountering edges directed in the anti-causal direction. SID measures the similarity between two graphs based on their interventional distributions. Due to the exponential runtime scaling of SID, we could not evaluate it on the largest datasets: Hailfinder, HEPAR2, and Win95pts. (Evaluation did not conclude after 5 days.) We report upper and lower bounds for SID, which consider undirected edges to either be counted as correct or incorrect, respectively. Directing further edges can, therefore, never improve the lower SID bound but can reduce the upper bound.

| | SHD Ranks ($\downarrow$) | $\text{SHD}_{\text{double}}$ Ranks ($\downarrow$) | $\text{SID}_{\text{min}}$ Ranks ($\downarrow$) | $\text{SID}_{\text{max}}$ Ranks ($\downarrow$) | Precision Ranks ($\downarrow$) | Recall Ranks ($\downarrow$) | $\text{F}_1$ Ranks ($\downarrow$) |
|---|---|---|---|---|---|---|---|
| PC | $4.32_{\pm 0.38}$ | $4.27_{\pm 0.33}$ | $3.15_{\pm 0.28}$ | $3.98_{\pm 0.48}$ | $3.18_{\pm 0.39}$ | $4.41_{\pm 0.34}$ | $4.45_{\pm 0.31}$ |
| GES | $3.54_{\pm 0.51}$ | $3.07_{\pm 0.47}$ | $1.27_{\pm 0.13}$ | $3.17_{\pm 0.51}$ | $2.69_{\pm 0.51}$ | $3.17_{\pm 0.42}$ | $3.13_{\pm 0.51}$ |
| Typed-PC (Naive) | $3.42_{\pm 0.34}$ | $3.62_{\pm 0.33}$ | $3.10_{\pm 0.27}$ | $3.20_{\pm 0.33}$ | $3.07_{\pm 0.42}$ | $3.47_{\pm 0.34}$ | $3.59_{\pm 0.39}$ |
| Typed-PC (Maj.) | $2.97_{\pm 0.28}$ | $3.25_{\pm 0.34}$ | $3.40_{\pm 0.38}$ | $3.27_{\pm 0.37}$ | $2.84_{\pm 0.35}$ | $3.05_{\pm 0.26}$ | $3.20_{\pm 0.24}$ |
| Tagged-PC (AntiV) | $3.60_{\pm 0.45}$ | $4.05_{\pm 0.38}$ | $4.54_{\pm 0.32}$ | $4.05_{\pm 0.46}$ | $4.25_{\pm 0.40}$ | $3.67_{\pm 0.37}$ | $4.11_{\pm 0.40}$ |
| Tagged-PC | $2.66_{\pm 0.32}$ | $3.00_{\pm 0.41}$ | $2.95_{\pm 0.49}$ | $2.67_{\pm 0.50}$ | $3.07_{\pm 0.48}$ | $2.89_{\pm 0.28}$ | $2.99_{\pm 0.38}$ |
| Tagged-GES | $\mathbf{2.11}_{\pm 0.55}$ | $\mathbf{1.73}_{\pm 0.39}$ | $\mathbf{1.11}_{\pm 0.09}$ | $\mathbf{1.68}_{\pm 0.37}$ | $\mathbf{2.24}_{\pm 0.51}$ | $\mathbf{1.77}_{\pm 0.42}$ | $\mathbf{1.70}_{\pm 0.47}$ |

Table 1: **Average Ranks for Best Configurations over all Datasets (lower is better).** Tagging improves upon PC and GES, and both tagging methods outperform their typed counterparts. Tagged-GES achieves the best results overall, yielding the lowest ranks across all metrics. While the AntiV baseline offers a slight improvement over PC, it is surpassed by both typing and tagging approaches. Detailed absolute results are presented in Tab. 5 in the appendix. $_{\pm y}$ indicates standard deviation.

## 4.1 Tagging Performance

Results over all parameter configurations permitted by our method are evaluated on all datasets. For comparison with the PC, GES baselines, and typing approaches, we select the best-performing configuration for each method individually. We will use the best-performing parameters in the follow-up ablations. Because SHD and SID metrics scale with the number of variables and the sparsity of the graphs, we use the average $\text{F}_1$-score across all datasets to identify the best-performing configuration. We find the overall best strategy is obtained by not employing the AntiV approach, removing single tags, without the specificity prior, without applying Meek rules after every step, redirecting some existing edges, and using the tagging sets produced by Claude 4.6. More information on the best performing parameter configurations can be found in App. E.2

We compare our method with PC and GES baselines (Tagged-PC and Tagged-GES) and with standard PC and GES, as well as with previous typing approaches using both naive and majority strategies. Brouillard et al. (2022) presented another *t-Propagation* approach which, however, requires strict type consistency. As strict type consistency rarely holds for any non-constructed types, we omit this last step and additionally compare to this approach in Tab. 5 in the appendix.) Contrary to the experimental section in Brouillard et al. (2022), which employed synthetic types, matching the ground-truth causal ordering, we again use LLMs to provide types for all variables. Because SHD and SID scale non-linearly with dataset size, we rank absolute results per dataset and average the resulting ranks across all seeds. Results are presented in Tab. 1. A detailed report on all absolute metric scores across all datasets is provided in Tab. 5 in the appendix.

**Results.** We find that GES approaches generally perform better across most metrics. *Both tagging approaches, ignoring the AntiV variant, consistently improve upon their PC and GES baselines.* When comparing typing approaches, the 'majority' variant performs somewhat better than the 'naive' variant. Among the PC-based approaches, Tagged-PC improves upon both PC and the typing approaches in most or all metrics. Inspecting the datasets individually (Tab. 5), we observe a larger benefit of Tagged-PC on datasets with a medium or larger number of variables. This aligns with our theoretical results of Sec. 3.1 on the benefits of considering tag sets on sufficiently large graphs.

Comparing overall performance, *we find that average Tagged-GES results consistently rank the best across all methods and metrics*, even including $\text{SID}_{\text{min}}$ and Precision, which are both metrics that disproportionately favor keeping edges undirected. While areas of standard deviation sometimes overlap, Tagged-GES performs clearly better on $\text{SID}_{\text{max}}$, which is commonly considered an important measure for the ability of causal discovery methods to truthfully recover causal graphs. The results indicate our approach's ability to demonstrably reduce the Markov equivalence class of the initial CPDAGs and, therefore, reliably improve the given causal graph towards the underlying ground truth. Finally, we find the AntiV variant, adopted from the typing approaches, ranks worst among all typing and tagging methods, indicating that the heuristic rather harms than helps with variable tagging. Overall, this answers **Q1** affirmatively.

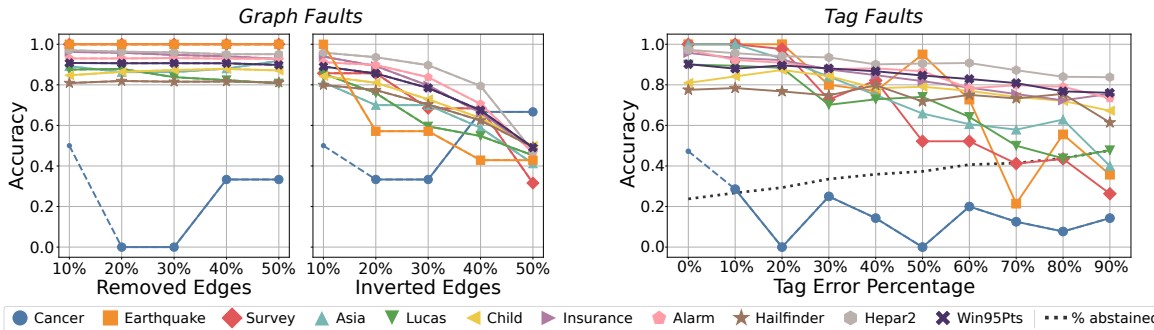

Figure 3: **Robustness to Faults.** Tags generated by Claude 4.6. Smaller point and dashed lines indicate 0 edge directions (defaulted to 50%). **Graph Faults (left):** Removing edges has little effect on performance, whereas inverting edges has a stronger negative impact on accuracy. **Tag Faults (right):** Accuracy decreases with increasing noise, but remains substantially better than random chance. The dotted line shows an increasing abstention rate. Discussion on outliers (Cancer dataset) and other LLMs in Apps. E.6 and E.7.

**Influence of LLM.** We now take a closer look at the impact of tagging annotations produced by the individual LLMs on the resulting performance. For this evaluation, we select the best configurations of hyperparameters measured by $F_1$ score *per LLM*. This allows every LLM to choose its optimal parameter setting, which might not align with the Claude 4.6 configuration, which was determined to be the globally best configuration. The selected parameter configurations per LLM and their $F_1$ scores are shown in Tab. 3. We observe no big qualitative differences between models. The resulting $F_1$ scores lie closely together, demonstrating that our algorithm is robust to changes in tagging sets. (The impact of other parameters is also relatively small; see further analysis of this in App. E.1.)

## 4.2 Robustness Analysis

In Sec. 3.2, we proposed multiple strategies for dealing with noisy settings. These are put in place to make the algorithm direct edges when evidence is strong and abstain when evidence is weak. To validate this behavior we stress-test how well tagging performs under settings of increasingly noisy/faulty graphs. We present the results for a select few experiments here and include full experiments in App. E.

Our tagging heuristic can possibly serve to complement any CD algorithm that produces partially directed graphs. While our experiments indicate that it performs well on error-free graphs (Apps. E.4 and E.5), we now examine the impact of different fault types. To this end, we gradually introduce faults into the ground truth graph by removing or flipping several edges. Subsequently, we undirect one of the remaining edges, disabling Meek rules to focus fully on the tag statistics. We either evaluate all possible problem instances or sample 20,000 random instances for larger graphs. Results for the removal and flipping of edges are shown in Fig. 3 (left). We find that tagging remains stable under the mild induction of faults on larger datasets if these do not insert incorrect edge directions. Performance is observed to suffer most from flipped edges, which introduce erroneous information and even affect larger datasets, whereas edge removal keeps performance stable, except for the very small Cancer dataset. Note that at 50% inverted edges, half the evidence is "correct" and half "incorrect". The resulting drop in performance to around 50% is, thus, unsurprising.

As LLM-generated causal knowledge can be unreliable, we also include two experiments that investigate performance under noisy tag conditions. To this end, we add $X\%$ noise by removing $\frac{X\%}{2}$ existing tags, and adding $\frac{X\%}{2}$ tags to variables that were not assigned this tag by the LLM. We then undirect one edge and measure the accuracy of our edge direction based on tags, following the setup from the graph faults experiment. In Figure 3 (right), we see that noisy tags slightly reduce the accuracy, but performance still remains relatively high. This can be explained by our aggregation of tag relations, which ensures that "bad" tags have effectively random accuracy, so that actually informative relations can tip the otherwise uncertain decision in the right direction. We also repeated our main evaluation from Sec. 4.1 with noisy tags of 50%, following the same noise setup as in the ablation experiment. Even in this realistic setting, performance only

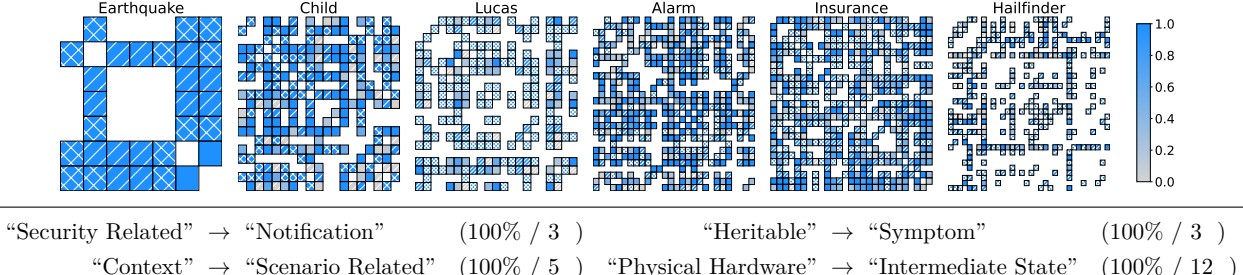

| "Security Related" → "Notification" | (100% / 3 ) | "Heritable" → "Symptom" | (100% / 3 ) |
| "Context" → "Scenario Related" | (100% / 5 ) | "Physical Hardware" → "Intermediate State" | (100% / 12 ) |

Figure 4: **Top: Tag Homogeneity for Claude 4.6.** A tag relation has high homogeneity (1) if variables of one tag consistently cause the other and low homogeneity (0) if the cause-and-effect relationship between these variables is random. We observe high homogeneity throughout all datasets. Hatched areas might appear white on screen due to anti-aliasing. See App. E.8 for high-resolution plots. **Bottom: Tag Relations.** Considering the most prominent tag pairs per dataset reveals high-level mechanisms present in the domain. We list tag pairs, with accuracy and support in brackets. Full list in App. E.8.

drops slightly compared to the noise-free tags (Tab. 9). We thus conclude that tagging further improves graph quality and achieves robust performance, even under the introduction of faults in the initial graph and when noisy tags are used, thereby answering **Q2** positively.

### 4.3 Mining Abstract Causal Relations

In the previous sections, we considered how the informative value of tags can help direct edges within graphs. Now, we examine the structure of this knowledge and focus on the abstract causal relations. We consider the ground-truth graphs as a data source for systematically discovering relations between abstract causal concepts, as realized by the annotated tags. Not all tags represent abstract concepts that are suited to infer causal relations. In Sec. 3, we discussed the homogeneity assumption (Assm. 1) that relates the homogeneity of tagging pairs to their predictive performance. In our previous experiments, we implicitly showed that this assumption holds for our selected datasets, as without it, edge recovery would not have been as successful. We take a closer look at the actual tagging homogeneity within the ground-truth graphs and invert the direction of inference by mining abstract causal relations from these observations.

Tagging homogeneity of Claude 4.6 tags across different datasets is shown in Fig. 4 (top). We find that tagging pairs, when they exist, generally yield high homogeneity with reasonable support across all considered datasets. As we initially instructed the LLMs to produce high-level tags, finding tag pairs with high homogeneity can, therefore, be thought of as extracting relations between high-level concepts. In Fig 4 (bottom), we list some of the most exemplary tag pairs with the highest homogeneity and support among their respective datasets (further examples are presented in App. E.8). Among the identified relations, top-ranking examples align with commonly held knowledge. Overall, we find that tagging provides robust statistics that offer useful insights into higher-level relations within the data. This answers **Q3** affirmatively.

## 5 Conclusion

While both data and semantic knowledge are valuable sources of information for identifying causal relations, hybrid approaches that integrate both sources of information appear to be a valuable avenue for further refining causal graphs. Our tagging approach advances this direction by utilizing semantic tags as lightweight meta-information, thereby allowing existing data to be leveraged more effectively. We showed that the employed probabilistic approach for further refining edge directions is particularly robust under tagging noise and graph faults, achieving noticeable improvements across all experiments.

One observed limitation of our approach might be the possible trade-off between directing as many edges as possible and abstaining from a decision, particularly in smaller graphs with insufficient grounding. Extending our method to account for confounders or unknown latent variables is an important future direction.

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

## Appendix "Tags for DAGs: Graph Refinement with Meta-Informed Relations"

This appendix is structured as follows. First, we go into a theoretical analysis of our tagging approach in App. A. Then, we describe a possible extension of the approach with a specificity prior in App. B. Next, we present the pseudo-code for our approach in App. C. App. D provides detailed information on the LLM prompts used to assign types and tags in this paper. Finally, we present extensive experimental details and results in App. E.

## A    Theoretical Analysis

Let us say that two variables $X_i$ and $X_j$ are only tagged with a single tag each, i.e., $(X_i, X_j) \in \mathbf{E}^{\mathcal{Y}, \mathrm{t}^a, \mathrm{t}^b}$ with $\mathrm{tag}(X_i) = \mathbf{t}_i = \{\mathrm{t}^a\}$ and $\mathrm{tag}(X_j) = \mathbf{t}_j = \{\mathrm{t}^b\}$. Consider the case that $X_i \to X_j$ is true, but unknown. The tag informative value $I^{\mathcal{Y}}(\mathrm{t}^a, \mathrm{t}^b)$ describes the probability provided by the tag pair $(\mathrm{t}^a, \mathrm{t}^b)$ according to which $\mathrm{t}^a$ is a cause of $\mathrm{t}^b$. Thus, correctly predicting the direction of an undirected edge follows a simple Bernoulli distribution and is correct with probability $d_{ij}'^{\mathcal{Y}} = I^{\mathcal{Y}}(\mathrm{t}^a, \mathrm{t}^b)$ (see Eq. 3). For ease of notation, we assume that all considered tag pairs have evidence ($o^{\mathcal{Y}}(t^a, t^b) = 1, \ \forall t^a, t^b$) and that the edge-specific noise $N_{ij}$ does not influence the prediction.

For assigning tags, we assume a noisy oracle that assigns tags that result in a particular accuracy of $\hat{A} := \mathbb{E}(I^{\mathcal{Y}}(\mathrm{t}^a, \mathrm{t}^b))$. Generally, we assume that the tag informative values follow a Beta distribution, the conjugate prior of the Bernoulli distribution:

$$I^{\mathcal{Y}}(\mathrm{t}^a, \mathrm{t}^b) \sim \mathrm{Beta}(\alpha, \beta) \text{ with } \hat{A} = \frac{\alpha}{\alpha + \beta} \tag{8}$$

for some $\alpha, \beta \in \mathbb{R}$. Here, $\alpha$ and $\beta$ can be seen as some underlying evidence for directing the edge $X_i - X_j$ in either direction. Following Assumption 1, according to which the tag informative values of different datasets converge towards each other (i.e., tag relations observed in some data help with correctly predicting the edges of new data), we can assume that, on average, $\alpha > \beta$ if $X_i \to X_j$ and vice versa.

Under this setting, we are now interested in the probability of correctly predicting the edge direction using our tagging approach $\mathbb{P}[d_{ij}'^{\mathcal{Y}} > 0.5]$. In the case $n = |\mathbf{t}_i \times \mathbf{t}_j| = 1$, we directly use the only tag informative value for our prediction:

$$\mathbb{P}[\hat{p}_1(X_i \to X_j) > 0.5] = \mathbb{P}[I^{\mathcal{Y}}(\mathrm{t}^a, \mathrm{t}^b) > 0.5] = 1 - \mathcal{I}_{0.5}(\alpha, \beta), \tag{9}$$

where $\mathcal{I}_{0.5}(\alpha, \beta) = \frac{\int_0^{0.5} u^{\alpha-1}(1-u)^{\beta-1} \, du}{\int_0^1 u^{\alpha-1}(1-u)^{\beta-1} \, du}$ is the regularized incomplete beta function[2], i.e., the cumulative distribution function (CDF) of the beta distribution. Intuitively, for a single tag pair, the probability of correctly predicting the edge $X_i \to X_j$ when $\alpha > \beta$ is the same as the probability of sampling a probability (tag informative value) higher than 0.5 from the underlying beta distribution.

The previous case is similar to the typing scenario, since we only considered a single tag pair influencing the decision. Let us now consider the more general scenario with multiple tags: $\mathbf{t}_i = \{\mathrm{t}_1^a, \mathrm{t}_2^a, \ldots, \mathrm{t}_k^a\}$ and $\mathbf{t}_j = \{\mathrm{t}_1^b, \mathrm{t}_2^b, \ldots, \mathrm{t}_l^b\}$, resulting in $n = |\mathbf{t}_i \times \mathbf{t}_j|$ tag pairs and tag informative values $\mathbf{I}^{\mathcal{Y}} = \{I_1^{\mathcal{Y}}, I_2^{\mathcal{Y}}, \ldots, I_n^{\mathcal{Y}}\}$. To allow for a simple formal analysis, we assume these $n$ tag informative values to be independent and identically distributed (i.i.d.). Our tagging approach makes the following prediction:

$$d_{ij}'^{\mathcal{Y}} = \frac{1}{n} \sum_{m=1}^{n} I_m^{\mathcal{Y}}. \tag{10}$$

We can see that the mean is the same as the mean of the underlying beta distribution

$$\mathbb{E}[d_{ij}'^{\mathcal{Y}}] = \mathbb{E}[\frac{1}{n} \sum_{m=1}^{n} \mathrm{Beta}(\alpha, \beta)] = \frac{1}{n} \sum_{m=1}^{n} \mathbb{E}[\mathrm{Beta}(\alpha, \beta)] = \frac{1}{n} \sum_{m=1}^{n} \frac{\alpha}{\alpha + \beta} = \frac{\alpha}{\alpha + \beta}, \tag{11}$$

---

[2]We use $\mathcal{I}$ instead of the commonly used $I$ to separate it from the tag informative value $I$.

which then tells us that we are making the correct predictions in the mean as $\mathbb{P}[\frac{\alpha}{\alpha+\beta} > 0.5]$ is true following our assumption that $\alpha > \beta$. More generally, we know that the probability of correctly predicting an edge using tagging is

$$\mathbb{P}[d_{ij}'^{\mathcal{Y}} > 0.5] = \mathbb{P}[\frac{1}{n} \sum_{m=1}^{n} I_m^{\mathcal{Y}} > 0.5] = 1 - F_{0.5}^n(\alpha, \beta), \tag{12}$$

where $F_{0.5}^n(\alpha, \beta)$ is the CDF of the distribution of the average of beta-distributed random variables ($I^{\mathcal{Y}}$) to which there exists no closed-form expression. However, given the variance of the beta distribution $\text{Var}[I^{\mathcal{Y}}] = \frac{\alpha\beta}{(\alpha+\beta)^2(\alpha+\beta+1)}$, we can make use of the Central Limit Theorem (CLT), according to which our prediction is approximately normally distributed with variance

$$\text{Var}[d_{ij}'^{\mathcal{Y}}] = \text{Var}[\frac{1}{n} \sum_{m=1}^{n} I_m^{\mathcal{Y}}] = \frac{\alpha\beta}{(\alpha+\beta)^2(\alpha+\beta+1)n}. \tag{13}$$

Since $\text{Var}[d_{ij}'^{\mathcal{Y}}]$ becomes smaller for larger $n$, we know that the CDF $F^n(\alpha, \beta)$ will continually amass more density around the mean (as it is normally distributed following the CLT). Therefore, $F_{0.5}^n(\alpha, \beta)$ will get smaller, the more $n$ increases, since the mean is not in the interval $[0, 0.5]$. In other words, the more tag pairs we utilize for our prediction, the more likely it is that our prediction is correct, exceeding the probability of only using a single tag pair or type. In the theoretical setting of $n \to \infty$, it even follows that $\text{Var}[d_{ij}'^{\mathcal{Y}}]_{n\to\infty} = 0$ and, therefore, $\mathbb{P}[d_{ij}'^{\mathcal{Y}} > 0.5] = 1$.

The opposite case of $X_i \leftarrow X_j$ follows by analogy.

## B   Specificity Prior

Upon considering the directionality of edges, one might want to weigh in tagging pairs $(\text{t}^a, \text{t}^b)$ more strongly, which feature a higher *specificity*. In general, specific tags might be used to indicate variables with deviating behavior, e.g., variables featuring different causal dynamics in contrast to the overall system. Such specific tags might indicate local environments where global assumptions might not hold, and upon encountering such tags, we want to prioritize them more strongly in our decision. We base the prior distribution on the Geometric series, counting the number of nodes assigned to any of the tags:

$$\hat{p}_{ab}^{\text{spec}} = \frac{1}{2^{|\{X_i \in \mathbf{X} \,|\, \text{t}^a \in \text{tagged}(X_i)\}| + |\{X_i \in \mathbf{X} \,|\, \text{t}^b \in \text{tagged}(X_i)\}| - 1}}. \tag{14}$$

Generally, the prior might or might not be used based on whether the specificity assumption is assumed to hold for the specific application. Thus, it is a hyperparameter for our algorithm. In case the prior is used, it is multiplied to the respective probability; otherwise, $\hat{p}_{ab}^{\text{spec}}$ is set to one:

$$\hat{p}_{ab}' = p_{ab}^{\text{spec}} \cdot \hat{p}_{ab}. \tag{15}$$

Note that Eq. 6 now requires a normalization term, since $\hat{p}_{ab}' + \hat{p}_{ba}'$ might no longer add up to one. We repeat the corresponding equation with normalization in the following:

$$Q(\text{E}_{ij}) = \frac{\sum_{\text{t}^a \in \mathbf{t}_i, \text{t}^b \in \mathbf{t}_j} \hat{p}_{ab}' \cdot o(t^a, t^b)}{\sum_{\text{t}^a \in \mathbf{t}_i, \text{t}^b \in \mathbf{t}_j} \hat{p}_{ab}' + \hat{p}_{ba}'}. \tag{16}$$

## C   Algorithm Pseudo Code

We introduce some methods required in our approach in Alg. 1 and then provide the pseudo-code for the actual tagging informed edge direction in Alg. 2.

---

**Algorithm 1** Tagging Helper Procedures

---

1: **procedure** COLLECT EVIDENCE($\mathcal{G}$, $\mathbf{t}$)
2:     $\mathbf{E} \leftarrow$ zero matrix of size $|\mathbf{t}| \times |\mathbf{t}|$
3:     **for** each directed edge $(u, v) \in \mathcal{G}$ **do**
4:         **for** each tag $t_u \in \mathbf{t}[u]$ **do**
5:             **for** each tag $t_v \in \mathbf{t}[v]$ **do**
6:                 $\mathbf{E}[t_u, t_v] \leftarrow \mathbf{E}[t_u, t_v] + 1$
7:             **end for**
8:         **end for**
9:     **end for**
10:     **return E**
11: **end procedure**
12: **procedure** FIND MOST PROMISING EDGE(Edges, $\mathbf{t}$, $\mathbf{E}$, OnlyBackward)
13:     BestEdge $\leftarrow$ None
14:     BestProb $\leftarrow 0$
15:     **for** each edge $(u, v) \in$ Edges **do**
16:         ForwardScore, BackwardScore, ForwardCount, BackwardCount $\leftarrow 0$
17:         **for** each tag $t_u \in \mathbf{t}[u]$ **do**
18:             **for** each tag $t_v \in \mathbf{t}[v]$ **do**
19:                 **if** $t_u \neq t_v$ **then**              ▷ Skip relations between identical tags
20:                     Forward $\leftarrow \mathbf{E}[t_u, t_v]$
21:                     **if** not(IncludeCurrentEdgeAsEvidence) **then**    ▷ Parameter for redirecting edges
22:                         Forward $\leftarrow$ Forward $- 1$          ▷ Exclude the current edge
23:                     **end if**
24:                     Backward $\leftarrow \mathbf{E}[t_v, t_u]$
25:                     **if** SpecificityPrior **then**         ▷ Parameter for applying the specificity prior
26:                         $p^{\mathrm{spec}} \leftarrow$ CALCSPECIFICITYPRIOR($t_u, t_v$)        ▷ Eq. 14
27:                     **else**
28:                         $p^{\mathrm{spec}} \leftarrow 1$
29:                     **end if**
30:                     ForwardScore $\leftarrow$ Forward $\cdot\, p^{\mathrm{spec}}$
31:                     BackwardScore $\leftarrow$ Backward $\cdot\, p^{\mathrm{spec}}$
32:                     ForwardCount $\leftarrow$ ForwardCount $+$ ForwardScore
33:                     BackwardCount $\leftarrow$ BackwardCount $+$ BackwardScore
34:                 **end if**
35:             **end for**
36:         **end for**
37:         **if** ForwardCount $+$ BackwardCount $<$ MinSamples **then**   ▷ Minimum tag pairs, we use either 1 or 2
38:             ForwardProb $\leftarrow 0.5$
39:         **else**
40:             ForwardProb $\leftarrow \frac{\text{ForwardScore}}{\text{ForwardScore} + \text{BackwardScore}}$
41:         **end if**
42:         BackwardProb $\leftarrow 1 -$ ForwardProb
43:         **if** not(BackwardOnly) **and** ForwardProb $>$ BestProb **then**     ▷ Ignore forward direction for redirections
44:             BestEdge $\leftarrow (u, v)$
45:             BestProb $\leftarrow$ ForwardProb
46:         **end if**
47:         **if** BackwardProb $>$ BestProb **then**
48:             BestEdge $\leftarrow (v, u)$
49:             BestProb $\leftarrow$ BackwardProb
50:         **end if**
51:     **end for**
52:     **return** BestEdge, BestProb
53: **end procedure**

---

**Algorithm 2** Tagging Informed Edge Direction

---

1: **procedure** TAGGING INFORMED EDGE DIRECTIONING(CPDAG $\mathcal{G}$, Tags **t**)
2:     $\mathbf{E} \leftarrow$ COLLECT EVIDENCE($\mathcal{G}$, **t**)
3:     **if** RedirectEdges **then**                 ▷ Parameter: consider redirectioning of edges?
4:         Loop $\leftarrow$ True
5:         DirEdges $\leftarrow$ directed_edges($\mathcal{G}$)
6:         RedirEdges $\leftarrow$ []
7:         IterationThreshold $\leftarrow$ 0
8:         **while** Loop **and** IterationThreshold $<$ 100 **do** ▷ Stop after 100 iterations to avoid infinite loops
9:             Loop $\leftarrow$ False
10:            $(u, v), p \leftarrow$ FIND MOST PROMISING EDGE(DirEdges, **t**, **E**, True)
11:            **if** $p > 0.6$ **then**
12:               $\mathcal{G}^* \leftarrow \mathcal{G}$.direct($(u, v)$)                   ▷ Redirect edge
13:               **if** is_acyclic($\mathcal{G}^*$) **then**
14:                   $\mathcal{G} \leftarrow$ copy($\mathcal{G}^*$)                 ▷ Accept redirection
15:                   DirEdges $\leftarrow$ directed_edges($\mathcal{G}$)
16:                   **if** RedirectStrategy == 0 **then**       ▷ Update evidence after each step
17:                     $\mathbf{E} \leftarrow$ COLLECT EVIDENCE($\mathcal{G}$, **t**)
18:                   **else**                   ▷ RedirectStrategy == 1
19:                     RedirEdges $\leftarrow$ RedirEdges.append($(u, v)$)
20:                     DirEdges $\leftarrow$ DirEdges.remove_all(RedirEdges) ▷ Remove already redirected edges
21:                   **end if**
22:               **else**                   ▷ Do not redirect
23:                 DirEdges $\leftarrow$ DirEdges.remove($(u, v)$)     ▷ Do not consider same edge again next
24:               **end if**
25:               **if** RedirectStrategy == 0 **then**
26:                 IterationThreshold $\leftarrow$ IterationThreshold + 1
27:               **end if**
28:               Loop $\leftarrow$ True
29:             **end if**
30:         **end while**
31:         $\mathbf{E} \leftarrow$ COLLECT EVIDENCE($\mathcal{G}$, **t**)
32:     **end if**
33:     **while** len(undirected_edges($\mathcal{G}$)) $> 0$ **do**           ▷ As long as there are undirected edges
34:         $(u, v), p \leftarrow$ FIND MOST PROMISING EDGE(undirected_edges($\mathcal{G}$), **t**, **E**, False)
35:         **if** $p > 0.5$ **then**
36:             $\mathcal{G}^* \leftarrow \mathcal{G}$.direct($(u, v)$)
37:             **if** $\mathcal{G}^*$.is_acyclic() **then**          ▷ If the direction would introduce cycles...
38:                $\mathcal{G} \leftarrow \mathcal{G}$.direct($(v, u)$)       ▷ ...instead add the edge in the opposite direction
39:             **else**
40:                $\mathcal{G} \leftarrow \mathcal{G}^*$            ▷ Otherwise, accept the directed edge
41:             **end if**
42:             **if** AlwaysMeek **then**      ▷ Parameter, to apply Meek Rules after every edge direction step
43:                $\mathcal{G} \leftarrow$ MEEK($\mathcal{G}$)            ▷ Meek Rules
44:             **end if**
45:         **else**          ▷ If $p = 0.5$ (can not be smaller), keep remaining edges undirected
46:             **break**
47:         **end if**
48:     **end while**
49:     $\mathcal{G} \leftarrow$ MEEK($\mathcal{G}$)                     ▷ Always apply one final Meek Rules
50:     **return** $\mathcal{G}$
51: **end procedure**

---

## D    LLM Prompts

In the following, we describe the details of LLM querying. All answers are generated using the same system prompt and either the following tagging or typing instructions. Model temperature was set to 1.0. All LLM answers are included within the code repository. Within every prompt, the `{variables}` fields were replaced by a comma-separated list of the individual dataset's variable names.

You are an expert in annotating variables to provide additional information that helps to support a causal discovery algorithm.

**System Prompt**

A tag is a single word or short phrase that describes a variable. Tags should be general enough to be applicable to multiple variables but specific enough to identify differences between similar variables. Tags will be used to identify causal directions between variables. Therefore, the individual sets of tags per variable should be discriminative enough to inform the algorithm. Variables can have multiple tags.\n
Consider the following variables: `{variables}`.\n\n

Please generate a list of tags that can be assigned to one or multiple variables. Generate the number of tags necessary to strike a good balance between expressivity and specificity. Avoid duplicate tags that contain the same set of variables. Reply with one line per tag, where each line starts with the name of the tag, followed by a colon, and then a comma-separated list of variables that have that tag. The output should be machine parsable. For that reason, do not include any explanations or additional comments.

**Tagging Instruction Prompt**

A type is a single word or short phrase that describes a variable. Types should be general enough to be applicable to multiple variables but specific enough to identify differences between similar variables. Types will be used to identify causal directions between variables. Therefore, the individual types should be discriminative enough to inform the algorithm. Variables are assigned to a single type only.\n
Consider the following variables: `{variables}`.\n\n

Please generate a list of types that can be assigned to one or multiple variables. Generate the number of types necessary to strike a good balance between expressivity and specificity. Reply with one line per type, where each line starts with the name of the type, followed by a colon, and then a comma-separated list of variables that belong to that type. Make sure that no variable appears in more than one the lists. The output should be machine parsable. For that reason, do not include any explanations or additional comments.

**Typing Instruction Prompt**

# E  Further Experimental Results

In this section, we provide additional details on the experiments. Here, we first describe two additional parameters of the actual algorithm's implementation (Sec. E.1) regarding the preprocessing of tagging sets. Second, we give more information on the best parameter configurations of our main evaluation and the difference between these with respect to different LLMs (Sec. E.2). Then, we show the results on all datasets, supplementing the purely ranking-based results of the main paper (Sec. E.3). We consider error-free settings by investigating how well our tagging-based method performs on the ground-truth CPDAG (Sec. E.4) and in an ablation study where we undirect some edges starting from the ground-truth graph (Sec. E.5). After that, we move onto experiments on the robustness of our approach, starting with settings where edges have been removed or inverted (Sec. E.6), followed by different experiments where we add noise to the tag assignment and investigate the effect on the tagging accuracy (Sec. E.7). Lastly, we complement our evaluations of the main paper investigation into homogeneity and tag relations by providing a larger amount of experimental results in Sec. E.8.

## E.1  Additional Algorithm Parameters

In addition to the algorithm parameters discussed in the main text, the code realization of the algorithm considers two additional parameters: a 'specificity prior' and 'tag set deduplication'. Both parameters were dedicated to balancing the influence of certain tagging sets, but turned out to not play a major role in the eventual performance of the algorithm:

**Specificity Prior.** We furthermore considered the use of a 'specificity prior' that was intended to reward in-group specificity by weighting tags occurring in specific sub-parts of the causal graphs more strongly. Experiments, however, found no empirical improvements from the introduction of this weighting term. A full description is given in App. B for completeness.

**Tag Set Deduplication.** As the same concept might be described by similar tags, we always deduplicate tags covering the same set of variables to avoid stronger weighting of certain concepts due to duplicate tags. To further differentiate from a typing approach, we optionally exclude singleton tagging sets. In our experiments, we measure the performance of both variants. We observed no major qualitative differences but found that the best-performing configuration included singular tagging sets.

## E.2  Best Parameter Configurations

There are a total of 560 possible parameter configurations per dataset and seed. In Tab. 2, we show the 10 best parameter configurations overall. As in the main paper, these are determined by their average $F_1$ score across all datasets and seeds to enable easy comparability between datasets (since SHD and SID values tend to be much larger for bigger datasets). It can be seen that the top ten scores are not far from each other, indicating a certain amount of robustness of our results with respect to the set of all possible parameters.

Tagging on top of the PC performs worse than the GES variant, and is achieved using tags from Gemini 3 Pro, requiring only 1 sample per tag pair, without removing single tags, using the specificity prior, without applying Meek rules after every step, and by redirecting edges with iterative updates and considering the edge under consideration of being redirected as part of the evidence. For the typing approaches, the best performing tags are generated by the following LLMs (consider App. E.3 for more information on t-propagation):

- Typed-PC (Naive): Claude 4.6,

- Typed-PC (Maj): Gemini 3 Pro,

- PC + t-Propagation: GLM 5,

- GES + t-Propagation: GLM 5.

In Tab. 3, we consider the best parameter configuration for each LLM. We observe that the choice of LLM used for tagging does not appear to play a crucial role in obtaining good results.

Tab. 4 shows the difference in $F_1$ score when changing one parameter compared to the optimal configuration (Tagged-GES). Although the differences between LLMs are also relatively small, the choice of the LLM (i.e., the set of tags) is clearly the most dominant hyperparameter. The second most impactful parameter reduces the average $F_1$ score by less than 0.01, and the remaining ones make an even smaller difference.

| LLM | Min-Samples | Fewer Tags | S. Prior | Always Meek | Redirect | Strategy | Include Edge | $F_1$ |
|---|---|---|---|---|---|---|---|---|
| Claude 4.6 | 2 | False | False | False | True | 0 | True | 0.7954 |
| Claude 4.6 | 2 | True | False | False | True | 0 | True | 0.7953 |
| Claude 4.6 | 2 | False | False | False | True | 1 | True | 0.7948 |
| Claude 4.6 | 2 | True | False | False | True | 1 | True | 0.7947 |
| Claude 4.6 | 2 | True | False | True | True | 0 | True | 0.7946 |
| Claude 4.6 | 2 | False | False | True | True | 0 | True | 0.7945 |
| Claude 4.6 | 2 | True | False | True | True | 1 | True | 0.7940 |
| Claude 4.6 | 2 | False | False | True | True | 1 | True | 0.7939 |
| Claude 4.6 | 1 | False | False | False | True | 0 | True | 0.7935 |
| Claude 4.6 | 1 | True | False | False | True | 0 | True | 0.7934 |

Table 2: **10 Best Configurations by $F_1$ score.** The parameters (columns) are: which LLM is used for determining tags (LLM), the number of tag combinations that must have been observed in order to allow for directing an edge (Min-Samples), whether to include all tags or remove those that are only assigned to single variables (Fewer Tags), whether to apply the specificity prior (S. Prior, see Eq. 14), whether to apply Meek rules after every direction or only at the end (Always Meek), whether to redirect edges at all (Redirect), which strategy to use for updating the evidence during edge redirection (Strategy; with 0 indicating iterative updates and 1 no updates), and whether to include the edge that is under consideration for being redirected as part of the evidence to decide on redirection (Include Edge). The score in the final column is the average $F_1$ score across all seeds and datasets. All results are achieved on the CPDAG from GES (results on the CPDAG output by PC performed worse).

| LLM | Min-Samples | Fewer Tags | S. Prior | Always Meek | Redirect | Strategy | Include Edge | $F_1$ |
|---|---|---|---|---|---|---|---|---|
| Claude 4.6 | 2 | False | False | False | True | 0 | True | 0.7954 |
| Gemini 3 Pro | 2 | True | False | True | True | 0 | True | 0.7893 |
| Qwen 3.5 | 1 | False | True | True | True | 1 | True | 0.7780 |
| Llama 3.3 | 1 | False | False | True | False | - | - | 0.7754 |
| GPT 5.2 | 1 | False | True | True | True | 1 | True | 0.7710 |
| Minimax 2.5 | 2 | False | False | False | True | 1 | True | 0.7678 |
| GLM 5 | 2 | False | False | False | True | 1 | True | 0.7656 |

Table 3: **Best Configurations by $F_1$ score per LLM.** The parameters (columns) are: which LLM is used for determining tags (LLM), the number of tag combinations that must have been observed in order to allow for directing an edge (Min-Samples), whether to include all tags or remove those that are only assigned to single variables (Fewer Tags), whether to apply the specificity prior (S. Prior, see Eq. 14), whether to apply Meek rules after every direction or only at the end (Always Meek), whether to redirect edges at all (Redirect), which strategy to use for updating the evidence during edge redirection (Strategy; with 0 indicating iterative updates and 1 no updates), and whether to include the edge that is under consideration for being redirected as part of the evidence to decide on redirection (Include Edge). The score in the final column is the average $F_1$ score across all seeds and datasets. All results are achieved on the CPDAG from GES (results on the CPDAG output by PC performed worse). The performance of all LLMs is relatively close to each other, with some deviations in the optimal parameter for their respective configuration.

| Parameter | Value | $\Delta F_1$ |
|---|---|---|
| LLM | Qwen 3.5 | -0.039881 |
| LLM | GPT 5.2 | -0.036041 |
| LLM | Llama 3.3 | -0.035907 |
| LLM | GLM 5 | -0.032763 |
| LLM | Minimax 2.5 | -0.027631 |
| LLM | Gemini 3 Pro | -0.018575 |
| Redirect | False | -0.007511 |
| S. Prior | True | -0.005633 |
| Include Edge | False | -0.005052 |
| Min-Samples | 1 | -0.001902 |
| Always Meek | True | -0.000924 |
| Strategy | 1 | -0.000574 |
| Fewer Tags | True | -0.000093 |

Table 4: **One-At-A-Time Sensitivity of Tagged-GES.** Each row shows the change in $F_1$ after varying a single hyperparameter while keeping all others fixed at the best setting. The difference in choosing the best tag set (i.e., LLM) is more important than any other hyperparameter.

### E.3   Evaluation Results on All Datasets

In the main body, we show the rankings of our evaluation in Tab. 1. We now include the detailed results in Tab. 5, where we show the unranked evaluation results per dataset. Here, we also include some steps of *t-Propagation* as proposed in Brouillard et al. (2022). However, we were unable to apply the last step of *t-Propagation*, where all type-consistent graphs are enumerated and aggregated into a final graph, since type consistency is not always satisfied using our datasets and types. The results for t-Propagation in the table are thus obtained by skipping this step (only propagating types and applying Meek rules), which turned out to perform rather badly on top of PC, while being more competitive than our method on GES. However, note that in smaller datasets where only a few edges need to be directed, using tags instead of types might not be necessary, as the information given by types could be sufficient. Thus, on the 4 smallest datasets, GES with t-Propagation sometimes achieves better scores than our tagging approach. For the remaining datasets, the tagging approach matches the t-Propagation in 2 cases and exceeds it in the remaining 5 cases, indicating an increased benefit of tagging over typing on larger, more complex datasets. This also aligns with our theoretical analysis in App. A.

**LLM-based Prediction.** As a complementary approach to data-based causal discovery, we consider the common pure LLM-based approach of Jiralerspong et al. (2024). LLM-based methods that query for the individual pairwise existence of edges (Kıcıman et al., 2023; Zečević et al., 2023) between all possible pairs of variables require a quadratic number of queries. These approaches quickly become computationally infeasible for larger graphs and lead to unreliable discovery, as no information about previous predictions is leveraged per query, and the acyclicity of the recovered graph can not be guaranteed. To this, the work of Jiralerspong et al. (2024) employs a breadth-first search, which starts by identifying root nodes (nodes without parents) from the complete set of nodes, and then traverses individual nodes, directly querying for all children of a particular node, excluding already visited nodes. The algorithm is linear in the number of steps required, and the recovered graph is acyclic by construction. The approach is denoted as '*LLM Root Recursive*' in all results tables.

Alternatively, we employ LLMs on top of causal discovery methods. Given a discovered MEC, we query LLMs to indicate the direction of all undirected edges. Results on top of GES and PC are reported as '*LLM on PC*' and '*LLM on GES*', respectively. While this approach often meets or exceeds the tag-based prediction accuracy, there are two aspects to consider when comparing them. For one, the prior knowledge is vastly different; tagging merely requires that some semantic similarity can be identified between variables, while direct LLM orienting only works if some knowledge about the causal relations themselves is already encoded in the LLM. Secondly, these datasets are well known throughout the CD literature and used frequently when

benchmarking CD methods. Thus, it is highly likely that they have been included as part of the training datasets for the tested LLMs. As data contamination through LLM memorization can not be ruled out, LLM performance might vary for novel domains.

### E.4    Evaluation on Ground-Truth CPDAG

In addition to our main evaluation, we considered the ground-truth CPDAG and applied tagging to discover whether our approach successfully directs undirected edges when the causal discovery algorithm is applied before identifying the CPDAG without error. Tab. 6 shows that tagging does, on average, improve the given CPDAGs by directing more undirected edges correctly than incorrectly. In addition to the ranked results, we include absolute scores on all datasets in Tab. 7.

### E.5    Directing of Undirected Edges

We conduct an additional experiment to investigate how our approach manages to direct undirected edges in settings without errors. To this end, we use the ground-truth graphs and undirect 10% to 50% of all edges. In the same manner as in Sec. 4.2, we either consider all possible combinations or 20,000 random edge combinations per dataset and number of edges to undirect. We also disable any application of Meek rules and acyclicity constraints, measuring only the correctness of directing the undirected edges according to the tagging statistics. We report results in Tab. 8, containing the number of correct and incorrect predictions, as well as the number of edges that our approach kept as undirected edges. Every edge direction is counted as one correct / incorrect / undecided count (therefore, the sum of these can exceed 20k). Overall, tagging predicts edges correctly much more often than not, although there are many cases where the edge remains undirected because of high uncertainty.

| Evaluation Results | SHD | $\text{SHD}_{\text{double}}$ | $\text{SID}_{\text{min}}$ | $\text{SID}_{\text{max}}$ | Precision | Recall | $F_1$ |
|---|---|---|---|---|---|---|---|
| Dataset Cancer | | | | | | | |
| PC | $2.00_{\pm 0.45}$ | $3.90_{\pm 0.70}$ | $9.60_{\pm 1.56}$ | $9.60_{\pm 1.56}$ | $0.51_{\pm 0.09}$ | $0.50_{\pm 0.11}$ | $0.50_{\pm 0.10}$ |
| GES | $0.80_{\pm 1.60}$ | $0.80_{\pm 1.60}$ | $\mathbf{0.20}_{\pm 0.40}$ | $2.20_{\pm 4.40}$ | $0.80_{\pm 0.40}$ | $0.80_{\pm 0.40}$ | $0.80_{\pm 0.40}$ |
| Typed-PC (Naive) | $2.00_{\pm 0.45}$ | $3.90_{\pm 0.70}$ | $9.60_{\pm 1.56}$ | $9.60_{\pm 1.56}$ | $0.51_{\pm 0.09}$ | $0.50_{\pm 0.11}$ | $0.50_{\pm 0.10}$ |
| Typed-PC (Maj.) | $2.00_{\pm 0.45}$ | $3.90_{\pm 0.70}$ | $9.60_{\pm 1.56}$ | $9.60_{\pm 1.56}$ | $0.51_{\pm 0.09}$ | $0.50_{\pm 0.11}$ | $0.50_{\pm 0.10}$ |
| PC + t-Propagation | $2.00_{\pm 0.00}$ | $3.80_{\pm 0.60}$ | $9.10_{\pm 2.70}$ | $9.60_{\pm 1.20}$ | $0.55_{\pm 0.15}$ | $0.50_{\pm 0.00}$ | $0.52_{\pm 0.05}$ |
| GES + t-Propagation | $0.40_{\pm 0.80}$ | $0.40_{\pm 0.80}$ | $\mathbf{0.20}_{\pm 0.40}$ | $1.20_{\pm 2.40}$ | $\mathbf{1.00}_{\pm 0.00}$ | $0.90_{\pm 0.20}$ | $0.93_{\pm 0.13}$ |
| Tagged-PC (AntiV) | $2.00_{\pm 0.45}$ | $3.90_{\pm 0.70}$ | $9.60_{\pm 1.56}$ | $9.60_{\pm 1.56}$ | $0.51_{\pm 0.09}$ | $0.50_{\pm 0.11}$ | $0.50_{\pm 0.10}$ |
| Tagged-PC | $2.00_{\pm 0.45}$ | $3.90_{\pm 0.70}$ | $9.60_{\pm 1.56}$ | $9.60_{\pm 1.56}$ | $0.51_{\pm 0.09}$ | $0.50_{\pm 0.11}$ | $0.50_{\pm 0.10}$ |
| Tagged-GES | $0.80_{\pm 1.60}$ | $0.80_{\pm 1.60}$ | $\mathbf{0.20}_{\pm 0.40}$ | $2.20_{\pm 4.40}$ | $0.80_{\pm 0.40}$ | $0.80_{\pm 0.40}$ | $0.80_{\pm 0.40}$ |
| LLM Root Recursive | $3.00_{\pm 0.00}$ | $3.00_{\pm 0.00}$ | $1.00_{\pm 0.00}$ | $1.00_{\pm 0.00}$ | $0.60_{\pm 0.00}$ | $0.75_{\pm 0.00}$ | $0.67_{\pm 0.00}$ |
| LLM on PC | $2.00_{\pm 0.45}$ | $3.90_{\pm 0.70}$ | $9.60_{\pm 1.56}$ | $9.60_{\pm 1.56}$ | $0.51_{\pm 0.09}$ | $0.50_{\pm 0.11}$ | $0.50_{\pm 0.10}$ |
| LLM on GES | $\mathbf{0.20}_{\pm 0.40}$ | $\mathbf{0.20}_{\pm 0.40}$ | $\mathbf{0.20}_{\pm 0.40}$ | $\mathbf{0.20}_{\pm 0.40}$ | $\mathbf{1.00}_{\pm 0.00}$ | $\mathbf{0.95}_{\pm 0.10}$ | $\mathbf{0.97}_{\pm 0.06}$ |
| Dataset Earthquake | | | | | | | |
| PC | $0.60_{\pm 1.50}$ | $0.80_{\pm 1.83}$ | $\mathbf{0.00}_{\pm 0.00}$ | $1.40_{\pm 4.20}$ | $0.90_{\pm 0.30}$ | $0.90_{\pm 0.30}$ | $0.90_{\pm 0.30}$ |
| GES | $\mathbf{0.00}_{\pm 0.00}$ | $\mathbf{0.00}_{\pm 0.00}$ | $\mathbf{0.00}_{\pm 0.00}$ | $\mathbf{0.00}_{\pm 0.00}$ | $\mathbf{1.00}_{\pm 0.00}$ | $\mathbf{1.00}_{\pm 0.00}$ | $\mathbf{1.00}_{\pm 0.00}$ |
| Typed-PC (Naive) | $0.40_{\pm 0.92}$ | $0.60_{\pm 1.28}$ | $\mathbf{0.00}_{\pm 0.00}$ | $0.90_{\pm 2.70}$ | $\mathbf{1.00}_{\pm 0.00}$ | $0.95_{\pm 0.15}$ | $0.97_{\pm 0.10}$ |
| Typed-PC (Maj.) | $0.40_{\pm 0.92}$ | $0.60_{\pm 1.28}$ | $\mathbf{0.00}_{\pm 0.00}$ | $0.90_{\pm 2.70}$ | $\mathbf{1.00}_{\pm 0.00}$ | $0.95_{\pm 0.15}$ | $0.97_{\pm 0.10}$ |
| PC + t-Propagation | $2.40_{\pm 0.92}$ | $4.40_{\pm 1.20}$ | $9.00_{\pm 3.00}$ | $10.80_{\pm 2.75}$ | $0.50_{\pm 0.22}$ | $0.45_{\pm 0.15}$ | $0.47_{\pm 0.16}$ |
| GES + t-Propagation | $\mathbf{0.00}_{\pm 0.00}$ | $\mathbf{0.00}_{\pm 0.00}$ | $\mathbf{0.00}_{\pm 0.00}$ | $\mathbf{0.00}_{\pm 0.00}$ | $\mathbf{1.00}_{\pm 0.00}$ | $\mathbf{1.00}_{\pm 0.00}$ | $\mathbf{1.00}_{\pm 0.00}$ |
| Tagged-PC (AntiV) | $0.40_{\pm 0.92}$ | $0.60_{\pm 1.28}$ | $\mathbf{0.00}_{\pm 0.00}$ | $0.90_{\pm 2.70}$ | $\mathbf{1.00}_{\pm 0.00}$ | $0.95_{\pm 0.15}$ | $0.97_{\pm 0.10}$ |
| Tagged-PC | $0.60_{\pm 1.50}$ | $0.80_{\pm 1.83}$ | $\mathbf{0.00}_{\pm 0.00}$ | $1.40_{\pm 4.20}$ | $0.90_{\pm 0.30}$ | $0.90_{\pm 0.30}$ | $0.90_{\pm 0.30}$ |
| Tagged-GES | $\mathbf{0.00}_{\pm 0.00}$ | $\mathbf{0.00}_{\pm 0.00}$ | $\mathbf{0.00}_{\pm 0.00}$ | $\mathbf{0.00}_{\pm 0.00}$ | $\mathbf{1.00}_{\pm 0.00}$ | $\mathbf{1.00}_{\pm 0.00}$ | $\mathbf{1.00}_{\pm 0.00}$ |
| LLM Root Recursive | $4.00_{\pm 0.00}$ | $4.00_{\pm 0.00}$ | $10.00_{\pm 0.00}$ | $10.00_{\pm 0.00}$ | $0.00_{\pm 0.00}$ | $0.00_{\pm 0.00}$ | $0.00_{\pm 0.00}$ |
| LLM on PC | $0.20_{\pm 0.40}$ | $0.20_{\pm 0.40}$ | $\mathbf{0.00}_{\pm 0.00}$ | $\mathbf{0.00}_{\pm 0.00}$ | $0.96_{\pm 0.08}$ | $\mathbf{1.00}_{\pm 0.00}$ | $0.98_{\pm 0.04}$ |
| LLM on GES | $\mathbf{0.00}_{\pm 0.00}$ | $\mathbf{0.00}_{\pm 0.00}$ | $\mathbf{0.00}_{\pm 0.00}$ | $\mathbf{0.00}_{\pm 0.00}$ | $\mathbf{1.00}_{\pm 0.00}$ | $\mathbf{1.00}_{\pm 0.00}$ | $\mathbf{1.00}_{\pm 0.00}$ |
| Dataset Survey | | | | | | | |
| PC | $2.30_{\pm 0.64}$ | $4.30_{\pm 0.64}$ | $11.10_{\pm 1.92}$ | $11.80_{\pm 2.44}$ | $0.64_{\pm 0.05}$ | $0.62_{\pm 0.11}$ | $0.63_{\pm 0.08}$ |
| GES | $4.20_{\pm 1.89}$ | $4.30_{\pm 1.79}$ | $8.50_{\pm 2.06}$ | $15.80_{\pm 7.52}$ | $0.48_{\pm 0.48}$ | $0.30_{\pm 0.31}$ | $0.36_{\pm 0.37}$ |
| Typed-PC (Naive) | $2.40_{\pm 0.66}$ | $4.40_{\pm 0.66}$ | $11.50_{\pm 1.86}$ | $12.30_{\pm 2.33}$ | $0.64_{\pm 0.05}$ | $0.60_{\pm 0.11}$ | $0.62_{\pm 0.09}$ |
| Typed-PC (Maj.) | $2.20_{\pm 0.60}$ | $4.30_{\pm 0.90}$ | $13.30_{\pm 0.90}$ | $13.30_{\pm 0.90}$ | $0.64_{\pm 0.08}$ | $0.63_{\pm 0.10}$ | $0.64_{\pm 0.09}$ |
| PC + t-Propagation | $5.40_{\pm 0.49}$ | $10.70_{\pm 0.90}$ | $27.10_{\pm 1.92}$ | $27.10_{\pm 1.92}$ | $0.10_{\pm 0.08}$ | $0.10_{\pm 0.08}$ | $0.10_{\pm 0.08}$ |
| GES + t-Propagation | $4.20_{\pm 1.89}$ | $5.50_{\pm 3.26}$ | $12.70_{\pm 7.62}$ | $15.80_{\pm 7.52}$ | $0.48_{\pm 0.48}$ | $0.30_{\pm 0.31}$ | $0.36_{\pm 0.37}$ |
| Tagged-PC (AntiV) | $2.60_{\pm 0.49}$ | $4.60_{\pm 0.49}$ | $13.10_{\pm 0.30}$ | $13.60_{\pm 0.49}$ | $0.63_{\pm 0.03}$ | $0.57_{\pm 0.08}$ | $0.59_{\pm 0.06}$ |
| Tagged-PC | $2.20_{\pm 0.40}$ | $4.30_{\pm 0.64}$ | $11.20_{\pm 2.04}$ | $11.20_{\pm 2.04}$ | $0.64_{\pm 0.05}$ | $0.63_{\pm 0.07}$ | $0.64_{\pm 0.06}$ |
| Tagged-GES | $4.20_{\pm 1.89}$ | $4.30_{\pm 1.79}$ | $8.50_{\pm 2.06}$ | $15.80_{\pm 7.52}$ | $0.48_{\pm 0.48}$ | $0.30_{\pm 0.31}$ | $0.36_{\pm 0.37}$ |
| LLM Root Recursive | $6.00_{\pm 0.00}$ | $6.00_{\pm 0.00}$ | $17.00_{\pm 0.00}$ | $17.00_{\pm 0.00}$ | $0.00_{\pm 0.00}$ | $0.00_{\pm 0.00}$ | $0.00_{\pm 0.00}$ |
| LLM on PC | $2.10_{\pm 0.30}$ | $4.10_{\pm 0.30}$ | $11.10_{\pm 1.92}$ | $11.10_{\pm 1.92}$ | $0.66_{\pm 0.02}$ | $0.65_{\pm 0.05}$ | $0.65_{\pm 0.04}$ |
| LLM on GES | $\mathbf{2.00}_{\pm 0.45}$ | $\mathbf{2.10}_{\pm 0.54}$ | $\mathbf{7.30}_{\pm 2.00}$ | $\mathbf{7.30}_{\pm 2.00}$ | $\mathbf{0.98}_{\pm 0.06}$ | $\mathbf{0.67}_{\pm 0.07}$ | $\mathbf{0.79}_{\pm 0.06}$ |
| Dataset Asia | | | | | | | |
| PC | $6.10_{\pm 0.30}$ | $6.40_{\pm 0.80}$ | $14.40_{\pm 3.90}$ | $28.00_{\pm 2.68}$ | $0.95_{\pm 0.15}$ | $0.25_{\pm 0.00}$ | $0.39_{\pm 0.02}$ |
| GES | $3.00_{\pm 0.00}$ | $3.00_{\pm 0.00}$ | $0.80_{\pm 0.40}$ | $8.80_{\pm 1.60}$ | $\mathbf{1.00}_{\pm 0.00}$ | $0.62_{\pm 0.00}$ | $0.77_{\pm 0.00}$ |
| Typed-PC (Naive) | $3.00_{\pm 1.26}$ | $3.60_{\pm 2.24}$ | $16.60_{\pm 6.97}$ | $17.70_{\pm 7.48}$ | $0.91_{\pm 0.20}$ | $0.64_{\pm 0.15}$ | $0.75_{\pm 0.17}$ |
| Typed-PC (Maj.) | $3.40_{\pm 1.20}$ | $4.40_{\pm 2.24}$ | $20.00_{\pm 8.39}$ | $21.10_{\pm 8.38}$ | $0.84_{\pm 0.21}$ | $0.59_{\pm 0.15}$ | $0.69_{\pm 0.17}$ |
| PC + t-Propagation | $3.40_{\pm 0.66}$ | $4.40_{\pm 0.66}$ | $21.10_{\pm 0.54}$ | $21.10_{\pm 0.54}$ | $0.81_{\pm 0.05}$ | $0.59_{\pm 0.06}$ | $0.68_{\pm 0.05}$ |
| GES + t-Propagation | $0.80_{\pm 0.40}$ | $0.80_{\pm 0.40}$ | $0.80_{\pm 0.40}$ | $0.80_{\pm 0.40}$ | $\mathbf{1.00}_{\pm 0.00}$ | $0.90_{\pm 0.05}$ | $\mathbf{0.95}_{\pm 0.03}$ |
| Tagged-PC (AntiV) | $2.60_{\pm 0.80}$ | $2.80_{\pm 1.25}$ | $14.40_{\pm 3.90}$ | $14.40_{\pm 3.90}$ | $0.95_{\pm 0.11}$ | $0.69_{\pm 0.08}$ | $0.80_{\pm 0.09}$ |
| Tagged-PC | $5.10_{\pm 0.30}$ | $5.40_{\pm 0.92}$ | $14.90_{\pm 5.39}$ | $24.20_{\pm 3.12}$ | $0.93_{\pm 0.16}$ | $0.38_{\pm 0.00}$ | $0.53_{\pm 0.04}$ |
| Tagged-GES | $3.00_{\pm 0.00}$ | $3.00_{\pm 0.00}$ | $0.80_{\pm 0.40}$ | $8.80_{\pm 1.60}$ | $\mathbf{1.00}_{\pm 0.00}$ | $0.62_{\pm 0.00}$ | $0.77_{\pm 0.00}$ |
| LLM Root Recursive | $4.00_{\pm 0.00}$ | $4.00_{\pm 0.00}$ | $\mathbf{0.00}_{\pm 0.00}$ | $\mathbf{0.00}_{\pm 0.00}$ | $0.67_{\pm 0.00}$ | $\mathbf{1.00}_{\pm 0.00}$ | $0.80_{\pm 0.00}$ |
| LLM on PC | $2.60_{\pm 0.80}$ | $2.80_{\pm 1.25}$ | $14.40_{\pm 3.90}$ | $14.40_{\pm 3.90}$ | $0.95_{\pm 0.11}$ | $0.69_{\pm 0.08}$ | $0.80_{\pm 0.09}$ |
| LLM on GES | $\mathbf{0.80}_{\pm 0.40}$ | $\mathbf{0.80}_{\pm 0.40}$ | $0.80_{\pm 0.40}$ | $0.80_{\pm 0.40}$ | $\mathbf{1.00}_{\pm 0.00}$ | $0.90_{\pm 0.05}$ | $\mathbf{0.95}_{\pm 0.03}$ |

| Evaluation Results | SHD | $\text{SHD}_{\text{double}}$ | $\text{SID}_{\text{min}}$ | $\text{SID}_{\text{max}}$ | Precision | Recall | $F_1$ |
|---|---|---|---|---|---|---|---|
| Dataset Lucas | | | | | | | |
| PC | $1.00_{\pm 0.00}$ | $1.00_{\pm 0.00}$ | $\mathbf{0.00}_{\pm 0.00}$ | $7.00_{\pm 0.00}$ | $\mathbf{1.00}_{\pm 0.00}$ | $0.92_{\pm 0.00}$ | $0.96_{\pm 0.00}$ |
| GES | $1.00_{\pm 0.00}$ | $1.00_{\pm 0.00}$ | $\mathbf{0.00}_{\pm 0.00}$ | $7.00_{\pm 0.00}$ | $\mathbf{1.00}_{\pm 0.00}$ | $0.92_{\pm 0.00}$ | $0.96_{\pm 0.00}$ |
| Typed-PC (Naive) | $1.00_{\pm 0.00}$ | $1.00_{\pm 0.00}$ | $\mathbf{0.00}_{\pm 0.00}$ | $7.00_{\pm 0.00}$ | $\mathbf{1.00}_{\pm 0.00}$ | $0.92_{\pm 0.00}$ | $0.96_{\pm 0.00}$ |
| Typed-PC (Maj.) | $1.00_{\pm 0.00}$ | $1.00_{\pm 0.00}$ | $\mathbf{0.00}_{\pm 0.00}$ | $7.00_{\pm 0.00}$ | $\mathbf{1.00}_{\pm 0.00}$ | $0.92_{\pm 0.00}$ | $0.96_{\pm 0.00}$ |
| PC + t-Propagation | $12.00_{\pm 0.00}$ | $12.00_{\pm 0.00}$ | $\mathbf{0.00}_{\pm 0.00}$ | $99.00_{\pm 0.00}$ | $0.00_{\pm 0.00}$ | $0.00_{\pm 0.00}$ | $0.00_{\pm 0.00}$ |
| GES + t-Propagation | $1.00_{\pm 0.00}$ | $1.00_{\pm 0.00}$ | $\mathbf{0.00}_{\pm 0.00}$ | $7.00_{\pm 0.00}$ | $\mathbf{1.00}_{\pm 0.00}$ | $0.92_{\pm 0.00}$ | $0.96_{\pm 0.00}$ |
| Tagged-PC (AntiV) | $2.00_{\pm 0.00}$ | $4.00_{\pm 0.00}$ | $16.00_{\pm 0.00}$ | $16.00_{\pm 0.00}$ | $0.83_{\pm 0.00}$ | $0.83_{\pm 0.00}$ | $0.83_{\pm 0.00}$ |
| Tagged-PC | $\mathbf{0.00}_{\pm 0.00}$ | $\mathbf{0.00}_{\pm 0.00}$ | $\mathbf{0.00}_{\pm 0.00}$ | $\mathbf{0.00}_{\pm 0.00}$ | $\mathbf{1.00}_{\pm 0.00}$ | $\mathbf{1.00}_{\pm 0.00}$ | $\mathbf{1.00}_{\pm 0.00}$ |
| Tagged-GES | $\mathbf{0.00}_{\pm 0.00}$ | $\mathbf{0.00}_{\pm 0.00}$ | $\mathbf{0.00}_{\pm 0.00}$ | $\mathbf{0.00}_{\pm 0.00}$ | $\mathbf{1.00}_{\pm 0.00}$ | $\mathbf{1.00}_{\pm 0.00}$ | $\mathbf{1.00}_{\pm 0.00}$ |
| LLM Root Recursive | $12.00_{\pm 0.00}$ | $12.00_{\pm 0.00}$ | $51.00_{\pm 0.00}$ | $51.00_{\pm 0.00}$ | $0.00_{\pm 0.00}$ | $0.00_{\pm 0.00}$ | $0.00_{\pm 0.00}$ |
| LLM on PC | $\mathbf{0.00}_{\pm 0.00}$ | $\mathbf{0.00}_{\pm 0.00}$ | $\mathbf{0.00}_{\pm 0.00}$ | $\mathbf{0.00}_{\pm 0.00}$ | $\mathbf{1.00}_{\pm 0.00}$ | $\mathbf{1.00}_{\pm 0.00}$ | $\mathbf{1.00}_{\pm 0.00}$ |
| LLM on GES | $\mathbf{0.00}_{\pm 0.00}$ | $\mathbf{0.00}_{\pm 0.00}$ | $\mathbf{0.00}_{\pm 0.00}$ | $\mathbf{0.00}_{\pm 0.00}$ | $\mathbf{1.00}_{\pm 0.00}$ | $\mathbf{1.00}_{\pm 0.00}$ | $\mathbf{1.00}_{\pm 0.00}$ |
| Dataset Child | | | | | | | |
| PC | $5.20_{\pm 1.40}$ | $8.60_{\pm 2.24}$ | $76.00_{\pm 22.26}$ | $100.80_{\pm 26.43}$ | $0.85_{\pm 0.04}$ | $0.80_{\pm 0.06}$ | $0.82_{\pm 0.05}$ |
| GES | $12.00_{\pm 0.00}$ | $12.00_{\pm 0.00}$ | $\mathbf{0.00}_{\pm 0.00}$ | $228.00_{\pm 0.00}$ | $\mathbf{1.00}_{\pm 0.00}$ | $0.52_{\pm 0.00}$ | $0.68_{\pm 0.00}$ |
| Typed-PC (Naive) | $4.30_{\pm 2.05}$ | $7.50_{\pm 3.17}$ | $69.60_{\pm 26.76}$ | $78.20_{\pm 34.51}$ | $0.86_{\pm 0.05}$ | $0.83_{\pm 0.09}$ | $0.85_{\pm 0.07}$ |
| Typed-PC (Maj.) | $2.60_{\pm 1.20}$ | $4.70_{\pm 2.28}$ | $38.10_{\pm 23.41}$ | $44.30_{\pm 20.53}$ | $0.91_{\pm 0.05}$ | $0.90_{\pm 0.05}$ | $0.91_{\pm 0.05}$ |
| PC + t-Propagation | $11.10_{\pm 2.07}$ | $20.60_{\pm 3.95}$ | $156.00_{\pm 25.65}$ | $184.40_{\pm 29.29}$ | $0.59_{\pm 0.08}$ | $0.56_{\pm 0.08}$ | $0.58_{\pm 0.08}$ |
| GES + t-Propagation | $3.00_{\pm 0.00}$ | $3.00_{\pm 0.00}$ | $\mathbf{0.00}_{\pm 0.00}$ | $44.00_{\pm 0.00}$ | $\mathbf{1.00}_{\pm 0.00}$ | $0.88_{\pm 0.00}$ | $0.94_{\pm 0.00}$ |
| Tagged-PC (AntiV) | $3.30_{\pm 0.46}$ | $6.50_{\pm 0.81}$ | $55.50_{\pm 2.20}$ | $55.50_{\pm 2.20}$ | $0.87_{\pm 0.02}$ | $0.87_{\pm 0.02}$ | $0.87_{\pm 0.02}$ |
| Tagged-PC | $1.30_{\pm 1.19}$ | $2.50_{\pm 2.42}$ | $21.00_{\pm 23.48}$ | $21.00_{\pm 23.48}$ | $0.95_{\pm 0.05}$ | $0.95_{\pm 0.05}$ | $0.95_{\pm 0.05}$ |
| Tagged-GES | $3.00_{\pm 0.00}$ | $3.00_{\pm 0.00}$ | $\mathbf{0.00}_{\pm 0.00}$ | $44.00_{\pm 0.00}$ | $\mathbf{1.00}_{\pm 0.00}$ | $0.88_{\pm 0.00}$ | $0.94_{\pm 0.00}$ |
| LLM Root Recursive | $8.00_{\pm 0.00}$ | $8.00_{\pm 0.00}$ | $54.00_{\pm 0.00}$ | $54.00_{\pm 0.00}$ | $0.84_{\pm 0.00}$ | $0.84_{\pm 0.00}$ | $0.84_{\pm 0.00}$ |
| LLM on PC | $4.00_{\pm 1.26}$ | $7.90_{\pm 2.62}$ | $76.80_{\pm 22.99}$ | $76.80_{\pm 22.99}$ | $0.84_{\pm 0.05}$ | $0.84_{\pm 0.05}$ | $0.84_{\pm 0.05}$ |
| LLM on GES | $\mathbf{1.00}_{\pm 0.00}$ | $\mathbf{2.00}_{\pm 0.00}$ | $4.00_{\pm 0.00}$ | $\mathbf{4.00}_{\pm 0.00}$ | $0.96_{\pm 0.00}$ | $\mathbf{0.96}_{\pm 0.00}$ | $\mathbf{0.96}_{\pm 0.00}$ |
| Dataset Alarm | | | | | | | |
| PC | $11.50_{\pm 3.96}$ | $13.40_{\pm 5.71}$ | $58.80_{\pm 39.62}$ | $125.90_{\pm 68.80}$ | $0.93_{\pm 0.07}$ | $0.79_{\pm 0.07}$ | $0.85_{\pm 0.07}$ |
| GES | $8.80_{\pm 2.18}$ | $9.90_{\pm 2.70}$ | $47.40_{\pm 9.87}$ | $96.20_{\pm 12.66}$ | $0.94_{\pm 0.03}$ | $0.84_{\pm 0.03}$ | $0.89_{\pm 0.03}$ |
| Typed-PC (Naive) | $9.10_{\pm 2.51}$ | $10.90_{\pm 3.83}$ | $53.40_{\pm 35.37}$ | $93.90_{\pm 36.60}$ | $0.94_{\pm 0.05}$ | $0.84_{\pm 0.04}$ | $0.89_{\pm 0.04}$ |
| Typed-PC (Maj.) | $8.40_{\pm 1.43}$ | $11.50_{\pm 1.69}$ | $90.10_{\pm 18.08}$ | $99.90_{\pm 15.97}$ | $0.91_{\pm 0.02}$ | $0.85_{\pm 0.02}$ | $0.88_{\pm 0.02}$ |
| PC + t-Propagation | $25.70_{\pm 2.83}$ | $45.30_{\pm 5.39}$ | $450.00_{\pm 70.04}$ | $482.50_{\pm 56.05}$ | $0.53_{\pm 0.08}$ | $0.48_{\pm 0.07}$ | $0.50_{\pm 0.07}$ |
| GES + t-Propagation | $7.80_{\pm 2.18}$ | $8.90_{\pm 2.70}$ | $47.40_{\pm 9.87}$ | $87.20_{\pm 12.66}$ | $0.94_{\pm 0.03}$ | $0.86_{\pm 0.03}$ | $0.90_{\pm 0.03}$ |
| Tagged-PC (AntiV) | $8.90_{\pm 1.22}$ | $13.40_{\pm 1.69}$ | $115.00_{\pm 26.50}$ | $115.00_{\pm 26.50}$ | $0.88_{\pm 0.03}$ | $0.84_{\pm 0.02}$ | $0.86_{\pm 0.02}$ |
| Tagged-PC | $7.10_{\pm 3.11}$ | $9.20_{\pm 4.85}$ | $72.40_{\pm 45.53}$ | $81.70_{\pm 50.81}$ | $0.93_{\pm 0.06}$ | $0.88_{\pm 0.05}$ | $0.91_{\pm 0.05}$ |
| Tagged-GES | $\mathbf{4.50}_{\pm 1.28}$ | $\mathbf{4.50}_{\pm 1.28}$ | $\mathbf{18.40}_{\pm 0.80}$ | $36.40_{\pm 0.80}$ | $\mathbf{0.97}_{\pm 0.02}$ | $\mathbf{0.93}_{\pm 0.01}$ | $\mathbf{0.95}_{\pm 0.01}$ |
| LLM Root Recursive | $87.00_{\pm 0.00}$ | $87.00_{\pm 0.00}$ | $24.00_{\pm 0.00}$ | $\mathbf{24.00}_{\pm 0.00}$ | $0.34_{\pm 0.00}$ | $0.91_{\pm 0.00}$ | $0.49_{\pm 0.00}$ |
| LLM on PC | $6.50_{\pm 2.50}$ | $8.20_{\pm 4.24}$ | $62.60_{\pm 40.67}$ | $62.60_{\pm 40.67}$ | $0.93_{\pm 0.06}$ | $0.90_{\pm 0.04}$ | $0.91_{\pm 0.05}$ |
| LLM on GES | $4.60_{\pm 1.80}$ | $5.70_{\pm 2.33}$ | $47.40_{\pm 9.87}$ | $47.40_{\pm 9.87}$ | $0.95_{\pm 0.03}$ | $0.93_{\pm 0.02}$ | $0.94_{\pm 0.03}$ |
| Dataset Insurance | | | | | | | |
| PC | $22.00_{\pm 2.61}$ | $30.20_{\pm 4.14}$ | $371.30_{\pm 37.41}$ | $393.30_{\pm 50.29}$ | $0.78_{\pm 0.04}$ | $0.59_{\pm 0.05}$ | $0.67_{\pm 0.05}$ |
| GES | $24.50_{\pm 2.73}$ | $28.10_{\pm 5.63}$ | $\mathbf{230.70}_{\pm 61.36}$ | $341.70_{\pm 23.83}$ | $0.85_{\pm 0.11}$ | $0.57_{\pm 0.03}$ | $0.68_{\pm 0.06}$ |
| Typed-PC (Naive) | $18.70_{\pm 2.05}$ | $26.20_{\pm 3.57}$ | $354.10_{\pm 33.73}$ | $354.10_{\pm 33.73}$ | $0.81_{\pm 0.05}$ | $0.65_{\pm 0.03}$ | $0.72_{\pm 0.04}$ |
| Typed-PC (Maj.) | $20.70_{\pm 3.49}$ | $26.80_{\pm 5.40}$ | $346.70_{\pm 58.53}$ | $374.20_{\pm 58.37}$ | $0.83_{\pm 0.06}$ | $0.61_{\pm 0.07}$ | $0.70_{\pm 0.06}$ |
| PC + t-Propagation | $26.10_{\pm 3.11}$ | $40.70_{\pm 6.42}$ | $451.50_{\pm 41.94}$ | $451.80_{\pm 41.61}$ | $0.64_{\pm 0.08}$ | $0.51_{\pm 0.06}$ | $0.57_{\pm 0.07}$ |
| GES + t-Propagation | $19.90_{\pm 4.66}$ | $23.50_{\pm 7.58}$ | $\mathbf{230.70}_{\pm 61.36}$ | $312.70_{\pm 26.03}$ | $0.86_{\pm 0.10}$ | $0.66_{\pm 0.07}$ | $0.74_{\pm 0.08}$ |
| Tagged-PC (AntiV) | $19.90_{\pm 3.56}$ | $28.60_{\pm 6.33}$ | $413.50_{\pm 66.30}$ | $413.50_{\pm 66.30}$ | $0.78_{\pm 0.07}$ | $0.63_{\pm 0.07}$ | $0.69_{\pm 0.07}$ |
| Tagged-PC | $20.90_{\pm 2.21}$ | $30.00_{\pm 4.00}$ | $389.60_{\pm 48.01}$ | $390.20_{\pm 48.46}$ | $0.77_{\pm 0.05}$ | $0.61_{\pm 0.04}$ | $0.68_{\pm 0.04}$ |
| Tagged-GES | $18.40_{\pm 4.15}$ | $22.40_{\pm 6.55}$ | $250.30_{\pm 37.97}$ | $259.30_{\pm 30.59}$ | $0.86_{\pm 0.08}$ | $0.69_{\pm 0.06}$ | $0.76_{\pm 0.07}$ |
| LLM Root Recursive | $53.00_{\pm 0.00}$ | $53.00_{\pm 0.00}$ | $238.00_{\pm 0.00}$ | $238.00_{\pm 0.00}$ | $0.49_{\pm 0.00}$ | $0.67_{\pm 0.00}$ | $0.57_{\pm 0.00}$ |
| LLM on PC | $19.30_{\pm 2.10}$ | $27.40_{\pm 3.69}$ | $371.30_{\pm 37.41}$ | $371.30_{\pm 37.41}$ | $0.79_{\pm 0.04}$ | $0.64_{\pm 0.04}$ | $0.71_{\pm 0.04}$ |
| LLM on GES | $\mathbf{17.10}_{\pm 5.63}$ | $\mathbf{20.70}_{\pm 8.56}$ | $\mathbf{230.70}_{\pm 61.36}$ | $\mathbf{230.70}_{\pm 61.36}$ | $\mathbf{0.87}_{\pm 0.10}$ | $\mathbf{0.71}_{\pm 0.09}$ | $\mathbf{0.78}_{\pm 0.09}$ |

| Evaluation Results | SHD | SHD$_{double}$ | SID$_{min}$ | SID$_{max}$ | Precision | Recall | F$_1$ |
|---|---|---|---|---|---|---|---|
| Dataset Hailfinder | | | | | | | |
| PC | 46.00$\pm$2.83 | 49.40$\pm$4.18 | - | - | 0.83$\pm$0.06 | 0.39$\pm$0.04 | 0.53$\pm$0.04 |
| GES | **39.00**$\pm$15.21 | **42.20**$\pm$15.28 | - | - | 0.69$\pm$0.12 | **0.65**$\pm$0.12 | **0.67**$\pm$0.12 |
| Typed-PC (Naive) | 45.40$\pm$2.50 | 48.90$\pm$3.48 | - | - | 0.83$\pm$0.06 | 0.40$\pm$0.02 | 0.54$\pm$0.03 |
| Typed-PC (Maj.) | 45.80$\pm$2.23 | 48.70$\pm$2.65 | - | - | **0.84**$\pm$0.05 | 0.40$\pm$0.00 | 0.54$\pm$0.01 |
| PC + t-Propagation | 62.80$\pm$3.03 | 80.80$\pm$5.40 | - | - | 0.36$\pm$0.06 | 0.14$\pm$0.02 | 0.20$\pm$0.03 |
| GES + t-Propagation | **39.00**$\pm$15.21 | **42.20**$\pm$15.28 | - | - | 0.69$\pm$0.12 | **0.65**$\pm$0.12 | **0.67**$\pm$0.12 |
| Tagged-PC (AntiV) | 49.30$\pm$2.69 | 59.20$\pm$3.03 | - | - | 0.63$\pm$0.06 | 0.34$\pm$0.02 | 0.44$\pm$0.03 |
| Tagged-PC | 41.90$\pm$3.45 | 44.70$\pm$3.93 | - | - | 0.80$\pm$0.04 | 0.45$\pm$0.04 | 0.58$\pm$0.04 |
| Tagged-GES | **39.00**$\pm$15.21 | **42.20**$\pm$15.28 | - | - | 0.69$\pm$0.12 | **0.65**$\pm$0.12 | **0.67**$\pm$0.12 |
| LLM Root Recursive | 180.00$\pm$0.00 | 180.00$\pm$0.00 | - | - | 0.13$\pm$0.00 | 0.32$\pm$0.00 | 0.19$\pm$0.00 |
| LLM on PC | 41.50$\pm$2.54 | 45.10$\pm$3.51 | - | - | 0.76$\pm$0.05 | 0.46$\pm$0.02 | 0.57$\pm$0.03 |
| LLM on GES | **39.00**$\pm$15.21 | **42.20**$\pm$15.28 | - | - | 0.69$\pm$0.12 | **0.65**$\pm$0.12 | **0.67**$\pm$0.12 |
| Dataset Hepar2 | | | | | | | |
| PC | 94.80$\pm$4.85 | 126.60$\pm$9.84 | - | - | 0.50$\pm$0.07 | 0.28$\pm$0.04 | 0.35$\pm$0.05 |
| GES | 57.60$\pm$3.23 | 64.90$\pm$3.36 | - | - | 0.90$\pm$0.01 | 0.53$\pm$0.03 | 0.67$\pm$0.02 |
| Typed-PC (Naive) | 95.40$\pm$3.20 | 127.70$\pm$8.44 | - | - | 0.49$\pm$0.04 | 0.27$\pm$0.02 | 0.35$\pm$0.03 |
| Typed-PC (Maj.) | 75.10$\pm$5.43 | 86.70$\pm$9.97 | - | - | 0.77$\pm$0.06 | 0.44$\pm$0.04 | 0.56$\pm$0.05 |
| PC + t-Propagation | 83.40$\pm$3.50 | 106.90$\pm$7.37 | - | - | 0.62$\pm$0.05 | 0.37$\pm$0.03 | 0.46$\pm$0.03 |
| GES + t-Propagation | 53.40$\pm$1.80 | 61.30$\pm$2.19 | - | - | 0.90$\pm$0.01 | 0.57$\pm$0.01 | 0.69$\pm$0.01 |
| Tagged-PC (AntiV) | 77.30$\pm$2.15 | 93.20$\pm$5.13 | - | - | 0.71$\pm$0.04 | 0.42$\pm$0.01 | 0.53$\pm$0.02 |
| Tagged-PC | 94.20$\pm$9.90 | 127.90$\pm$19.41 | - | - | 0.48$\pm$0.13 | 0.28$\pm$0.08 | 0.35$\pm$0.10 |
| Tagged-GES | **50.30**$\pm$1.55 | **55.00**$\pm$2.32 | - | - | **0.94**$\pm$0.02 | **0.59**$\pm$0.01 | **0.73**$\pm$0.01 |
| LLM Root Recursive | 127.00$\pm$0.00 | 127.00$\pm$0.00 | - | - | 0.42$\pm$0.00 | 0.08$\pm$0.00 | 0.14$\pm$0.00 |
| LLM on PC | 90.90$\pm$5.24 | 122.20$\pm$9.96 | - | - | 0.51$\pm$0.07 | 0.31$\pm$0.04 | 0.38$\pm$0.05 |
| LLM on GES | 53.70$\pm$1.79 | 62.50$\pm$2.16 | - | - | 0.89$\pm$0.01 | 0.57$\pm$0.01 | 0.69$\pm$0.01 |
| Dataset Win95Pts | | | | | | | |
| PC | 58.80$\pm$4.17 | 66.10$\pm$6.28 | - | - | **0.85**$\pm$0.05 | 0.52$\pm$0.03 | 0.65$\pm$0.04 |
| GES | 52.20$\pm$6.43 | 57.90$\pm$6.82 | - | - | 0.76$\pm$0.03 | 0.71$\pm$0.03 | 0.73$\pm$0.03 |
| Typed-PC (Naive) | 60.70$\pm$2.79 | 69.90$\pm$2.91 | - | - | 0.82$\pm$0.03 | 0.51$\pm$0.02 | 0.63$\pm$0.02 |
| Typed-PC (Maj.) | 55.40$\pm$4.34 | 64.70$\pm$8.31 | - | - | 0.83$\pm$0.06 | 0.55$\pm$0.04 | 0.66$\pm$0.04 |
| PC + t-Propagation | 83.00$\pm$3.55 | 122.70$\pm$5.35 | - | - | 0.44$\pm$0.03 | 0.31$\pm$0.03 | 0.36$\pm$0.03 |
| GES + t-Propagation | 48.40$\pm$6.28 | 55.30$\pm$6.90 | - | - | 0.76$\pm$0.03 | 0.74$\pm$0.03 | 0.75$\pm$0.03 |
| Tagged-PC (AntiV) | 51.60$\pm$3.23 | 59.70$\pm$5.44 | - | - | 0.84$\pm$0.04 | 0.59$\pm$0.03 | 0.69$\pm$0.03 |
| Tagged-PC | 48.70$\pm$3.61 | 57.00$\pm$6.36 | - | - | 0.84$\pm$0.04 | 0.61$\pm$0.02 | 0.71$\pm$0.03 |
| Tagged-GES | 45.90$\pm$5.96 | 50.00$\pm$6.29 | - | - | 0.79$\pm$0.03 | 0.76$\pm$0.03 | 0.77$\pm$0.03 |
| LLM Root Recursive | 149.00$\pm$0.00 | 149.00$\pm$0.00 | - | - | 0.24$\pm$0.00 | 0.15$\pm$0.00 | 0.19$\pm$0.00 |
| LLM on PC | 47.90$\pm$3.56 | 55.30$\pm$6.02 | - | - | 0.85$\pm$0.04 | 0.62$\pm$0.03 | 0.72$\pm$0.03 |
| LLM on GES | **44.00**$\pm$6.07 | **49.60**$\pm$6.41 | - | - | 0.78$\pm$0.03 | **0.78**$\pm$0.03 | **0.78**$\pm$0.03 |

Table 5: **Absolute Scores of Best Configuration per Dataset.** For each method, the parameter configuration (including LLM) that performed best for itself was chosen. Split across two pages due to the large table size. Certain SID results were omitted due to the high computation time. $\pm y$ indicates standard deviation.

|  | SHD Ranks | SHD$_{double}$ Ranks | SID$_{min}$ Ranks | SID$_{max}$ Ranks | Precision Ranks | Recall Ranks | F$_1$ Ranks |
|---|---|---|---|---|---|---|---|
| GT CPDAG | $1.64_{\pm 0.00}$ | $1.45_{\pm 0.00}$ | $\mathbf{1.00}_{\pm 0.00}$ | $1.38_{\pm 0.00}$ | $\mathbf{1.00}_{\pm 0.00}$ | $1.64_{\pm 0.00}$ | $1.55_{\pm 0.00}$ |
| Tagging on GT CPDAG | $\mathbf{1.00}_{\pm 0.00}$ | $\mathbf{1.18}_{\pm 0.00}$ | $1.25_{\pm 0.00}$ | $\mathbf{1.12}_{\pm 0.00}$ | $1.36_{\pm 0.00}$ | $\mathbf{1.00}_{\pm 0.00}$ | $\mathbf{1.09}_{\pm 0.00}$ |

Table 6: **Average Ranks for Ground Truth CPDAG over all Datasets (lower is better).** We use the parameters that performed best in the main evaluation as shown in Table 1. SID$_{min}$ and Precision are highest on the GT CPDAG, as there are no errors in the graph, so these metrics can not be improved upon. However, the other metrics show that our approach, on average, improves the graph by directing more edges correctly than incorrectly. $_{\pm y}$ indicates std. deviation.

| Evaluation Results | SHD | SHD$_{double}$ | SID$_{min}$ | SID$_{max}$ | Precision | Recall | F$_1$ |
|---|---|---|---|---|---|---|---|
| Dataset Cancer |  |  |  |  |  |  |  |
| GT CPDAG | $\mathbf{0.00}_{\pm 0.00}$ | $\mathbf{0.00}_{\pm 0.00}$ | $\mathbf{0.00}_{\pm 0.00}$ | $\mathbf{0.00}_{\pm 0.00}$ | $\mathbf{1.00}_{\pm 0.00}$ | $\mathbf{1.00}_{\pm 0.00}$ | $\mathbf{1.00}_{\pm 0.00}$ |
| Tagging on GT CPDAG | $\mathbf{0.00}_{\pm 0.00}$ | $\mathbf{0.00}_{\pm 0.00}$ | $\mathbf{0.00}_{\pm 0.00}$ | $\mathbf{0.00}_{\pm 0.00}$ | $\mathbf{1.00}_{\pm 0.00}$ | $\mathbf{1.00}_{\pm 0.00}$ | $\mathbf{1.00}_{\pm 0.00}$ |
| Dataset Earthquake |  |  |  |  |  |  |  |
| GT CPDAG | $\mathbf{0.00}_{\pm 0.00}$ | $\mathbf{0.00}_{\pm 0.00}$ | $\mathbf{0.00}_{\pm 0.00}$ | $\mathbf{0.00}_{\pm 0.00}$ | $\mathbf{1.00}_{\pm 0.00}$ | $\mathbf{1.00}_{\pm 0.00}$ | $\mathbf{1.00}_{\pm 0.00}$ |
| Tagging on GT CPDAG | $\mathbf{0.00}_{\pm 0.00}$ | $\mathbf{0.00}_{\pm 0.00}$ | $\mathbf{0.00}_{\pm 0.00}$ | $\mathbf{0.00}_{\pm 0.00}$ | $\mathbf{1.00}_{\pm 0.00}$ | $\mathbf{1.00}_{\pm 0.00}$ | $\mathbf{1.00}_{\pm 0.00}$ |
| Dataset Survey |  |  |  |  |  |  |  |
| GT CPDAG | $\mathbf{0.00}_{\pm 0.00}$ | $\mathbf{0.00}_{\pm 0.00}$ | $\mathbf{0.00}_{\pm 0.00}$ | $\mathbf{0.00}_{\pm 0.00}$ | $\mathbf{1.00}_{\pm 0.00}$ | $\mathbf{1.00}_{\pm 0.00}$ | $\mathbf{1.00}_{\pm 0.00}$ |
| Tagging on GT CPDAG | $\mathbf{0.00}_{\pm 0.00}$ | $\mathbf{0.00}_{\pm 0.00}$ | $\mathbf{0.00}_{\pm 0.00}$ | $\mathbf{0.00}_{\pm 0.00}$ | $\mathbf{1.00}_{\pm 0.00}$ | $\mathbf{1.00}_{\pm 0.00}$ | $\mathbf{1.00}_{\pm 0.00}$ |
| Dataset Asia |  |  |  |  |  |  |  |
| GT CPDAG | $\mathbf{3.00}_{\pm 0.00}$ | $\mathbf{3.00}_{\pm 0.00}$ | $\mathbf{0.00}_{\pm 0.00}$ | $\mathbf{12.00}_{\pm 0.00}$ | $\mathbf{1.00}_{\pm 0.00}$ | $\mathbf{0.62}_{\pm 0.00}$ | $\mathbf{0.77}_{\pm 0.00}$ |
| Tagging on GT CPDAG | $\mathbf{3.00}_{\pm 0.00}$ | $\mathbf{3.00}_{\pm 0.00}$ | $\mathbf{0.00}_{\pm 0.00}$ | $\mathbf{12.00}_{\pm 0.00}$ | $\mathbf{1.00}_{\pm 0.00}$ | $\mathbf{0.62}_{\pm 0.00}$ | $\mathbf{0.77}_{\pm 0.00}$ |
| Dataset Lucas |  |  |  |  |  |  |  |
| GT CPDAG | $1.00_{\pm 0.00}$ | $1.00_{\pm 0.00}$ | $\mathbf{0.00}_{\pm 0.00}$ | $7.00_{\pm 0.00}$ | $\mathbf{1.00}_{\pm 0.00}$ | $0.92_{\pm 0.00}$ | $0.96_{\pm 0.00}$ |
| Tagging on GT CPDAG | $\mathbf{0.00}_{\pm 0.00}$ | $\mathbf{0.00}_{\pm 0.00}$ | $\mathbf{0.00}_{\pm 0.00}$ | $\mathbf{0.00}_{\pm 0.00}$ | $\mathbf{1.00}_{\pm 0.00}$ | $\mathbf{1.00}_{\pm 0.00}$ | $\mathbf{1.00}_{\pm 0.00}$ |
| Dataset Child |  |  |  |  |  |  |  |
| GT CPDAG | $10.00_{\pm 0.00}$ | $10.00_{\pm 0.00}$ | $\mathbf{0.00}_{\pm 0.00}$ | $209.00_{\pm 0.00}$ | $\mathbf{1.00}_{\pm 0.00}$ | $0.60_{\pm 0.00}$ | $0.75_{\pm 0.00}$ |
| Tagging on GT CPDAG | $\mathbf{2.00}_{\pm 0.00}$ | $\mathbf{3.00}_{\pm 0.00}$ | $20.00_{\pm 0.00}$ | $\mathbf{43.00}_{\pm 0.00}$ | $0.96_{\pm 0.00}$ | $\mathbf{0.92}_{\pm 0.00}$ | $\mathbf{0.94}_{\pm 0.00}$ |
| Dataset Alarm |  |  |  |  |  |  |  |
| GT CPDAG | $4.00_{\pm 0.00}$ | $4.00_{\pm 0.00}$ | $\mathbf{0.00}_{\pm 0.00}$ | $46.00_{\pm 0.00}$ | $\mathbf{1.00}_{\pm 0.00}$ | $0.91_{\pm 0.00}$ | $0.95_{\pm 0.00}$ |
| Tagging on GT CPDAG | $\mathbf{2.00}_{\pm 0.00}$ | $\mathbf{3.00}_{\pm 0.00}$ | $42.00_{\pm 0.00}$ | $60.00_{\pm 0.00}$ | $0.98_{\pm 0.00}$ | $\mathbf{0.96}_{\pm 0.00}$ | $\mathbf{0.97}_{\pm 0.00}$ |
| Dataset Insurance |  |  |  |  |  |  |  |
| GT CPDAG | $2.00_{\pm 0.00}$ | $2.00_{\pm 0.00}$ | $\mathbf{0.00}_{\pm 0.00}$ | $75.00_{\pm 0.00}$ | $\mathbf{1.00}_{\pm 0.00}$ | $0.96_{\pm 0.00}$ | $0.98_{\pm 0.00}$ |
| Tagging on GT CPDAG | $\mathbf{1.00}_{\pm 0.00}$ | $\mathbf{1.00}_{\pm 0.00}$ | $\mathbf{0.00}_{\pm 0.00}$ | $\mathbf{49.00}_{\pm 0.00}$ | $\mathbf{1.00}_{\pm 0.00}$ | $\mathbf{0.98}_{\pm 0.00}$ | $\mathbf{0.99}_{\pm 0.00}$ |
| Dataset Hailfinder |  |  |  |  |  |  |  |
| GT CPDAG | $17.00_{\pm 0.00}$ | $\mathbf{17.00}_{\pm 0.00}$ | - | - | $\mathbf{1.00}_{\pm 0.00}$ | $0.74_{\pm 0.00}$ | $0.85_{\pm 0.00}$ |
| Tagging on GT CPDAG | $\mathbf{9.00}_{\pm 0.00}$ | $18.00_{\pm 0.00}$ | - | - | $0.86_{\pm 0.00}$ | $\mathbf{0.86}_{\pm 0.00}$ | $\mathbf{0.86}_{\pm 0.00}$ |
| Dataset Hepar2 |  |  |  |  |  |  |  |
| GT CPDAG | $7.00_{\pm 0.00}$ | $7.00_{\pm 0.00}$ | - | - | $\mathbf{1.00}_{\pm 0.00}$ | $0.94_{\pm 0.00}$ | $0.97_{\pm 0.00}$ |
| Tagging on GT CPDAG | $\mathbf{3.00}_{\pm 0.00}$ | $\mathbf{3.00}_{\pm 0.00}$ | - | - | $\mathbf{1.00}_{\pm 0.00}$ | $\mathbf{0.98}_{\pm 0.00}$ | $\mathbf{0.99}_{\pm 0.00}$ |
| Dataset Win95Pts |  |  |  |  |  |  |  |
| GT CPDAG | $8.00_{\pm 0.00}$ | $\mathbf{8.00}_{\pm 0.00}$ | - | - | $\mathbf{1.00}_{\pm 0.00}$ | $0.93_{\pm 0.00}$ | $\mathbf{0.96}_{\pm 0.00}$ |
| Tagging on GT CPDAG | $\mathbf{7.00}_{\pm 0.00}$ | $11.00_{\pm 0.00}$ | - | - | $0.96_{\pm 0.00}$ | $\mathbf{0.94}_{\pm 0.00}$ | $0.95_{\pm 0.00}$ |

Table 7: **Absolute Scores for Ground Truth CPDAG over all Datasets.** Certain SID results were omitted due to the high computation time. We use the parameters that performed best in the main evaluation as shown in Table 1. Direction using tagging improves results on several datasets. $_{\pm y}$ indicates std. deviation.

### E.6 Further Results on Graph Faults

We report the plots on removed and inverted edges in Fig. 5. All results were obtained in the same manner as in Sec. 4.2 of the main body. While performance varies slightly between the LLMs that generated the

tags, we observe no qualitative difference to our reported results on Claude 4.6. Note that some outlier behavior can be observed for Cancer and Earthquake. As these datasets are very small (4 edges each), our tagging statistics are highly susceptible to noise, and edges can be directed incorrectly due to just a small amount of bad evidence.

### E.7 Edge Directions under Noisy or Sparse Tags

In this section, we include several experiments to show our method's robustness in settings with noisy and sparse tags. The first experiment here uses a setup similar to the previous experiment on undirecting directed edges (Sec. E.5). We consider the ground truth causal graph and, considering every edge once, undirect this single edge and record whether it again gets directed correctly, incorrectly, or remains undirected. Edges that would be directed using Meek rules are excluded from this evaluation, as we only want to investigate the effects of tagging. We differ from the previous experiment by adding $N\%$ tag faults with $N = [0, 10, 20, 30, 40, 50, 60, 70, 80, 90]$ (with 100% representing the current number of tags). To be precise, we add $\frac{N}{2}\%$ random tags to variables that do not already have this tag and remove $\frac{X}{2}\%$ tags that were originally assigned by the LLM. Then, for any edge in the graph, we undirected it and record the correctness of the prediction according to tagging based on the noisy tag set and all the other directed edges. We average across 10 seeds and record the accuracy of correct predictions (ignoring edges that are not directed). The results are shown in Figure 6.

We can see that even if a large number of faults are introduced, the average accuracy remains relatively stable. This demonstrates that our approach is quite robust to suboptimal tagging, as incoherent tags get down-weighted by our approach. Note that even if 50% of the tags are faulty, tagging still results in better-than-random predictions on average. While accuracy further decreases with noise going up to 90%, average accuracy is still not clearly harmful (i.e., below 50%), and the larger datasets in particular tend to maintain a substantially higher accuracy than random guessing across LLMs. Note that some outlier behavior can be observed for Cancer and Earthquake. As these datasets are very small (4 edges each), our tagging statistics are highly susceptible to noise, and edges can be directed incorrectly due to just a small amount of bad evidence. The black, dotted line shows that the number of abstentions increases when evidence becomes less clear, highlighting the robustness of our approach in noisy settings (abstention rates are shown on a per-dataset basis in Fig. 7).

Additionally, we run our evaluation from Section 4.1 but now with 50% noise added in the same manner as for the aforementioned experiments. The comparison of the main evaluation results to the noisy results can be found in Tab. 9. We see that the non-noisy setup performs better on average, but the reduction in performance is relatively small, even for this highly noisy setup.

In the last experiment in this section, we repeat our main evaluation once more. However, now, instead of introducing several types of tag errors, we simply reduce the tag amount, rerunning the evaluation with 25%, 50%, and 75% fewer tags per dataset. Results are shown in Tab. 10. Unsurprisingly, using a smaller number of tags does not lead to better results. This provides further evidence for our hypothesis that a larger number of tags generally only improves tagging accuracy, as noisy tags can be ignored, while useful tags are leveraged to improve prediction accuracy.

### E.8 Further Results on Homogeneity

We report the plots on tag homogeneity in Fig. 8 in the same manner as in Sec. 4.3 for all LLMs. To highlight low-evidence cases in the homogeneity plots, elements supported by only a single edge are marked with an 'X' pattern, while those supported by exactly two edges are marked with diagonal lines ('/'). While tags differ, all LLMs show high levels of homogeneity. In Tab. 11, we include all tag relations with a positive homogeneity for Claude 4.6 that have at least three edges as evidence (sorting by accuracy and only showing the top 7 relations if there are more).

| | | Ca | Ea | Su | As | Lu | Ch | In | Al | Ha | He | Wi |
|---|---|---|---|---|---|---|---|---|---|---|---|---|
| GPT 5.2 | 10% | 0 / 0 / 0 | 0 / 0 / 0 | 5 / 0 / 1 | 7 / 0 / 1 | 7 / 0 / 5 | 272 / 0 / 328 | 60.9k / 4.1k / 35k | 33.7k / 5.9k / 60.5k | 10.5k / 0 / 129.5k | 62.7k / 0 / 177.3k | 37.5k / 5.7k / 176.9k |
| | 20% | 0 / 0 / 4 | 0 / 0 / 0 | 5 / 0 / 1 | 43 / 0 / 13 | 74 / 2 / 56 | 37.7k / 0 / 62.3k | 114k / 8.5k / 77.4k | 55.5k / 9.5k / 115k | 15.8k / 0 / 244.2k | 124.9k / 0 / 375.1k | 74.1k / 13.3k / 352.7k |
| | 30% | 0 / 0 / 4 | 0 / 0 / 4 | 16 / 1 / 13 | 43 / 0 / 13 | 992 / 72 / 916 | 48.8k / 0 / 111.2k | 167k / 13.5k / 139.5k | 74.2k / 12.5k / 193.3k | 18.2k / 0 / 381.8k | 172.5k / 0 / 567.5k | 109.3k / 20.6k / 550.1k |
| | 40% | 0 / 0 / 8 | 0 / 0 / 4 | 16 / 1 / 13 | 110 / 0 / 58 | 1.8k / 172 / 2k | 51.8k / 0 / 148.2k | 201k / 16.8k / 202.2k | 83k / 13.6k / 263.4k | 17.8k / 0 / 502.2k | 205.2k / 0 / 774.8k | 134.4k / 25.4k / 740.2k |
| | 50% | 4 / 0 / 8 | 0 / 0 / 4 | 18 / 5 / 37 | 150 / 0 / 130 | 2.3k / 270 / 3k | 52.4k / 0 / 187.6k | 224.1k / 18.4k / 277.4k | 87.2k / 14k / 358.8k | 15.3k / 0 / 644.7k | 216.5k / 0 / 1023.5k | 148k / 27.3k / 944.7k |
| Claude 4.6 | 10% | 0 / 0 / 0 | 0 / 0 / 0 | 5 / 0 / 1 | 5 / 0 / 3 | 9 / 1 / 2 | 409 / 79 / 112 | 86.1k / 3.3k / 10.7k | 78.6k / 5.3k / 16.1k | 80.6k / 19.3k / 40.1k | 207.7k / 6.9k / 25.4k | 178.7k / 17.6k / 23.7k |
| | 20% | 0 / 0 / 4 | 0 / 0 / 0 | 5 / 0 / 1 | 33 / 4 / 19 | 91 / 13 / 28 | 67.6k / 10.8k / 21.6k | 170.7k / 7.4k / 21.9k | 142.3k / 10.7k / 27k | 148.5k / 33.8k / 77.6k | 430.8k / 16.6k / 52.5k | 351.9k / 35.5k / 52.6k |
| | 30% | 0 / 0 / 4 | 0 / 0 / 3 | 20 / 0 / 10 | 33 / 4 / 19 | 1.1k / 184 / 686 | 107.1k / 15.5k / 37.4k | 267.6k / 14.1k / 38.3k | 222.1k / 17.3k / 40.6k | 224.2k / 50.2k / 125.6k | 632.2k / 27k / 80.8k | 534.1k / 54.9k / 91k |
| | 40% | 0 / 4 / 8 | 0 / 0 / 3 | 20 / 0 / 10 | 88 / 14 / 66 | 2k / 385 / 1.6k | 131.4k / 18.6k / 50k | 342.2k / 21.8k / 56k | 284.9k / 22.4k / 52.7k | 283.9k / 63.5k / 172.6k | 827.8k / 38.8k / 113.4k | 691.8k / 73.7k / 134.5k |
| | 50% | 0 / 4 / 8 | 0 / 0 / 3 | 34 / 0 / 26 | 132 / 18 / 130 | 2.5k / 527 / 2.6k | 153.4k / 21.4k / 65.2k | 408.8k / 31.8k / 79.4k | 359.7k / 28.4k / 71.8k | 342.1k / 76.8k / 241.1k | 1027.7k / 52.5k / 159.7k | 832.8k / 93k / 194.2k |
| Gemini 3 Pro | 10% | 0 / 0 / 0 | 0 / 0 / 0 | 4 / 1 / 1 | 7 / 0 / 1 | 6 / 0 / 6 | 522 / 25 / 53 | 67k / 9.4k / 23.6k | 77.8k / 8.1k / 14.1k | 50.8k / 21.6k / 67.6k | 174.4k / 11.1k / 54.5k | 155.7k / 20.4k / 43.9k |
| | 20% | 2 / 0 / 2 | 3 / 0 / 1 | 4 / 1 / 1 | 46 / 1 / 9 | 72 / 12 / 48 | 84.9k / 4.8k / 10.3k | 132.6k / 19.6k / 47.8k | 140k / 14k / 26k | 97.4k / 43.5k / 119.1k | 361k / 17.5k / 121.6k | 302.1k / 40.9k / 97k |
| | 30% | 2 / 0 / 2 | 3 / 0 / 1 | 16 / 0 / 14 | 46 / 1 / 9 | 1.0k / 220 / 743 | 129.7k / 9k / 21.3k | 204k / 31.5k / 84.4k | 216.8k / 21.3k / 42k | 150.6k / 68.4k / 181k | 526.5k / 21.2k / 192.3k | 442.8k / 61.6k / 175.6k |
| | 40% | 2 / 0 / 4 | 2 / 1 / 9 | 16 / 0 / 14 | 123 / 8 / 37 | 1.9k / 487 / 1.6k | 155.3k / 12.1k / 32.6k | 254.1k / 41.6k / 124.4k | 276.4k / 26.7k / 56.9k | 194.9k / 89.5k / 235.6k | 681.3k / 23.1k / 275.7k | 547.6k / 77.7k / 274.7k |
| | 50% | 8 / 0 / 4 | 2 / 1 / 9 | 26 / 2 / 32 | 177 / 22 / 81 | 2.4k / 686 / 2.4k | 176.5k / 15.2k / 48.2k | 291.8k / 51.6k / 176.6k | 345.5k / 33.1k / 81.3k | 239.5k / 112.9k / 307.6k | 834.5k / 22.6k / 382.8k | 620.2k / 88.7k / 411.1k |
| Llama 3.3 | 10% | 0 / 0 / 0 | 0 / 0 / 0 | 5 / 0 / 1 | 5 / 0 / 3 | 3 / 1 / 8 | 298 / 24 / 278 | 41.4k / 14.1k / 44.5k | 33.3k / 5.5k / 61.2k | 23.9k / 3.9k / 112.3k | 141.8k / 8.3k / 89.9k | 103.6k / 9.8k / 106.6k |
| | 20% | 0 / 0 / 4 | 0 / 0 / 0 | 5 / 0 / 1 | 26 / 1 / 29 | 33 / 11 / 88 | 43.3k / 4k / 52.6k | 81.9k / 24.3k / 93.8k | 53.8k / 8k / 118.2k | 39.7k / 6.6k / 213.7k | 289.1k / 15.8k / 195.1k | 194.4k / 19.3k / 226.3k |
| | 30% | 0 / 0 / 4 | 0 / 0 / 0 | 22 / 0 / 8 | 26 / 1 / 29 | 468 / 140 / 1.4k | 60.8k / 7k / 92.2k | 128.8k / 35.4k / 155.9k | 70.4k / 10.3k / 199.3k | 51.7k / 8.9k / 339.4k | 418.4k / 21.1k / 300.6k | 273k / 27.6k / 377.4k |
| | 40% | 4 / 0 / 0 | 4 / 0 / 0 | 22 / 0 / 8 | 66 / 4 / 98 | 882 / 238 / 2.8k | 69.7k / 9k / 121.2k | 162.3k / 43.9k / 213.8k | 76.7k / 12k / 271.3k | 56.1k / 10.2k / 453.7k | 537.7k / 25.1k / 417.2k | 327.6k / 32.9k / 539.5k |
| | 50% | 12 / 0 / 0 | 12 / 0 / 0 | 36 / 0 / 24 | 104 / 6 / 170 | 1.1k / 266 / 4.1k | 76k / 10.6k / 153.4k | 187.3k / 51.1k / 281.6k | 76.6k / 13.8k / 369.7k | 54.7k / 11k / 594.2k | 654.1k / 27.2k / 558.7k | 358.1k / 35.4k / 726.5k |
| Qwen 3.5 | 10% | 0 / 0 / 0 | 0 / 0 / 0 | 6 / 0 / 0 | 3 / 0 / 5 | 7 / 2 / 3 | 335 / 57 / 208 | 18.4k / 3.8k / 77.9k | 32.1k / 0 / 67.9k | 24.2k / 9.5k / 106.3k | 170.3k / 10.7k / 59k | 74.3k / 8.5k / 137.2k |
| | 20% | 2 / 0 / 2 | 0 / 0 / 0 | 6 / 0 / 0 | 15 / 0 / 41 | 62 / 20 / 50 | 49.5k / 11.4k / 39.1k | 36.5k / 6.7k / 156.7k | 53.5k / 0 / 126.5k | 43.3k / 17.6k / 199.1k | 352.8k / 19.6k / 127.6k | 139.3k / 17.1k / 283.7k |
| | 30% | 2 / 0 / 2 | 4 / 0 / 0 | 22 / 0 / 8 | 15 / 0 / 41 | 873 / 280 / 827 | 69.6k / 20.3k / 70.1k | 54k / 9.2k / 256.8k | 73.3k / 0 / 206.7k | 61.7k / 25.4k / 312.9k | 515.2k / 26.1k / 198.8k | 190.9k / 23.2k / 465.9k |
| | 40% | 2 / 0 / 9 | 12 / 0 / 0 | 22 / 0 / 8 | 30 / 0 / 138 | 1.6k / 529 / 1.9k | 78.2k / 25.8k / 96k | 62k / 10.6k / 347.4k | 83.3k / 0 / 276.7k | 71.6k / 29.7k / 418.7k | 667.3k / 30.9k / 281.8k | 216.4k / 25.4k / 658.2k |
| | 50% | 3 / 0 / 9 | 12 / 0 / 0 | 32 / 0 / 28 | 30 / 0 / 250 | 1.9k / 683 / 3k | 82k / 30.6k / 127.4k | 63.5k / 11.5k / 445k | 86.5k / 0 / 373.5k | 74.4k / 31.1k / 554.5k | 815.7k / 33.2k / 391.1k | 217.7k / 24.6k / 877.7k |
| GLM 5 | 10% | 0 / 0 / 0 | 0 / 0 / 0 | 4 / 0 / 2 | 8 / 0 / 0 | 6 / 0 / 6 | 313 / 0 / 287 | 68.7k / 7.8k / 23.5k | 60.8k / 6.5k / 32.7k | 52.4k / 18.4k / 69.2k | 175.3k / 8.3k / 56.4k | 98.5k / 27.4k / 9.4k |
| | 20% | 2 / 0 / 2 | 0 / 0 / 4 | 4 / 0 / 2 | 50 / 0 / 6 | 71 / 8 / 53 | 52.8k / 49 / 47.2k | 133.0k / 15.6k / 50.8k | 110.9k / 11.2k / 57.9k | 89.3k / 31.1k / 139.6k | 364.3k / 18.6k / 117.1k | 185.4k / 51.5k / 203.1k |
| | 30% | 2 / 0 / 4 | 0 / 0 / 4 | 18 / 0 / 12 | 50 / 0 / 6 | 908 / 167 / 905 | 80.6k / 654 / 78.7k | 202.7k / 24k / 93.3k | 171.1k / 17.3k / 91.7k | 119.6k / 42.2k / 238.3k | 537.6k / 27k / 175.4k | 262.6k / 73.1k / 344.3k |
| | 40% | 0 / 0 / 12 | 0 / 0 / 12 | 18 / 0 / 12 | 130 / 0 / 38 | 1.7k / 323 / 1.9k | 95.2k / 1.8k / 103k | 251.7k / 30.5k / 137.7k | 215.2k / 22k / 122.8k | 133.3k / 48.5k / 338.2k | 706.5k / 34.2k / 239.3k | 315.6k / 88.2k / 496.3k |
| | 50% | 0 / 0 / 12 | 0 / 0 / 12 | 34 / 0 / 26 | 180 / 0 / 100 | 2.2k / 407 / 3k | 105.9k / 3.8k / 130.3k | 290.6k / 36.5k / 193k | 260k / 28k / 172k | 134.7k / 51.8k / 473.5k | 884.2k / 39.9k / 315.9k | 346k / 98.6k / 675.4k |
| Minimax 2.5 | 10% | 0 / 0 / 0 | 0 / 0 / 0 | 4 / 0 / 2 | 6 / 0 / 2 | 3 / 1 / 8 | 132 / 0 / 468 | 6.1k / 1.9k / 92k | 21.5k / 4 / 74.5k | 88.3k / 20.4k / 31.3k | 198.4k / 10.1k / 31.5k | 80.8k / 19.8k / 119.5k |
| | 20% | 2 / 0 / 2 | 2 / 0 / 2 | 4 / 0 / 2 | 36 / 0 / 20 | 27 / 13 / 92 | 16.6k / 0 / 83.4k | 9.4k / 3.8k / 186.8k | 33.2k / 6.4k / 140.4k | 161.7k / 40.8k / 57.5k | 404.8k / 20.3k / 74.8k | 153.8k / 42k / 244.1k |
| | 30% | 2 / 0 / 2 | 4 / 0 / 0 | 17 / 0 / 13 | 36 / 0 / 20 | 274 / 195 / 1.5k | 19k / 0 / 141k | 11.6k / 5.7k / 302.8k | 41.3k / 8k / 230.7k | 239.9k / 64.5k / 95.6k | 585k / 29.3k / 125.7k | 225.2k / 64.4k / 390.4k |
| | 40% | 8 / 0 / 4 | 12 / 0 / 0 | 17 / 0 / 13 | 98 / 4 / 66 | 430 / 364 / 3.2k | 18.2k / 0 / 181.8k | 12.5k / 6.7k / 400.8k | 44.3k / 8.4k / 307.3k | 297.7k / 83.6k / 138.7k | 751.2k / 37.8k / 191.1k | 279k / 83.3k / 537.8k |
| | 50% | 8 / 0 / 4 | 12 / 0 / 0 | 29 / 4 / 27 | 145 / 7 / 128 | 447 / 459 / 4.6k | 16.4k / 0 / 223.6k | 12.8k / 7.1k / 500.1k | 44.9k / 7.8k / 407.4k | 352.1k / 103.8k / 204.1k | 913.3k / 46.5k / 280.3k | 314k / 98.2k / 707.9k |

Table 8: **Predictions on Undirected Edges.** In this experiment, $\{10\%, \ldots, 50\%\}$ edges of the ground truth graph were undirected. The evaluation only tests the tagging quality, disabling any application of Meek rules or acyclicity constraints. All results are shown as *correct predictions / incorrect predictions / no prediction*, where *no prediction* indicates that the tagging algorithm kept the edge undirected.

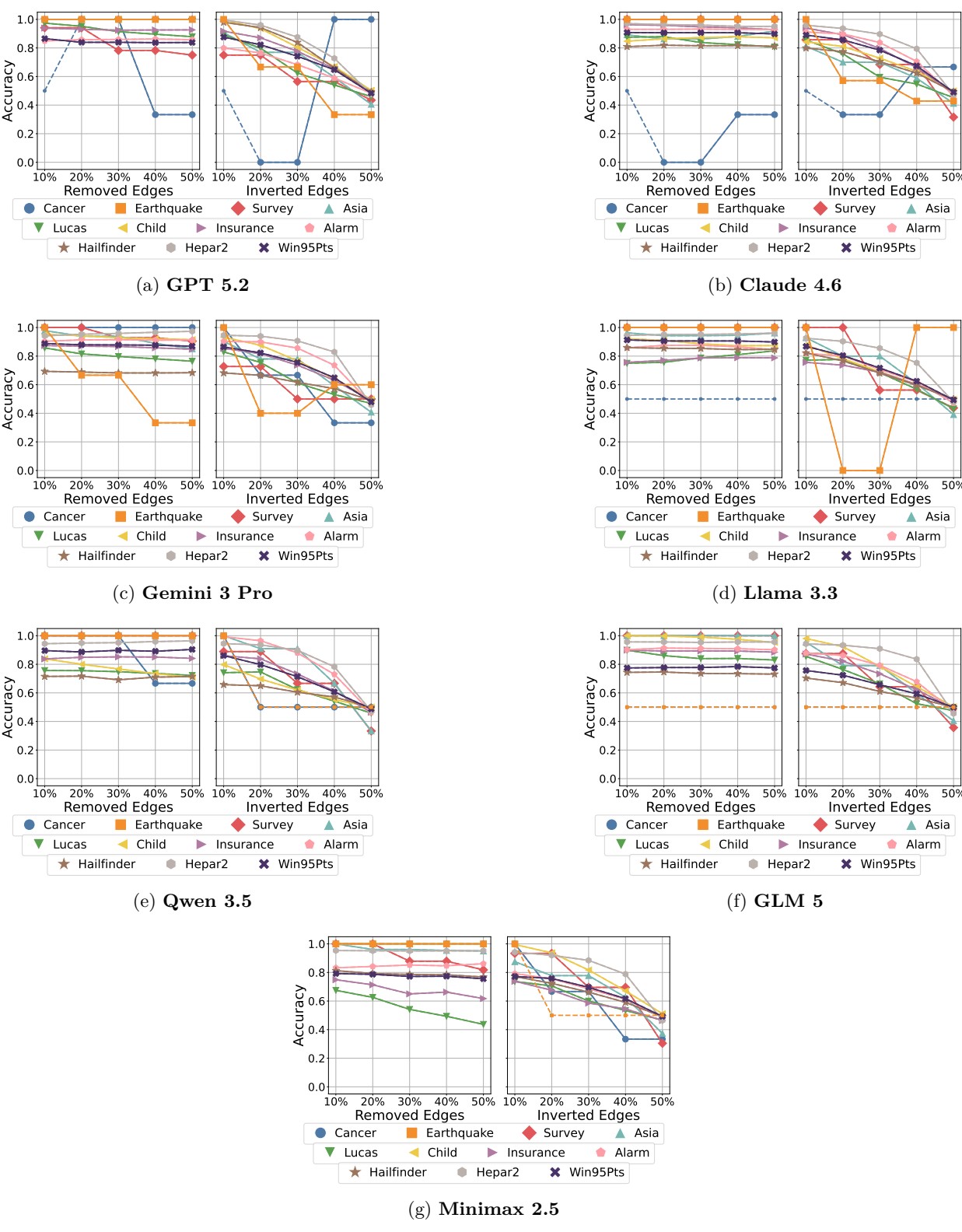

Figure 5: **Accuracy with Errors in the Graph.** Tags were generated by the respective LLM. We can see the same general patterns as for Claude 4.6 for the other LLMs, where removed edges decrease prediction accuracy slightly, while adding errors impacts performance more strongly. (We set accuracy to 50% for datasets where no edge was directed; shown by a smaller point and a dotted line.)

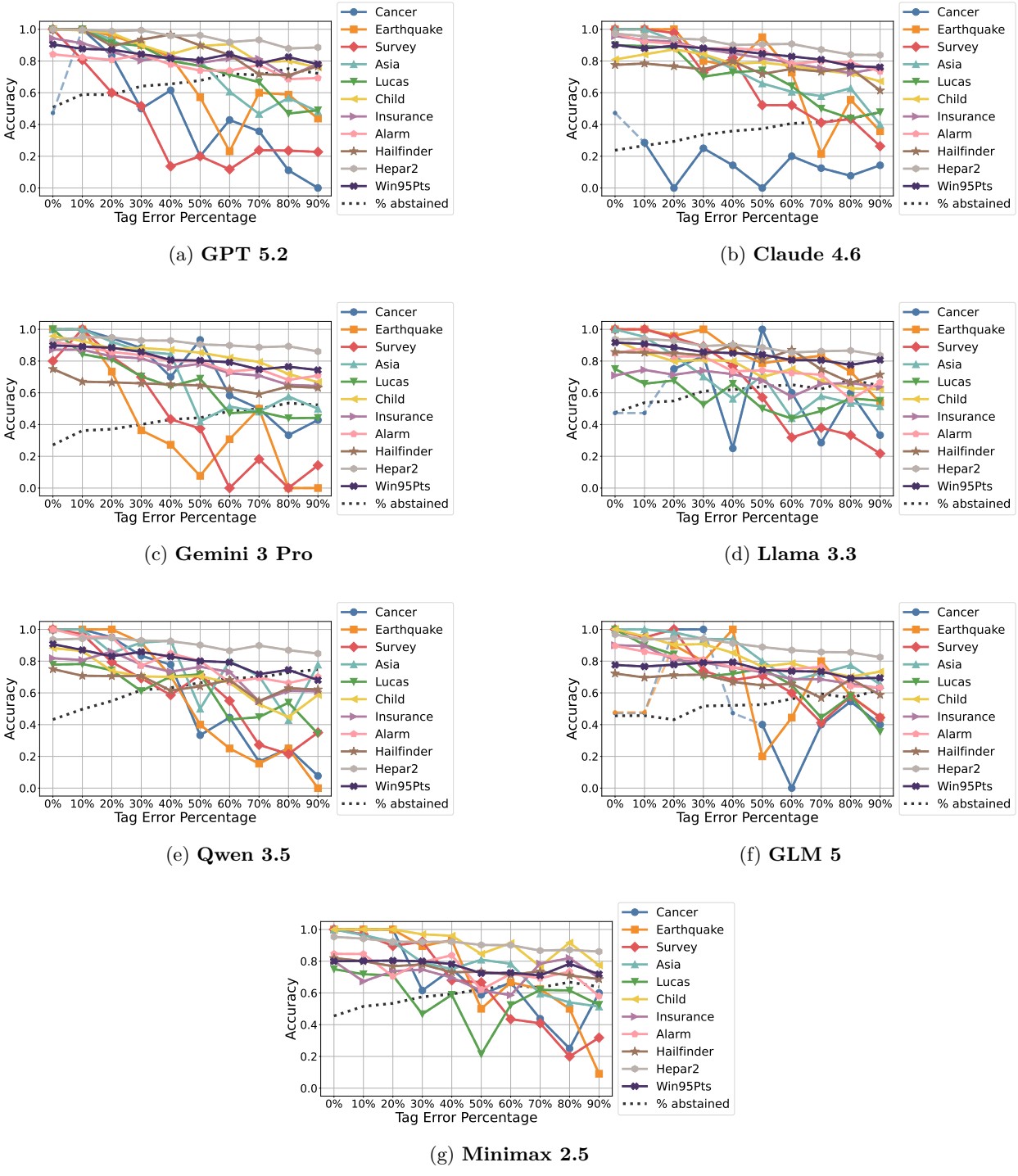

Figure 6: **Accuracy with Noisy Tags.** Tagging predicts the correct edge direction more often than not, even with high tag error percentages. (We set accuracy to 50% for datasets where no edge was directed; shown by a smaller point and a dashed line.) The black, dotted line shows how often the algorithm abstains from making a decision (also see Fig. 7).

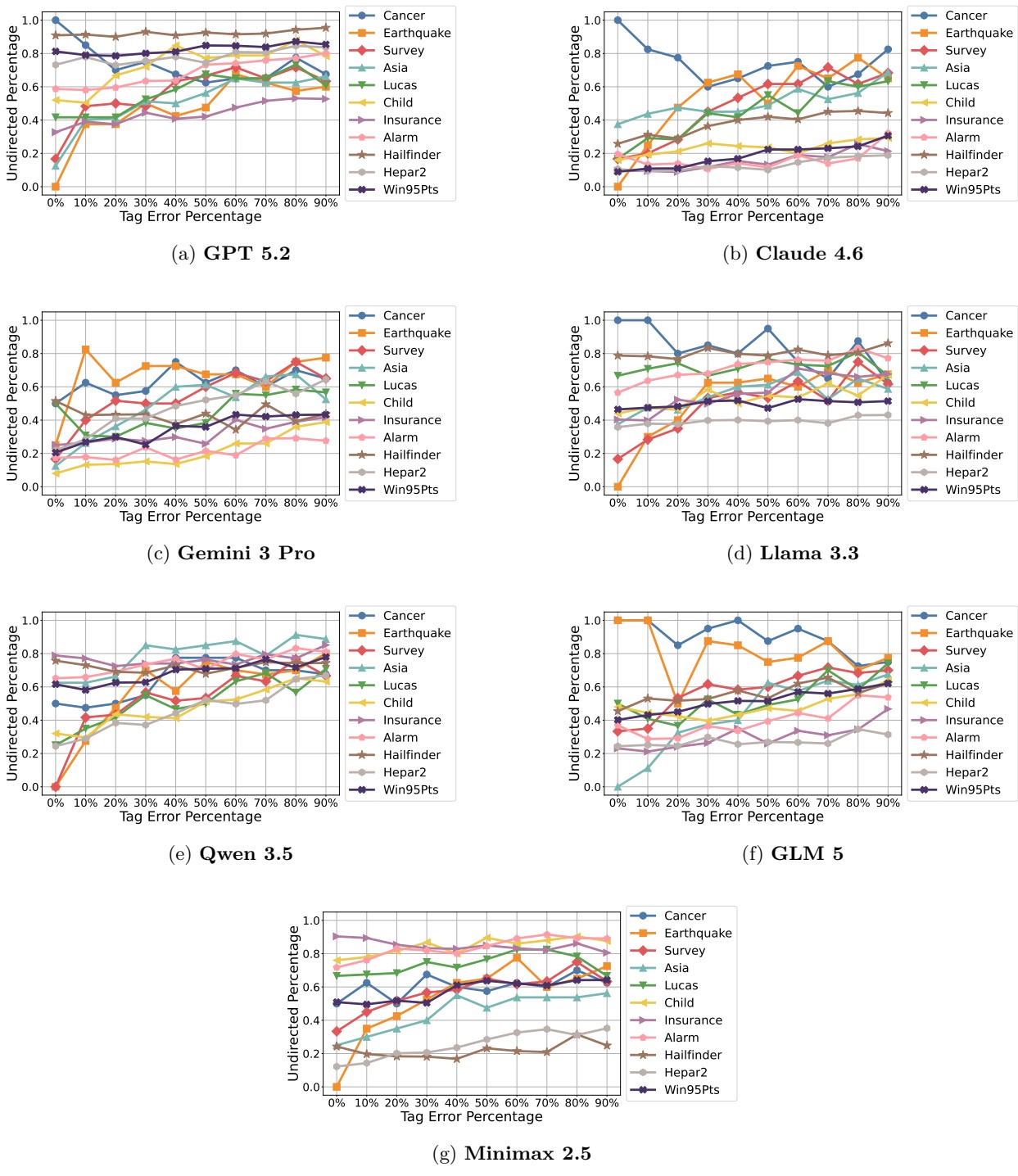

Figure 7: **Percentage of Abstentions under Noisy Tags.** There is a clear general trend where the algorithm abstains from directing an edge more often for the noisier tag settings.

| Evaluation Results | SHD | $\text{SHD}_{\text{double}}$ | $\text{SID}_{\text{min}}$ | $\text{SID}_{\text{max}}$ | Precision | Recall | $F_1$ |
|---|---|---|---|---|---|---|---|
| Dataset Cancer | | | | | | | |
| Tagged-PC | $\mathbf{2.00}_{\pm 0.45}$ | $\mathbf{3.90}_{\pm 0.70}$ | $\mathbf{9.60}_{\pm 1.56}$ | $\mathbf{9.60}_{\pm 1.56}$ | $\mathbf{0.51}_{\pm 0.09}$ | $\mathbf{0.50}_{\pm 0.11}$ | $\mathbf{0.50}_{\pm 0.10}$ |
| Tagged-PC (N) | $\mathbf{2.00}_{\pm 0.45}$ | $\mathbf{3.90}_{\pm 0.70}$ | $\mathbf{9.60}_{\pm 1.56}$ | $\mathbf{9.60}_{\pm 1.56}$ | $\mathbf{0.51}_{\pm 0.09}$ | $\mathbf{0.50}_{\pm 0.11}$ | $\mathbf{0.50}_{\pm 0.10}$ |
| Tagged-GES | $\mathbf{0.80}_{\pm 1.60}$ | $\mathbf{0.80}_{\pm 1.60}$ | $\mathbf{0.20}_{\pm 0.40}$ | $\mathbf{2.20}_{\pm 4.40}$ | $\mathbf{0.80}_{\pm 0.40}$ | $\mathbf{0.80}_{\pm 0.40}$ | $\mathbf{0.80}_{\pm 0.40}$ |
| Tagged-GES (N) | $\mathbf{0.80}_{\pm 1.60}$ | $\mathbf{0.80}_{\pm 1.60}$ | $\mathbf{0.20}_{\pm 0.40}$ | $\mathbf{2.20}_{\pm 4.40}$ | $\mathbf{0.80}_{\pm 0.40}$ | $\mathbf{0.80}_{\pm 0.40}$ | $\mathbf{0.80}_{\pm 0.40}$ |
| Dataset Earthquake | | | | | | | |
| Tagged-PC | $\mathbf{0.60}_{\pm 1.50}$ | $\mathbf{0.80}_{\pm 1.83}$ | $\mathbf{0.00}_{\pm 0.00}$ | $\mathbf{1.40}_{\pm 4.20}$ | $\mathbf{0.90}_{\pm 0.30}$ | $\mathbf{0.90}_{\pm 0.30}$ | $\mathbf{0.90}_{\pm 0.30}$ |
| Tagged-PC (N) | $\mathbf{0.60}_{\pm 1.50}$ | $\mathbf{0.80}_{\pm 1.83}$ | $\mathbf{0.00}_{\pm 0.00}$ | $\mathbf{1.40}_{\pm 4.20}$ | $\mathbf{0.90}_{\pm 0.30}$ | $\mathbf{0.90}_{\pm 0.30}$ | $\mathbf{0.90}_{\pm 0.30}$ |
| Tagged-GES | $\mathbf{0.00}_{\pm 0.00}$ | $\mathbf{0.00}_{\pm 0.00}$ | $\mathbf{0.00}_{\pm 0.00}$ | $\mathbf{0.00}_{\pm 0.00}$ | $\mathbf{1.00}_{\pm 0.00}$ | $\mathbf{1.00}_{\pm 0.00}$ | $\mathbf{1.00}_{\pm 0.00}$ |
| Tagged-GES (N) | $\mathbf{0.00}_{\pm 0.00}$ | $\mathbf{0.00}_{\pm 0.00}$ | $\mathbf{0.00}_{\pm 0.00}$ | $\mathbf{0.00}_{\pm 0.00}$ | $\mathbf{1.00}_{\pm 0.00}$ | $\mathbf{1.00}_{\pm 0.00}$ | $\mathbf{1.00}_{\pm 0.00}$ |
| Dataset Survey | | | | | | | |
| Tagged-PC | $\mathbf{2.20}_{\pm 0.40}$ | $\mathbf{4.30}_{\pm 0.64}$ | $11.20_{\pm 2.04}$ | $\mathbf{11.20}_{\pm 2.04}$ | $\mathbf{0.64}_{\pm 0.05}$ | $\mathbf{0.63}_{\pm 0.07}$ | $\mathbf{0.64}_{\pm 0.06}$ |
| Tagged-PC (N) | $2.30_{\pm 0.64}$ | $\mathbf{4.30}_{\pm 0.64}$ | $\mathbf{11.10}_{\pm 1.92}$ | $11.80_{\pm 2.44}$ | $\mathbf{0.64}_{\pm 0.05}$ | $0.62_{\pm 0.11}$ | $0.63_{\pm 0.08}$ |
| Tagged-GES | $4.20_{\pm 1.89}$ | $4.30_{\pm 1.79}$ | $\mathbf{8.50}_{\pm 2.06}$ | $15.80_{\pm 7.52}$ | $\mathbf{0.48}_{\pm 0.48}$ | $0.30_{\pm 0.31}$ | $0.36_{\pm 0.37}$ |
| Tagged-GES (N) | $\mathbf{4.10}_{\pm 1.97}$ | $\mathbf{4.20}_{\pm 1.89}$ | $\mathbf{8.50}_{\pm 2.06}$ | $\mathbf{15.70}_{\pm 7.60}$ | $\mathbf{0.48}_{\pm 0.48}$ | $\mathbf{0.32}_{\pm 0.33}$ | $\mathbf{0.38}_{\pm 0.38}$ |
| Dataset Asia | | | | | | | |
| Tagged-PC | $5.10_{\pm 0.30}$ | $5.40_{\pm 0.92}$ | $\mathbf{14.90}_{\pm 5.39}$ | $24.20_{\pm 3.12}$ | $\mathbf{0.93}_{\pm 0.16}$ | $0.38_{\pm 0.00}$ | $0.53_{\pm 0.04}$ |
| Tagged-PC (N) | $\mathbf{4.10}_{\pm 1.22}$ | $\mathbf{4.40}_{\pm 1.62}$ | $15.10_{\pm 4.21}$ | $\mathbf{20.80}_{\pm 6.65}$ | $0.91_{\pm 0.16}$ | $\mathbf{0.50}_{\pm 0.16}$ | $\mathbf{0.63}_{\pm 0.16}$ |
| Tagged-GES | $3.00_{\pm 0.00}$ | $3.00_{\pm 0.00}$ | $\mathbf{0.80}_{\pm 0.40}$ | $8.80_{\pm 1.60}$ | $\mathbf{1.00}_{\pm 0.00}$ | $0.62_{\pm 0.00}$ | $0.77_{\pm 0.00}$ |
| Tagged-GES (N) | $\mathbf{2.00}_{\pm 0.77}$ | $\mathbf{2.60}_{\pm 1.11}$ | $4.60_{\pm 3.93}$ | $\mathbf{6.70}_{\pm 4.03}$ | $0.92_{\pm 0.09}$ | $\mathbf{0.75}_{\pm 0.10}$ | $\mathbf{0.82}_{\pm 0.08}$ |
| Dataset Lucas | | | | | | | |
| Tagged-PC | $\mathbf{0.00}_{\pm 0.00}$ | $\mathbf{0.00}_{\pm 0.00}$ | $\mathbf{0.00}_{\pm 0.00}$ | $\mathbf{0.00}_{\pm 0.00}$ | $\mathbf{1.00}_{\pm 0.00}$ | $\mathbf{1.00}_{\pm 0.00}$ | $\mathbf{1.00}_{\pm 0.00}$ |
| Tagged-PC (N) | $0.30_{\pm 0.46}$ | $0.30_{\pm 0.46}$ | $\mathbf{0.00}_{\pm 0.00}$ | $2.10_{\pm 3.21}$ | $\mathbf{1.00}_{\pm 0.00}$ | $0.97_{\pm 0.04}$ | $0.99_{\pm 0.02}$ |
| Tagged-GES | $\mathbf{0.00}_{\pm 0.00}$ | $\mathbf{0.00}_{\pm 0.00}$ | $\mathbf{0.00}_{\pm 0.00}$ | $\mathbf{0.00}_{\pm 0.00}$ | $\mathbf{1.00}_{\pm 0.00}$ | $\mathbf{1.00}_{\pm 0.00}$ | $\mathbf{1.00}_{\pm 0.00}$ |
| Tagged-GES (N) | $0.60_{\pm 0.49}$ | $0.70_{\pm 0.64}$ | $1.00_{\pm 3.00}$ | $4.50_{\pm 3.77}$ | $0.99_{\pm 0.03}$ | $0.95_{\pm 0.04}$ | $0.97_{\pm 0.03}$ |
| Dataset Child | | | | | | | |
| Tagged-PC | $\mathbf{1.30}_{\pm 1.19}$ | $\mathbf{2.50}_{\pm 2.42}$ | $\mathbf{21.00}_{\pm 23.48}$ | $\mathbf{21.00}_{\pm 23.48}$ | $\mathbf{0.95}_{\pm 0.05}$ | $\mathbf{0.95}_{\pm 0.05}$ | $\mathbf{0.95}_{\pm 0.05}$ |
| Tagged-PC (N) | $4.10_{\pm 1.51}$ | $7.50_{\pm 2.33}$ | $76.00_{\pm 22.26}$ | $82.60_{\pm 32.13}$ | $0.86_{\pm 0.04}$ | $0.84_{\pm 0.06}$ | $0.85_{\pm 0.05}$ |
| Tagged-GES | $3.00_{\pm 0.00}$ | $3.00_{\pm 0.00}$ | $\mathbf{0.00}_{\pm 0.00}$ | $44.00_{\pm 0.00}$ | $\mathbf{1.00}_{\pm 0.00}$ | $0.88_{\pm 0.00}$ | $\mathbf{0.94}_{\pm 0.00}$ |
| Tagged-GES (N) | $\mathbf{2.50}_{\pm 1.28}$ | $4.30_{\pm 1.90}$ | $29.80_{\pm 19.40}$ | $\mathbf{37.30}_{\pm 19.57}$ | $0.93_{\pm 0.04}$ | $\mathbf{0.90}_{\pm 0.05}$ | $0.91_{\pm 0.04}$ |
| Dataset Alarm | | | | | | | |
| Tagged-PC | $\mathbf{7.10}_{\pm 3.11}$ | $\mathbf{9.20}_{\pm 4.85}$ | $\mathbf{72.40}_{\pm 45.53}$ | $\mathbf{81.70}_{\pm 50.81}$ | $\mathbf{0.93}_{\pm 0.06}$ | $\mathbf{0.88}_{\pm 0.05}$ | $\mathbf{0.91}_{\pm 0.05}$ |
| Tagged-PC (N) | $8.00_{\pm 2.79}$ | $10.50_{\pm 5.20}$ | $74.90_{\pm 55.18}$ | $87.10_{\pm 49.88}$ | $0.91_{\pm 0.07}$ | $0.86_{\pm 0.04}$ | $0.89_{\pm 0.05}$ |
| Tagged-GES | $\mathbf{4.50}_{\pm 1.28}$ | $\mathbf{4.50}_{\pm 1.28}$ | $\mathbf{18.40}_{\pm 0.80}$ | $\mathbf{36.40}_{\pm 0.80}$ | $\mathbf{0.97}_{\pm 0.02}$ | $\mathbf{0.93}_{\pm 0.01}$ | $\mathbf{0.95}_{\pm 0.01}$ |
| Tagged-GES (N) | $6.70_{\pm 2.24}$ | $9.00_{\pm 2.97}$ | $61.80_{\pm 18.25}$ | $74.00_{\pm 21.15}$ | $0.92_{\pm 0.04}$ | $0.88_{\pm 0.03}$ | $0.90_{\pm 0.03}$ |
| Dataset Insurance | | | | | | | |
| Tagged-PC | $\mathbf{20.90}_{\pm 2.21}$ | $\mathbf{30.00}_{\pm 4.00}$ | $\mathbf{389.60}_{\pm 48.01}$ | $\mathbf{390.20}_{\pm 48.46}$ | $\mathbf{0.77}_{\pm 0.05}$ | $\mathbf{0.61}_{\pm 0.04}$ | $\mathbf{0.68}_{\pm 0.04}$ |
| Tagged-PC (N) | $21.50_{\pm 2.54}$ | $31.50_{\pm 4.90}$ | $392.70_{\pm 47.13}$ | $395.70_{\pm 50.13}$ | $0.75_{\pm 0.06}$ | $0.60_{\pm 0.05}$ | $0.66_{\pm 0.05}$ |
| Tagged-GES | $\mathbf{18.40}_{\pm 4.15}$ | $\mathbf{22.40}_{\pm 6.55}$ | $\mathbf{250.30}_{\pm 37.97}$ | $\mathbf{259.30}_{\pm 30.59}$ | $\mathbf{0.86}_{\pm 0.08}$ | $\mathbf{0.69}_{\pm 0.06}$ | $\mathbf{0.76}_{\pm 0.07}$ |
| Tagged-GES (N) | $18.90_{\pm 4.83}$ | $23.30_{\pm 7.20}$ | $254.70_{\pm 46.74}$ | $275.50_{\pm 42.30}$ | $0.84_{\pm 0.09}$ | $0.68_{\pm 0.07}$ | $0.75_{\pm 0.08}$ |
| Dataset Hailfinder | | | | | | | |
| Tagged-PC | $\mathbf{41.90}_{\pm 3.45}$ | $\mathbf{44.70}_{\pm 3.93}$ | — | — | $\mathbf{0.80}_{\pm 0.04}$ | $\mathbf{0.45}_{\pm 0.04}$ | $\mathbf{0.58}_{\pm 0.04}$ |
| Tagged-PC (N) | $43.70_{\pm 2.87}$ | $47.60_{\pm 4.39}$ | — | — | $0.78_{\pm 0.07}$ | $0.43_{\pm 0.03}$ | $0.55_{\pm 0.04}$ |
| Tagged-GES | $\mathbf{39.00}_{\pm 15.21}$ | $\mathbf{42.20}_{\pm 15.28}$ | — | — | $\mathbf{0.69}_{\pm 0.12}$ | $\mathbf{0.65}_{\pm 0.12}$ | $\mathbf{0.67}_{\pm 0.12}$ |
| Tagged-GES (N) | $\mathbf{39.00}_{\pm 15.21}$ | $\mathbf{42.20}_{\pm 15.28}$ | — | — | $\mathbf{0.69}_{\pm 0.12}$ | $\mathbf{0.65}_{\pm 0.12}$ | $\mathbf{0.67}_{\pm 0.12}$ |
| Dataset Hepar2 | | | | | | | |
| Tagged-PC | $94.20_{\pm 9.90}$ | $127.90_{\pm 19.41}$ | — | — | $0.48_{\pm 0.13}$ | $0.28_{\pm 0.08}$ | $0.35_{\pm 0.10}$ |
| Tagged-PC (N) | $\mathbf{91.50}_{\pm 5.55}$ | $\mathbf{123.00}_{\pm 11.34}$ | — | — | $\mathbf{0.51}_{\pm 0.08}$ | $\mathbf{0.30}_{\pm 0.04}$ | $\mathbf{0.38}_{\pm 0.05}$ |
| Tagged-GES | $\mathbf{50.30}_{\pm 1.55}$ | $\mathbf{55.00}_{\pm 2.32}$ | — | — | $\mathbf{0.94}_{\pm 0.02}$ | $\mathbf{0.59}_{\pm 0.01}$ | $\mathbf{0.73}_{\pm 0.01}$ |
| Tagged-GES (N) | $53.80_{\pm 2.75}$ | $61.70_{\pm 3.23}$ | — | — | $0.90_{\pm 0.01}$ | $0.56_{\pm 0.02}$ | $0.69_{\pm 0.02}$ |
| Dataset Win95Pts | | | | | | | |
| Tagged-PC | $\mathbf{48.70}_{\pm 3.61}$ | $\mathbf{57.00}_{\pm 6.36}$ | — | — | $\mathbf{0.84}_{\pm 0.04}$ | $\mathbf{0.61}_{\pm 0.02}$ | $\mathbf{0.71}_{\pm 0.03}$ |
| Tagged-PC (N) | $52.00_{\pm 3.79}$ | $60.30_{\pm 5.69}$ | — | — | $0.83_{\pm 0.04}$ | $0.58_{\pm 0.03}$ | $0.69_{\pm 0.03}$ |
| Tagged-GES | $\mathbf{45.90}_{\pm 5.96}$ | $\mathbf{50.00}_{\pm 6.29}$ | — | — | $\mathbf{0.79}_{\pm 0.03}$ | $\mathbf{0.76}_{\pm 0.03}$ | $\mathbf{0.77}_{\pm 0.03}$ |
| Tagged-GES (N) | $49.30_{\pm 5.66}$ | $59.30_{\pm 6.17}$ | — | — | $0.74_{\pm 0.03}$ | $0.73_{\pm 0.03}$ | $0.73_{\pm 0.03}$ |

Table 9: **Absolute Scores Comparing Normal and Noisy Tags.** Results for the standard tag sets are the same as in Tab. 5. Below each method, we include the results for noisy (N) tags. Despite highly noisy tags, performance only decreases slightly.

| | Fewer Tags | SHD | $\text{SHD}_{\text{double}}$ | $\text{SID}_{\text{min}}$ | $\text{SID}_{\text{max}}$ | Precision | Recall | $F_1$ |
|---|---|---|---|---|---|---|---|---|
| **Cancer** | | | | | | | | |
| | 0.00 | $0.80_{\pm1.60}$ | $0.80_{\pm1.60}$ | $0.20_{\pm0.40}$ | $2.20_{\pm4.40}$ | $0.80_{\pm0.40}$ | $0.80_{\pm0.40}$ | $0.80_{\pm0.40}$ |
| | 0.25 | $0.80_{\pm1.60}$ | $0.80_{\pm1.60}$ | $0.20_{\pm0.40}$ | $2.20_{\pm4.40}$ | $0.80_{\pm0.40}$ | $0.80_{\pm0.40}$ | $0.80_{\pm0.40}$ |
| | 0.50 | $0.80_{\pm1.60}$ | $0.80_{\pm1.60}$ | $0.20_{\pm0.40}$ | $2.20_{\pm4.40}$ | $0.80_{\pm0.40}$ | $0.80_{\pm0.40}$ | $0.80_{\pm0.40}$ |
| | 0.75 | $0.80_{\pm1.60}$ | $0.80_{\pm1.60}$ | $0.20_{\pm0.40}$ | $2.20_{\pm4.40}$ | $0.80_{\pm0.40}$ | $0.80_{\pm0.40}$ | $0.80_{\pm0.40}$ |
| **Earthquake** | | | | | | | | |
| | 0.00 | $0.00_{\pm0.00}$ | $0.00_{\pm0.00}$ | $0.00_{\pm0.00}$ | $0.00_{\pm0.00}$ | $1.00_{\pm0.00}$ | $1.00_{\pm0.00}$ | $1.00_{\pm0.00}$ |
| | 0.25 | $0.00_{\pm0.00}$ | $0.00_{\pm0.00}$ | $0.00_{\pm0.00}$ | $0.00_{\pm0.00}$ | $1.00_{\pm0.00}$ | $1.00_{\pm0.00}$ | $1.00_{\pm0.00}$ |
| | 0.50 | $0.00_{\pm0.00}$ | $0.00_{\pm0.00}$ | $0.00_{\pm0.00}$ | $0.00_{\pm0.00}$ | $1.00_{\pm0.00}$ | $1.00_{\pm0.00}$ | $1.00_{\pm0.00}$ |
| | 0.75 | $0.00_{\pm0.00}$ | $0.00_{\pm0.00}$ | $0.00_{\pm0.00}$ | $0.00_{\pm0.00}$ | $1.00_{\pm0.00}$ | $1.00_{\pm0.00}$ | $1.00_{\pm0.00}$ |
| **Survey** | | | | | | | | |
| | 0.00 | $4.20_{\pm1.89}$ | $4.30_{\pm1.79}$ | $8.50_{\pm2.06}$ | $15.80_{\pm7.52}$ | $0.48_{\pm0.48}$ | $0.30_{\pm0.31}$ | $0.36_{\pm0.37}$ |
| | 0.25 | $4.20_{\pm1.89}$ | $4.30_{\pm1.79}$ | $8.50_{\pm2.06}$ | $15.80_{\pm7.52}$ | $0.48_{\pm0.48}$ | $0.30_{\pm0.31}$ | $0.36_{\pm0.37}$ |
| | 0.50 | $4.20_{\pm1.89}$ | $4.30_{\pm1.79}$ | $8.50_{\pm2.06}$ | $15.80_{\pm7.52}$ | $0.48_{\pm0.48}$ | $0.30_{\pm0.31}$ | $0.36_{\pm0.37}$ |
| | 0.75 | $4.20_{\pm1.89}$ | $4.30_{\pm1.79}$ | $8.50_{\pm2.06}$ | $15.80_{\pm7.52}$ | $0.48_{\pm0.48}$ | $0.30_{\pm0.31}$ | $0.36_{\pm0.37}$ |
| **Asia** | | | | | | | | |
| | 0.00 | $3.00_{\pm0.00}$ | $3.00_{\pm0.00}$ | $0.80_{\pm0.40}$ | $8.80_{\pm1.60}$ | $1.00_{\pm0.00}$ | $0.62_{\pm0.00}$ | $0.77_{\pm0.00}$ |
| | 0.25 | $3.00_{\pm0.00}$ | $3.00_{\pm0.00}$ | $0.80_{\pm0.40}$ | $8.80_{\pm1.60}$ | $1.00_{\pm0.00}$ | $0.62_{\pm0.00}$ | $0.77_{\pm0.00}$ |
| | 0.50 | $3.10_{\pm0.30}$ | $3.20_{\pm0.60}$ | $1.70_{\pm2.79}$ | $9.70_{\pm2.90}$ | $0.98_{\pm0.06}$ | $0.61_{\pm0.04}$ | $0.75_{\pm0.05}$ |
| | 0.75 | $3.10_{\pm0.30}$ | $3.20_{\pm0.60}$ | $1.70_{\pm2.79}$ | $9.70_{\pm2.90}$ | $0.98_{\pm0.06}$ | $0.61_{\pm0.04}$ | $0.75_{\pm0.05}$ |
| **Lucas** | | | | | | | | |
| | 0.00 | $0.00_{\pm0.00}$ | $0.00_{\pm0.00}$ | $0.00_{\pm0.00}$ | $0.00_{\pm0.00}$ | $1.00_{\pm0.00}$ | $1.00_{\pm0.00}$ | $1.00_{\pm0.00}$ |
| | 0.25 | $0.70_{\pm0.46}$ | $0.70_{\pm0.46}$ | $0.00_{\pm0.00}$ | $4.90_{\pm3.21}$ | $1.00_{\pm0.00}$ | $0.94_{\pm0.04}$ | $0.97_{\pm0.02}$ |
| | 0.50 | $1.60_{\pm1.02}$ | $2.40_{\pm2.24}$ | $6.60_{\pm10.11}$ | $12.20_{\pm8.27}$ | $0.93_{\pm0.11}$ | $0.87_{\pm0.08}$ | $0.90_{\pm0.10}$ |
| | 0.75 | $1.20_{\pm0.60}$ | $1.50_{\pm1.02}$ | $4.50_{\pm6.92}$ | $10.80_{\pm7.65}$ | $0.97_{\pm0.04}$ | $0.90_{\pm0.05}$ | $0.93_{\pm0.04}$ |
| **Child** | | | | | | | | |
| | 0.00 | $3.00_{\pm0.00}$ | $3.00_{\pm0.00}$ | $0.00_{\pm0.00}$ | $44.00_{\pm0.00}$ | $1.00_{\pm0.00}$ | $0.88_{\pm0.00}$ | $0.94_{\pm0.00}$ |
| | 0.25 | $6.30_{\pm2.45}$ | $6.30_{\pm2.45}$ | $0.00_{\pm0.00}$ | $90.20_{\pm51.38}$ | $1.00_{\pm0.00}$ | $0.75_{\pm0.10}$ | $0.85_{\pm0.06}$ |
| | 0.50 | $7.50_{\pm2.58}$ | $7.80_{\pm2.32}$ | $5.80_{\pm17.40}$ | $124.60_{\pm59.79}$ | $0.99_{\pm0.04}$ | $0.70_{\pm0.10}$ | $0.81_{\pm0.06}$ |
| | 0.75 | $9.10_{\pm2.51}$ | $9.10_{\pm2.51}$ | $0.00_{\pm0.00}$ | $137.30_{\pm76.75}$ | $1.00_{\pm0.00}$ | $0.64_{\pm0.10}$ | $0.77_{\pm0.07}$ |
| **Alarm** | | | | | | | | |
| | 0.00 | $4.50_{\pm1.28}$ | $4.50_{\pm1.28}$ | $18.40_{\pm0.80}$ | $36.40_{\pm0.80}$ | $0.97_{\pm0.02}$ | $0.93_{\pm0.01}$ | $0.95_{\pm0.01}$ |
| | 0.25 | $4.90_{\pm1.87}$ | $5.60_{\pm2.33}$ | $34.00_{\pm17.48}$ | $46.60_{\pm18.76}$ | $0.96_{\pm0.03}$ | $0.92_{\pm0.02}$ | $0.94_{\pm0.03}$ |
| | 0.50 | $6.30_{\pm2.00}$ | $7.70_{\pm2.87}$ | $55.40_{\pm24.74}$ | $76.40_{\pm28.19}$ | $0.94_{\pm0.04}$ | $0.89_{\pm0.03}$ | $0.91_{\pm0.03}$ |
| | 0.75 | $8.00_{\pm2.49}$ | $10.00_{\pm3.82}$ | $69.80_{\pm33.01}$ | $99.60_{\pm32.56}$ | $0.92_{\pm0.05}$ | $0.85_{\pm0.04}$ | $0.89_{\pm0.04}$ |
| **Insurance** | | | | | | | | |
| | 0.00 | $18.40_{\pm4.15}$ | $22.40_{\pm6.55}$ | $250.30_{\pm37.97}$ | $259.30_{\pm30.59}$ | $0.86_{\pm0.08}$ | $0.69_{\pm0.06}$ | $0.76_{\pm0.07}$ |
| | 0.25 | $18.70_{\pm4.38}$ | $22.90_{\pm6.71}$ | $246.60_{\pm45.76}$ | $270.10_{\pm32.37}$ | $0.85_{\pm0.08}$ | $0.68_{\pm0.06}$ | $0.76_{\pm0.07}$ |
| | 0.50 | $20.70_{\pm3.47}$ | $24.60_{\pm5.37}$ | $244.80_{\pm63.22}$ | $302.30_{\pm36.76}$ | $0.85_{\pm0.09}$ | $0.64_{\pm0.05}$ | $0.73_{\pm0.06}$ |
| | 0.75 | $21.50_{\pm4.43}$ | $25.10_{\pm7.08}$ | $241.70_{\pm71.67}$ | $320.50_{\pm39.15}$ | $0.85_{\pm0.10}$ | $0.63_{\pm0.07}$ | $0.72_{\pm0.08}$ |
| **Hailfinder** | | | | | | | | |
| | 0.00 | $39.00_{\pm15.21}$ | $42.20_{\pm15.28}$ | - | - | $0.69_{\pm0.12}$ | $0.65_{\pm0.12}$ | $0.67_{\pm0.12}$ |
| | 0.25 | $39.00_{\pm15.21}$ | $42.20_{\pm15.28}$ | - | - | $0.69_{\pm0.12}$ | $0.65_{\pm0.12}$ | $0.67_{\pm0.12}$ |
| | 0.50 | $39.00_{\pm15.21}$ | $42.20_{\pm15.28}$ | - | - | $0.69_{\pm0.12}$ | $0.65_{\pm0.12}$ | $0.67_{\pm0.12}$ |
| | 0.75 | $39.00_{\pm15.21}$ | $42.20_{\pm15.28}$ | - | - | $0.69_{\pm0.12}$ | $0.65_{\pm0.12}$ | $0.67_{\pm0.12}$ |
| **Hepar2** | | | | | | | | |
| | 0.00 | $50.30_{\pm1.55}$ | $55.00_{\pm2.32}$ | - | - | $0.94_{\pm0.02}$ | $0.59_{\pm0.01}$ | $0.73_{\pm0.01}$ |
| | 0.25 | $52.80_{\pm3.03}$ | $58.80_{\pm5.00}$ | - | - | $0.92_{\pm0.03}$ | $0.57_{\pm0.02}$ | $0.71_{\pm0.03}$ |
| | 0.50 | $54.70_{\pm2.37}$ | $61.70_{\pm3.07}$ | - | - | $0.91_{\pm0.02}$ | $0.56_{\pm0.02}$ | $0.69_{\pm0.02}$ |
| | 0.75 | $57.40_{\pm3.53}$ | $64.80_{\pm3.57}$ | - | - | $0.90_{\pm0.01}$ | $0.53_{\pm0.03}$ | $0.67_{\pm0.02}$ |
| **Win95Pts** | | | | | | | | |
| | 0.00 | $45.90_{\pm5.96}$ | $50.00_{\pm6.29}$ | - | - | $0.79_{\pm0.03}$ | $0.76_{\pm0.03}$ | $0.77_{\pm0.03}$ |
| | 0.25 | $47.90_{\pm6.55}$ | $53.20_{\pm7.51}$ | - | - | $0.77_{\pm0.03}$ | $0.74_{\pm0.04}$ | $0.76_{\pm0.03}$ |
| | 0.50 | $50.30_{\pm6.99}$ | $56.30_{\pm8.22}$ | - | - | $0.76_{\pm0.04}$ | $0.72_{\pm0.04}$ | $0.74_{\pm0.04}$ |
| | 0.75 | $52.00_{\pm5.98}$ | $59.10_{\pm6.70}$ | - | - | $0.75_{\pm0.03}$ | $0.71_{\pm0.03}$ | $0.73_{\pm0.03}$ |

Table 10: **Reduced-Tags Evaluation Results for Tagged-GES.** Each row reports the mean over 10 seeds. The performance decreases or stays the same with fewer tags.

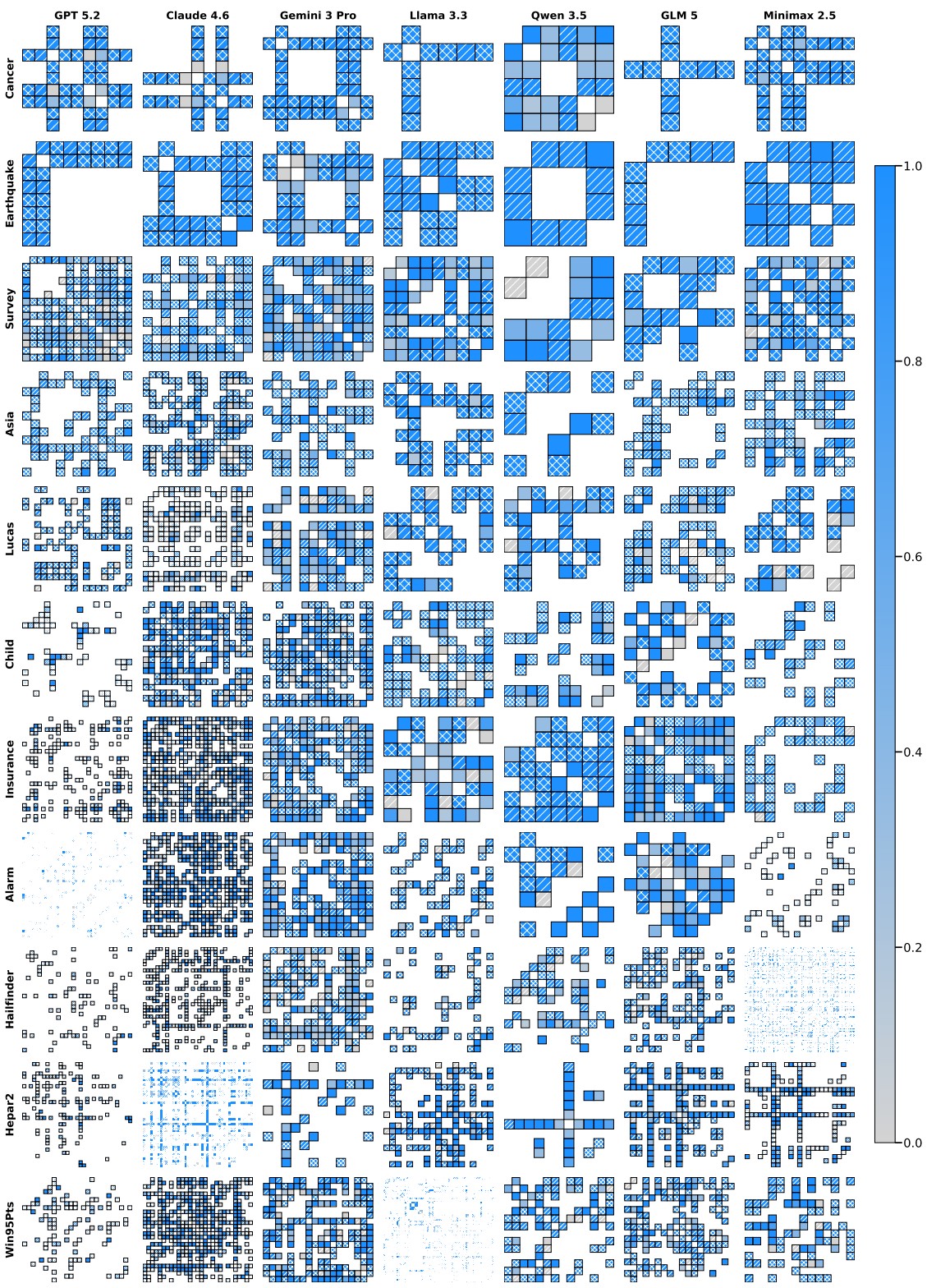

Figure 8: **Tag Homogeneity for all LLMs.** A tag relation has high homogeneity (1) if variables of one tag consistently cause the other and low homogeneity (0) if the cause-effect relationship between these variables is random. We observe high homogeneity throughout all datasets. Hatched areas might appear white on screen due to anti-aliasing.

**Cancer**

| | | |
|---|---|---|
| "respiratory" → "disease" | | (66% / 3 ) |

**Earthquake**

| | | |
|---|---|---|
| "security_related" → "notification" | | (100% / 3 ) |

**Survey**

| | | |
|---|---|---|
| "personal_attribute" → "choice_dependent" | | (100% / 4 ) |
| "personal_attribute" → "socioeconomic" | | (100% / 4 ) |
| "socioeconomic" → "geographic" | | (100% / 3 ) |
| "personal_attribute" → "human_capital" | | (100% / 3 ) |
| "time_dependent" → "choice_dependent" | | (80% / 5 ) |
| "choice_dependent" → "geographic" | | (75% / 4 ) |
| "human_capital" → "choice_dependent" | | (75% / 4 ) |

**Asia**

| | | |
|---|---|---|
| "lung-related" → "downstream effect" | | (100% / 4 ) |
| "downstream effect" → "clinical observation" | | (100% / 3 ) |
| "disease" → "downstream effect" | | (100% / 3 ) |
| "exogenous" → "disease" | | (100% / 3 ) |
| "exogenous" → "medical outcome" | | (100% / 3 ) |
| "exogenous" → "respiratory" | | (100% / 3 ) |
| "downstream effect" → "breathing" | | (66% / 3 ) |

**Lucas**

| | | |
|---|---|---|
| "respiratory" → "symptom" | | (100% / 5 ) |
| "heritable" → "downstream_effect" | | (100% / 4 ) |
| "risk_factor" → "accident_related" | | (100% / 3 ) |
| "disease" → "downstream_effect" | | (100% / 3 ) |
| "disease" → "symptom" | | (100% / 3 ) |
| "heritable" → "symptom" | | (100% / 3 ) |
| "respiratory" → "downstream_effect" | | (85% / 7 ) |

**Child**

| | | |
|---|---|---|
| "disease_mechanism" → "pathophysiology" | | (100% / 8 ) |
| "pathophysiology" → "measurement" | | (100% / 7 ) |
| "pathophysiology" → "oxygenation" | | (100% / 7 ) |
| "pathophysiology" → "blood_gas" | | (100% / 6 ) |
| "disease_mechanism" → "pulmonary" | | (100% / 6 ) |
| "etiology" → "disease_mechanism" | | (100% / 5 ) |
| "disease_mechanism" → "hemodynamic" | | (100% / 5 ) |

### Insurance

| | | |
|---|---|---|
| "demographic" → "socioeconomic_dependent" | (100% / 7 ) |
| "exogenous" → "vehicle_property" | (100% / 7 ) |
| "exogenous" → "vehicle_related" | (100% / 7 ) |
| "accident_related" → "downstream_consequence" | (100% / 6 ) |
| "demographic" → "age_dependent" | (100% / 6 ) |
| "age_dependent" → "risk_factor" | (100% / 6 ) |
| "socioeconomic_dependent" → "theft_related" | (100% / 6 ) |

### Alarm

| | | |
|---|---|---|
| "equipment" → "airway" | (100% / 8 ) |
| "root_cause" → "latent_physiology" | (100% / 8 ) |
| "cardiac" → "observed_indicator" | (100% / 7 ) |
| "equipment" → "measurement" | (100% / 7 ) |
| "equipment" → "observed_indicator" | (100% / 7 ) |
| "pathological_condition" → "latent_physiology" | (100% / 7 ) |
| "airway" → "latent_physiology" | (100% / 6 ) |

### Hailfinder

| | | |
|---|---|---|
| "context" → "observed_input" | (100% / 10 ) |
| "observed_input" → "derived_intermediate" | (100% / 8 ) |
| "observed_input" → "combined_aggregate" | (100% / 6 ) |
| "context" → "scenario_related" | (100% / 5 ) |
| "derived_intermediate" → "scenario_related" | (100% / 4 ) |
| "scenario_related" → "forecast_output" | (100% / 4 ) |
| "observed_input" → "cloud" | (100% / 3 ) |

### Hepar2

| | | |
|---|---|---|
| "liver_disease" → "laboratory_value" | (100% / 48 ) |
| "chronic_condition" → "laboratory_value" | (100% / 28 ) |
| "hepatitis" → "laboratory_value" | (100% / 27 ) |
| "liver_disease" → "liver_enzyme" | (100% / 19 ) |
| "biliary_system" → "laboratory_value" | (100% / 13 ) |
| "liver_disease" → "cholestatic_marker" | (100% / 12 ) |
| "hepatitis" → "liver_enzyme" | (100% / 12 ) |

### Win95Pts

| | | |
|---|---|---|
| "root_cause" → "intermediate_state" | (100% / 17 ) |
| "physical_hardware" → "intermediate_state" | (100% / 12 ) |
| "root_cause" → "functional_status" | (100% / 11 ) |
| "root_cause" → "data_flow" | (100% / 8 ) |
| "physical_hardware" → "functional_status" | (100% / 8 ) |
| "software" → "intermediate_state" | (100% / 8 ) |
| "physical_hardware" → "local_printing" | (100% / 8 ) |

Table 11: **Tag Relations for Claude 4.6.** We consider the most informative tag pairs per dataset for Claude 4.6 that have at least 3 edges of evidence. We list tag pair probability and support in brackets, sorted by accuracy. (A probability of 50% corresponds to a tag homogeneity of 0; 100% indicates full homogeneity.)

