# OpenReview forum: "Tags for DAGs: Graph Refinement with Meta-Informed Relations"
_TMLR — Under review for TMLR_

### Review · Reviewer_QkXi · 2026-06-22

**Summary Of Contributions:**

This paper proposes a tag-based method for refining CPDAGs produced by standard causal discovery algorithms. Each variable is assigned one or more semantic tags, and already-oriented edges are used to estimate directional preferences between tag pairs. These tag-level statistics are then used to orient remaining undirected edges when the evidence is sufficiently strong.

The main strength is the clean use of semantic information. The method does not ask an LLM to directly judge causal directions; it only uses LLMs or experts to annotate variables, while directionality is inferred from data-derived graph structure. This is a sensible hybrid design. The multi-tag formulation is also more flexible than prior single-type approaches.

The main weakness is the central transfer assumption: tag-direction statistics from already-oriented edges must be representative of unresolved edges. This may fail because CPDAG-oriented edges are selected by specific graph-theoretic patterns, not sampled uniformly from all causal edges. The empirical results are promising, but this failure mode is not directly tested.

**Audience:**

Yes

**Audience Explanation:**

The paper addresses a real limitation of observational causal discovery: many edges remain unoriented without additional information. The proposed tag layer is simple, interpretable, and relevant to researchers interested in causal discovery, LLM-assisted scientific workflows, and hybrid statistical-semantic methods. The work is especially interesting because it uses LLMs modestly as annotators rather than as causal oracles.

**Broader Impact Concerns:**

The main concern is overtrust. A graph with more oriented edges may look more actionable than the evidence supports, especially if semantic tags make the directions feel intuitive. This is risky in high-stakes domains such as medicine, finance, education, and policy.

There is also a risk that LLM-generated tags encode biased or overly coarse semantic categories, especially for demographic or socioeconomic variables. The paper should explicitly state that the method is a graph-refinement heuristic under assumptions, not automated causal truth discovery, and should recommend reporting evidence support, abstention rates, and sensitivity to tag perturbations in applied use.

**Claims And Evidence:**

Yes

**Claims Explanation:**

The core empirical claim is supported: across the evaluated benchmark datasets, the tagged variants generally improve over PC/GES and type-based baselines, especially with GES. The robustness experiments with graph faults and tag noise also support the claim that aggregating multiple tag relations can tolerate moderate noise.

However, the evidence is less complete for broader claims about reliability and abstract causal relation discovery. The method depends on tag distribution consistency, and the paper does not sufficiently test cases where already-oriented edges and unresolved edges have different tag-direction statistics. I would therefore view the results as convincing evidence for a useful heuristic, but not as evidence of broadly reliable causal abstraction.

**Requested Changes:**

Critical:
1. Add a direct stress test of the tag distribution consistency assumption. Construct or identify cases where already-oriented edges support one tag direction, while unresolved edges with the same tags follow the opposite or mixed direction. This would test the paper’s most important failure mode.
2. Separate oracle-tuned results from deployable results. The paper evaluates many configurations and reports best-performing settings. Please also report a fixed default setting, leave-one-dataset-out selection, or another realistic tuning protocol.
3. Clarify the calibration procedure for the conservativeness threshold. In particular, explain whether calibration uses only information available from the CPDAG or relies on ground-truth orientations in any experimental setting.

Important but not blocking:
1. Report the coverage-risk tradeoff: how many unresolved edges are oriented, and how accurate those oriented edges are as the threshold changes.
2. Add failure examples, not only successful tag relations. This would help readers understand when semantic tags mislead the method.
3. Qualify claims about abstract causal relations. The current evidence shows high tag homogeneity in some datasets, but does not establish stable domain-general abstract causal mechanisms.

---

### Review · Reviewer_PYnG · 2026-07-01

**Summary Of Contributions:**

This paper presents a novel and well-designed framework for causal orientation by leveraging multi-label soft probability statistics instead of conventional hard constraints. The proposed method effectively improves robustness against graph structure errors and LLM-induced label noise, while extensive experiments demonstrate consistent gains over strong baselines on key metrics such as SHD and F1. The ability to uncover high-level abstract causal relationships further highlights the potential of the proposed approach.

**Audience:**

Yes

**Audience Explanation:**

This paper sits at the intersection of causal inference, graphical models, and robust ML—core areas of TMLR's readership. It tackles the long-standing MEC identification problem, translates LLM-derived semantic tags into quantifiable probabilistic priors, and provides large-scale evidence of robust gains under noise. Although the i.i.d. assumption in theory may raise concerns, practitioners in causal representation learning, LLM-assisted discovery, and trustworthy AI will find its abstention-for-uncertainty philosophy and cross-LLM generalization findings of clear interest.

**Broader Impact Concerns:**

The method relies on LLMs for tag generation, which may inherit or amplify societal biases (e.g., gender or racial stereotypes) present in model training data. Since these tags directly influence causal direction decisions, such biases could propagate into the resulting causal graphs, potentially leading to discriminatory or unfair outcomes in downstream applications like healthcare or social policy.

**Claims And Evidence:**

Yes

**Claims Explanation:**

The paper provides strong empirical support for its central claim that multi-label soft statistics can effectively and robustly improve causal orientation. Extensive experiments across 11 datasets, seven LLMs, and over 60,000 evaluations consistently demonstrate superior performance over strong baselines, while robustness tests under graph structure errors and label noise are convincing. Although the convergence analysis in Appendix A relies on a relatively strong i.i.d. assumption, the breadth and depth of the experimental evidence sufficiently support the paper's main contributions.

**Requested Changes:**

1. The paper postulates Tag Distribution Consistency (Assumption 1) without specifying when it holds or breaks. The authors must clarify the necessary conditions (e.g., graph size, tag overlap, no domain shift) and provide concrete examples of failure modes.

2. No analysis is offered when Assumption 1 is violated. The paper should discuss whether the algorithm degrades gracefully or introduces systematic bias, along with a brief failure analysis to inform users of practical risks.

3. The decision rule uniformly averages all tag pair probabilities without justification. The authors should either provide a theoretical rationale for this choice or conduct an ablation comparing weighted averaging, Bayesian fusion, or attention mechanisms.

4. Recent LLM-assisted causal discovery methods (2023–2025) are only mentioned in the appendix, not in the main experiments. Including at least one or two of these in the primary comparison would more convincingly demonstrate the proposed method’s advantage over contemporary baselines.
5. The theoretical analysis assumes i.i.d. tag informative values for CLT convergence. However, tag pairs from the same variables are inherently correlated. The authors should either discuss the impact of such correlations on the guarantees or explicitly frame the analysis as an approximation rather than a strict proof.

---

### Review · Reviewer_5gM6 · 2026-07-03

**Summary Of Contributions:**

The authors extend type-based background knowledge to a flexible multi-tag setting, enabling richer and overlapping semantic annotations.
They introduce a probabilistic, aggregation-based decision rule with abstention, improving robustness to noisy tags and erroneous initial orientations. Additionally, the practical heuristics (minimum-evidence thresholds, greedy orientation with acyclicity checks, optional redirection) make the approach usable in real pipelines.

**Audience:**

Yes

**Audience Explanation:**

The approach is pragmatic and low-cost: tags are easier to elicit than precise edge directions and can be sourced from variable names/descriptions via LLMs. This makes the method attractive for practitioners wanting modest improvements over CPDAGs without heavy expert annotation.

**Broader Impact Concerns:**

no ethical concerns

**Claims And Evidence:**

Yes

**Claims Explanation:**

This paper proposes a practical and flexible extension of type-based background knowledge to overlapping, multi-tag annotations, using aggregated directional statistics from already-oriented edges to refine CPDAGs. The idea is intuitive, easy to implement, and empirically promising across a suite of standard benchmarks, with helpful robustness analyses. However, the theoretical support is mostly heuristic—variance reduction without a compelling argument for unbiasedness or identifiability—and key methodological details (especially α calibration) are under-specified and may entail leakage if ground truth is used. Optional redirection without CI/score checks also raises concerns.

To be specific,
1. The core estimator Q(Eij) is a straightforward averaging of pairwise tag-direction probabilities computed from directed edges. This is a reasonable heuristic, but it assumes the distribution of tag-direction relationships among compelled edges is representative of undirected edges (selection bias risk). Assumption 1 is intended to address this but is untestable and arguably unlikely to hold if orientation likelihood correlates with local structure (e.g., v-structures).
2. The Beta/CLT argument shows that averaging many tag-pair Bernoulli-like signals reduces variance as the number of tag pairs grows. However, this does not address estimator bias nor dependence among tag pairs (tags are not independent; tag pairs on the same edge share variables and context).
3. The optional edge redirection may compromise the faithfulness of the final graph to observed independences. Without an explicit check (e.g., re-running local CI tests or a score), this step risks trading beyond-MEC improvements for violations of the data-implied equivalence class.

It still has accurate, convincing and clear evidence, though it has room to improvement.

**Requested Changes:**

strengthen the work.

1. How many additional edges are oriented by your method on average per dataset, and with what precision? Please report coverage–accuracy tradeoffs (e.g., by varying α) and the abstention rate.
2. In the optional redirection step, how do you ensure that re-oriented edges remain consistent with the observed conditional independences? Would adding a local CI or score-based check help? Any empirical assessment of CI violations introduced by redirection?
3. Why not weight the per-tag-pair contributions by support (C_ab + C_ba) in Eq. 6, beyond a minimal-evidence threshold? Did you try Laplace/Beta priors to stabilize low-support p_ab?
4. Could you formalize the “homogeneity” measure and report its correlation with orientation accuracy and with the number of tags per variable? Is there an optimal range for tag granularity?

---

### Author Response · Authors · 2026-07-07
**Request for extension till 31/07**

Dear EICs and AE,

Thank for handling our submission and ensuring a smooth review process. We are writing this message to request for a 2 week extension for the rebuttal phase due to the ongoing ICML conference and the authors are travelling.

Thank you for your help.

Regards,

Authors

---

> ### Author Response · Authors · 2026-07-18
>
> Due to the concurrent ICML conference and current summer vacancies, we would like to kindly ask for your understanding that our comprehensive response to the detailed reviews is not yet finished and will arrive by 31.07. at the latest.